# Endothelial sensing of AHR ligands regulates intestinal homeostasis

Benjamin G. Wiggins[1,2 ✉], Yi-Fang Wang[2,5], Alice Burke[1,2,5], Nil Grunberg[1,2], Julia M. Vlachaki Walker[1,2], Marian Dore[2], Catherine Chahrour[2], Betheney R. Pennycook[1,2], Julia Sanchez-Garrido[3], Santiago Vernia[1,2], Alexis R. Barr[1,2], Gad Frankel[3], Graeme M. Birdsey[4], Anna M. Randi[4] & Chris Schiering[1,2 ✉]

Endothelial cells line the blood and lymphatic vasculature, and act as an essential physical barrier, control nutrient transport, facilitate tissue immunosurveillance and coordinate angiogenesis and lymphangiogenesis[1,2]. In the intestine, dietary and microbial cues are particularly important in the regulation of organ homeostasis. However, whether enteric endothelial cells actively sense and integrate such signals is currently unknown. Here we show that the aryl hydrocarbon receptor (AHR) acts as a critical node for endothelial cell sensing of dietary metabolites in adult mice and human primary endothelial cells. We first established a comprehensive single-cell endothelial atlas of the mouse small intestine, uncovering the cellular complexity and functional heterogeneity of blood and lymphatic endothelial cells. Analyses of AHR-mediated responses at single-cell resolution identified tissue-protective transcriptional signatures and regulatory networks promoting cellular quiescence and vascular normalcy at steady state. Endothelial AHR deficiency in adult mice resulted in dysregulated inflammatory responses and the initiation of proliferative pathways. Furthermore, endothelial sensing of dietary AHR ligands was required for optimal protection against enteric infection. In human endothelial cells, AHR signalling promoted quiescence and restrained activation by inflammatory mediators. Together, our data provide a comprehensive dissection of the effect of environmental sensing across the spectrum of enteric endothelia, demonstrating that endothelial AHR signalling integrates dietary cues to maintain tissue homeostasis by promoting endothelial cell quiescence and vascular normalcy.

Recent advances have led to an increased appreciation of endothelial cell cellular and functional diversity within vascular beds and have highlighted tissue origin as a critical determinant of endothelial cell heterogeneity[3-6]. Endothelial cells are long lived and exist in a state of functional quiescence, enabling them to be rapidly activated by inflammatory stimuli or tissue injury[1]. The signals regulating endothelial cell quiescence and vascular normalcy at the intestinal barrier, which is constantly exposed to diverse commensal microorganisms, pathogens and dietary factors, remain elusive.

The AHR, a ligand-activated transcription factor that is capable of sensing dietary micronutrients and microbial metabolites, has an essential role in maintaining intestinal homeostasis[7]. Genetic deficiency in AHR is associated with compromised intestinal barrier integrity, altered microbiota composition and dysregulated host responses to pathogens and injury[8-11]. AHR deficiency results in a number of developmental vascular defects in the liver, heart, kidney and eye[12-14]. Although it is known that endothelial cells can respond to AHR ligands in vitro[15-17], the role of AHR in the enteric vasculature is unknown.

Here we used single-cell RNA sequencing (scRNA-seq) to dissect the transcriptomic responses to AHR pathway activation across blood and lymphatic endothelial cell populations of the adult mouse small intestine, revealing substantial cellular heterogeneity within the enteric vascular bed. AHR signalling in mouse and human cells limited endothelial cell activation via inhibition of proliferative and pro-inflammatory pathways; whereas AHR deficiency or lack of dietary AHR ligands resulted in endothelial activation, VEGFA-dependent proliferation, and contributed to an increased susceptibility to intestinal bacterial infection. Our study demonstrates a requirement for AHR-mediated environmental sensing in enteric endothelial cells for the maintenance of endothelial quiescence.

## Enteric vasculature at single-cell resolution

To gain a deeper understanding of cellular complexity of the small intestine blood and lymphatic vasculature, we performed scRNA-seq on total small intestine endothelial cells. Wild-type mice were acutely exposed

[1]Institute of Clinical Sciences, Faculty of Medicine, Imperial College London, London, UK. [2]MRC London Institute of Medical Sciences, London, UK. [3]Department of Life Sciences, Imperial College London, London, UK. [4]National Heart and Lung Institute, Imperial College London, London, UK. [5]These authors contributed equally: Yi-Fang Wang, Alice Burke. ✉e-mail: bwiggins@ic.ac.uk; cschieri@ic.ac.uk

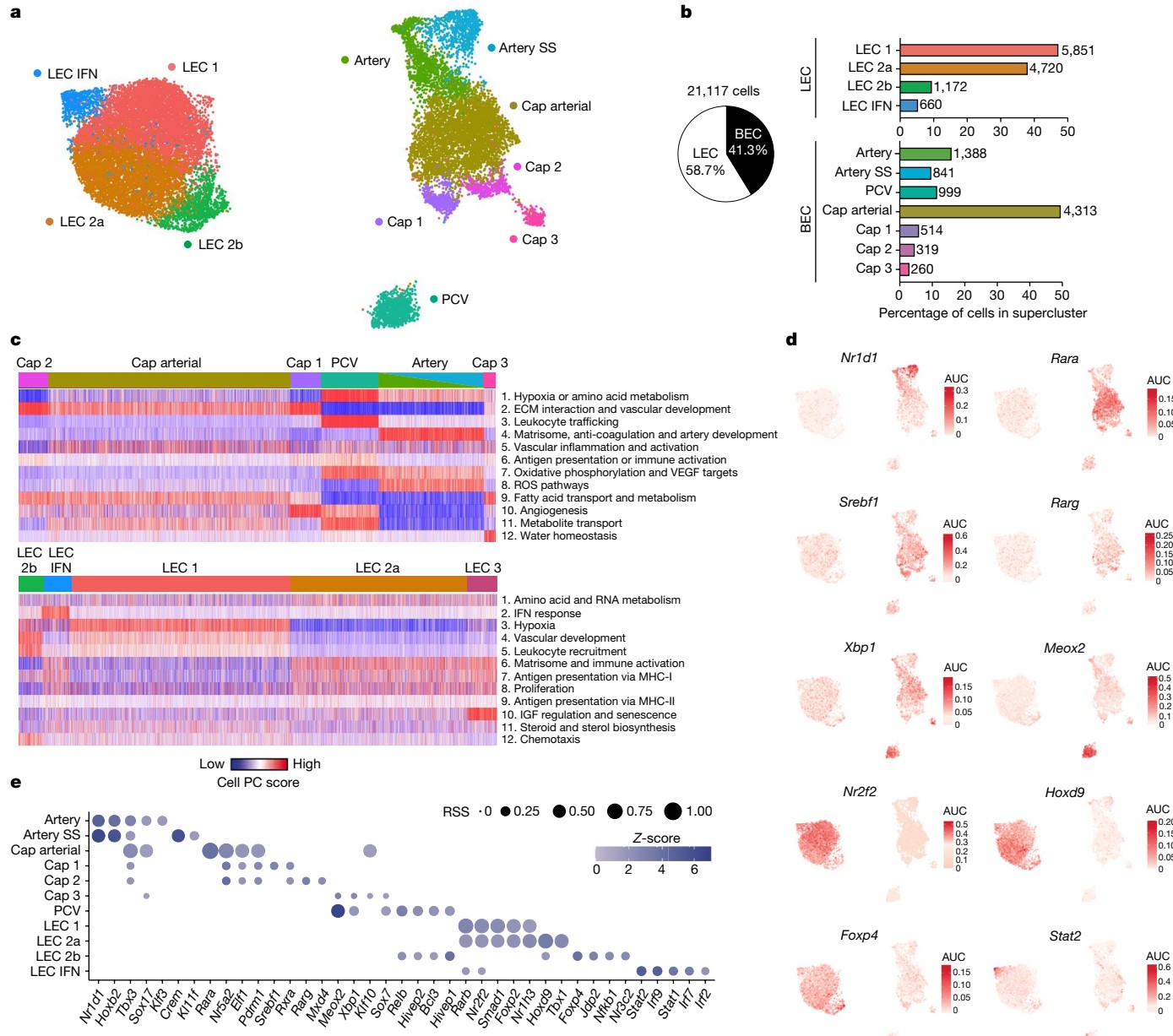

**Fig. 1 | Single-cell transcriptomics reveals the cellular complexity of enteric vasculature. a**, Uniform manifold approximation and projection (UMAP) of small intestine endothelial cells. Artery SS, artery shear stress; cap, capillary. **b**, Representation of BEC and LEC superclusters (pie chart) and supercluster breakdown (bar charts). Cell numbers given at the end of bars. **c**, Pagoda2 analysis of BEC and LEC subclusters. The heat map shows principal component (PC)/aspect scores for each cell assigned on the basis of the level of statistical enrichment within curated endothelial-related input gene sets (see Methods and Supplementary Table 3). Gene sets are clustered together on the basis of similarities within constituent genes and similar patterns of cell separation to create aspects (heat map rows; see also Supplementary Table 4). Top 12 aspects are annotated manually based on top constituent pathways. ECM, extracellular matrix; IGF, insulin-like growth factor; ROS, reactive oxygen species. **d**, Transcription factor network activity area under the curve (AUC) score distributions of selected top enriched regulons for each cluster following SCENIC analysis. **e**, Top 5 enriched regulons, by regulon specificity score (RSS), for each of the clusters. RSS and normalized regulon activity (z-score) are shown.

(for 3 h) to the AHR ligand 6-formylindolo(3,2-*b*)carbazole (FICZ) or vehicle, and total endothelial cells (CD31⁺CD45⁻) were sorted and sequenced (Extended Data Fig. 1a). After filtering, doublet exclusion and removal of contaminant clusters (Extended Data Fig. 1b, Supplementary Table 1 and Methods), our dataset comprised 21,117 high quality enteric endothelial cells. Analysis of the combined dataset (vehicle and ligand), revealed 11 endothelial clusters, clearly demarcated across two superclusters: lymphatic endothelial cells (LECs) and blood endothelial cells (BECs) (Fig. 1a,b). Analyses of known markers

of BECs (*Bcam*, *Esam*, *Ly6c1*, *Ly6a*, *Cd36*, *Sox17*, *Nrp1*, *Vwf* and *Plvap*) and LECs (*Prox1*, *Lyve1*, *Pdpn*, *Thy1*, *Mmrn1*, *Prss23*, *Fxyd6*, *Cp* and *Nrp2*) confirmed supercluster identity (Extended Data Fig. 1c,d), and our data contained negligible contamination from epithelial cells (marked by *Epcam*), mural cell types (*Acta2* and *Pdgfrb*), fibroblasts (*Col1a1*), erythrocytes (*Hba-a1*, *Hba-a2* and *Hbb-bs*) or immune cells (*Ptprc*)[5] (Extended Data Fig. 1e). Cluster annotation was based on known marker gene expression[3,4], beginning with nomenclature from a murine endothelial atlas[5]. Hierarchical clustering showed clear demarcation at the gene

level, with LEC 2a and LEC 1 being the most similar (Extended Data Fig. 1f). LEC clusters were more congruent with capillary than larger collecting vessel or valve lymphatics identified in mesenteric adipose[4] (Extended Data Fig. 1g).

Alongside the examination of enriched marker genes within each cluster (Extended Data Fig. 2a and Supplementary Table 2), we performed pathway and gene set overdispersion analysis (Pagoda2) on each supercluster, using endothelial-specific gene sets as input (Supplementary Table 3). Grouping by aspects (groups of similar gene sets based on gene components and cell separation across the data) enabled us to test how well different endothelial functions mapped onto our clustering[18] (Fig. 1c, Extended Data Fig. 2b and Supplementary Table 4). Additionally, to uncover novel active gene regulatory networks with important roles in the cellular identity and differentiation of our clusters, we used single-cell regulatory network inference and clustering[19] (SCENIC). We identified 167 unique regulons across the data and revealed enriched regulon activity for each cluster (Fig. 1d,e, Extended Data Fig. 2c,d and Supplementary Table 5).

As expected, post-capillary venules (PCVs) were best defined by genes and pathways involved in leukocyte trafficking, whereas artery development was most evident in arteries (Fig. 1c). We also detected an artery shear stress cluster (expressing *Slc6a6*, *S100a4* and *Pi16* and enriched for *Crem* and *Klf11* regulons) that was previously identified only in the brain[5] (Fig. 1d,e and Extended Data Fig. 2a). As a group, capillary endothelial cells were enriched for fatty acid transport and metabolism, whereas large vessel endothelial cells together show increased oxidative phosphorylation. The major BEC population were capillary arterial cells (50%), defined by high activity of *Rara*, suggestive of a role for the vitamin A metabolite retinoic acid in enteric capillary homeostasis (Fig. 1d,e). Capillary 1 cells were enriched for angiogenic or tip cell genes[3,20] (*Apln*, *Chrm2* and *Car4*), proliferative and chemotaxis pathways and *Srebf1* activity linked previously to promotion of VEGF-mediated angiogenesis[21], all in keeping with its identification as a novel gut angiogenic endothelial subset. Conversely, capillary 2 markers included *Ces2e*, *Ramp3* and *Rbp7* and presented with specific *Rarg* activity. Capillary 3 cells were reminiscent of previously described Aqp7[+] capillaries[5] and showed a preference for aquaporin-mediated water transport.

Our data resolved enteric LEC into four novel clusters. The major constituents were LEC 1 and LEC 2a, which share similar canonical LEC markers (for example, *Nrp2*, *Cp* and *Mmrn1*), but were separated by enrichment for hypoxia gene sets in LEC 1 and enrichment for major histocompatibility complex class I (MHC-I)-mediated presentation gene sets in LEC 2a (Fig. 1c and Extended Data Fig. 2a). We identified a clear interferon (IFN) response signature in IFN LECs at marker (*Ifit1–3*, *Rsad2* and *Isg15*), aspect (IFN response) and regulon (*Stat1–2*, *Irf2* and *Irf7*) levels. Of note, we also detected an immunomodulatory LEC population, LEC 2b, defined by metallothionein gene expression (*Mt1* and *Mt2*), and akin to PCVs, with enrichments in leukocyte trafficking pathways and many similar transcription factor activities, including NF-κB signalling[22] (*Relb* and *Hivep2*) (Fig. 1c). Collectively, these data deconstruct enteric endothelial heterogeneity at the level of marker, biological role and transcription factor-driven identity.

## Endothelial sensing of AHR ligands

Dividing the scRNA-seq dataset into vehicle- and AHR ligand-treated conditions, we noted that acute exposure to AHR ligand did not alter the relative proportions of endothelial cell subtypes (Fig. 2a and Extended Data Fig. 3a). Following ligand administration, we observed a marked and broad induction of the AHR-specific target gene *Cyp1a1* across all endothelial cell subtypes, and the additional AHR target *Cyp1b1* in all clusters except capillary 3 (Fig. 2b and Extended Data Fig. 3b,c), whereas in vehicle-treated mice, only a low proportion of

cells in capillary 1, LEC 1, LEC IFN expressed *Cyp1a1*. This demonstrates that sensitivity to AHR ligands is a universal feature of gut endothelial cell subtypes.

Next, we verified this finding using a *Cyp1a1* fate-reporter mouse strain (a reporter of AHR activity through eYFP induction activated via Cre recombinase in the mouse *Cyp1a1* locus[23]). Following administration of AHR ligand 3-methylcholanthrene (3-MC), both BEC (CD31[+]PDPN[−]) and LEC (CD31[+]PDPN[+]) showed AHR responsiveness (Fig. 2c and Extended Data Fig. 3d). Whole-mount gut imaging of ligand-treated *Cyp1a1*-reporter mice revealed that AHR ligand sensing appeared universal throughout blood and lymphatic vessels (Fig. 2d and Extended Data Fig. 3e,f). Further, there was no difference in ligand-induced *Cyp1a1* expression between duodenum, jejunum and ileum (Extended Data Fig. 3g,h). Indeed, beyond the small intestine, BEC AHR ligand sensitivity was noted in colon, liver, lung, spleen, kidney and adipose tissues, and LEC sensitivity was observed in colon and liver (Extended Data Fig. 3i). Together, these data suggest that AHR responsiveness is a conserved feature of enteric endothelial cells across vessel types along the length of the intestine, as well as endothelial cells of other organs.

We next analysed differentially expressed genes (DEGs) between ligand- and vehicle-treated cells within each scRNA-seq cluster. Alongside increased canonical AHR pathway genes (*Cyp1a1*, *Cyp1b1* and *Tiparp*), the changes in DEGs after ligand treatment were consistently associated with negative regulation of proliferative and angiogenic or lymphangiogenic processes. These included increased *Cdkn1a* and *Zfp36l1*, and decreased *Sox18* and *Nrp2* (refs. 24–27) (Fig. 2e, Extended Data Fig. 4a and Supplementary Table 6). Transcriptional responses in BEC clusters were predominantly unique to each cluster, whereas responses in LEC clusters were more similar to those in other LECs (Extended Data Fig. 4b). Twelve DEGs were shared among all endothelial cell clusters. Shared up regulated genes included the anti-proliferative *Cdkn1a*, the oxidative stress protector gene *Txnip* and the transcription factor gene *Klf9*, which is linked to quiescence in other cell types[28,29], while shared downregulated genes included the key endothelial motility gene *Marcks*[30] (Extended Data Fig. 4c–e and Supplementary Table 7). Following ligand treatment, we observed consistent downregulation of pathways related to angiogenesis, vasculogenesis, endothelial cell proliferation and endothelial cell migration among BEC clusters, whereas LEC clusters displayed reduced responses to TGFβ, inflammatory signalling (IL-1β and lipopolysaccharide (LPS)), and a consistent inhibition of cell migration and growth factor signalling (Fig. 2f, Extended Data Fig. 4f and Supplementary Table 8). This combined downregulation of angiogenic, inflammatory and TGFβ pathways supports the notion that AHR ligands provide key homeostatic environmental cues to ensure endothelial cell quiescence at the intestinal barrier[1].

## AHR regulates endothelial proliferation

To determine how a lack of responsiveness to AHR ligands affects intestinal endothelial cell function, we generated an inducible endothelial cell-specific *Ahr*-deficient mouse model—*Cdh5(PAC)*[creERT2]*Ahr*[fl/fl NuTRAP] (EC[ΔAhr]). Following tamoxifen treatment (five injections) in adult mice, we observed specific and efficient Cre induction in intestinal BECs and LECs (Extended Data Fig. 5a), with unchanged intestinal immune cell infiltrate in the small intestinal lamina propria during tamoxifen treatment (Extended Data Fig. 5b). To understand endothelial cell-specific transcriptomic changes, we administered EC[ΔAhr] and AHR wild-type (EC[WT]) control mice with short-term FICZ treatment (3 h) before sorting and bulk RNA sequencing (RNA-seq) of small intestine BECs and LECs (Extended Data Fig. 5c). *Ahr*-deficient BEC displayed differential expression of 664 genes, including a prominent downregulation of AHR target genes (*Cyp1a1*, *Cyp1b1*, *Tiparp*, *Nqo1* and *Ahrr*), indicative of a lack of responsiveness to AHR ligand stimulation (Fig. 3a and Supplementary

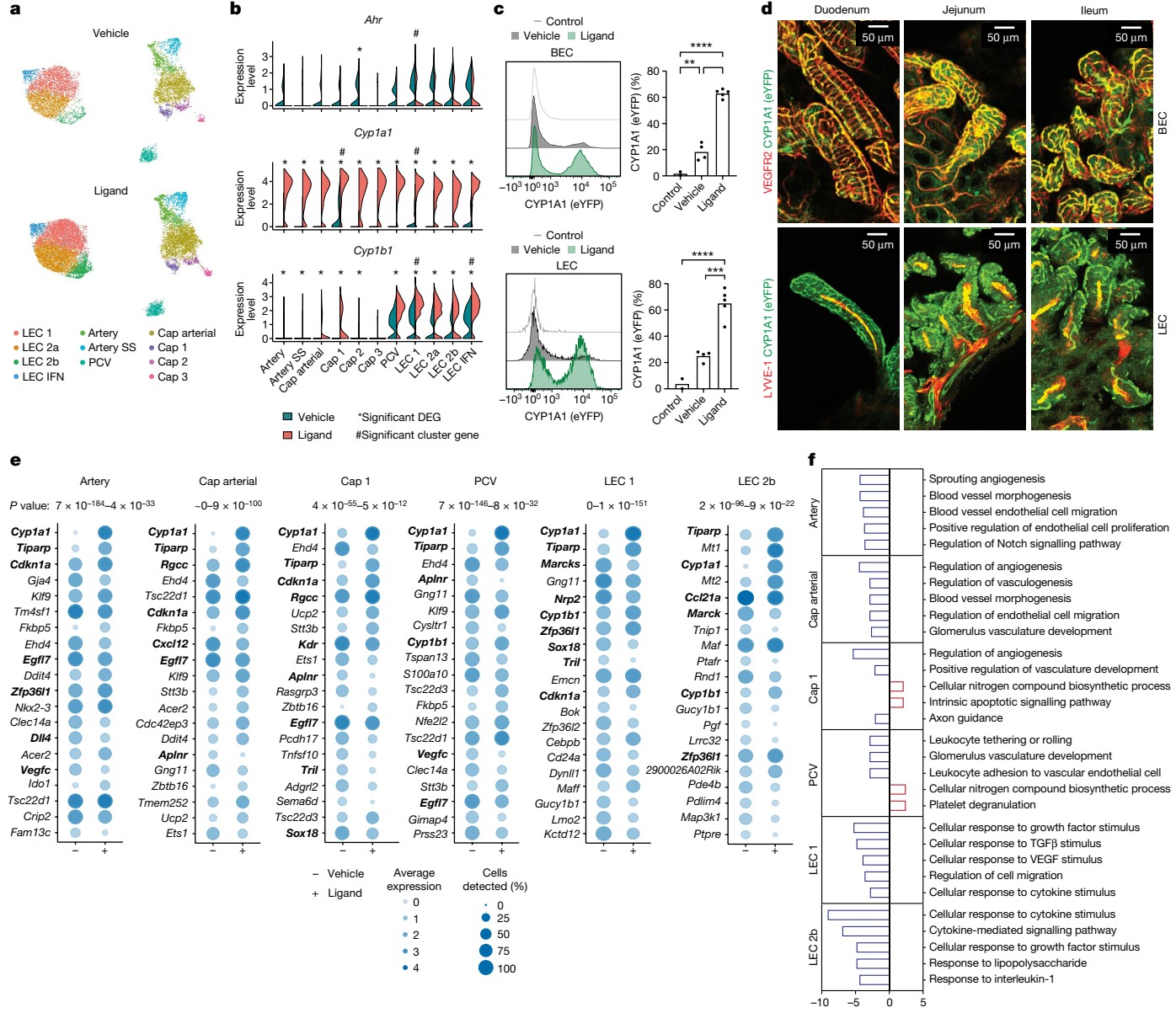

**Fig. 2 | AHR activation promotes vasculoprotective responses in endothelial subtypes. a**, UMAP plots split by condition (vehicle or ligand-treated). **b**, Expression of *Ahr*, *Cyp1a1* and *Cyp1b1* across the clusters. *Significant DEG (adjusted *P* < 0.05); #Conserved marker enriched in indicated cluster (adjusted *P* < 0.05). **c**, *Cyp1a1*-eYFP expression within BEC or LEC isolated from *Cyp1a1*-reporter mice following treatment with the AHR ligand 3-MC (*n* = 5), vehicle (*n* = 4) or (3-MC-treated) non-reporter controls (*n* = 2). Bars show mean and symbols represent individual mice. **d**, Representative whole-mount gut staining in *Cyp1a1*-reporter mice 5 days after 3-MC administration (4–6 images per

segment). **e**, Dot plots of top 20 differentially expressed (DE) genes for six selected scRNA-seq clusters in ligand-treated and vehicle-treated conditions, sorted by adjusted *P* value. The adjusted *P* value range for each cluster is shown above the plots. Genes related to the canonical AHR pathway or proliferation are in bold. **f**, Top 5 enriched gene sets in ligand-treated mice for the six indicated clusters. Upregulated and downregulated gene sets were tested separately and the top 5 combined enrichment scores are shown. *P* values calculated by Wilcoxon Rank Sum tests (**b**,**e**), one-way ANOVA with Tukey's multiple comparisons tests(**c**) or Fisher's exact tests (**f**).

Table 9). *Ahr*-deficient BECs showed marked enrichment for pathways relating to inflammatory response, mesenchymal transition, angiogenesis, cell motility and leukocyte recruitment (Fig. 3b,c and Supplementary Table 10). We identified 1,215 DEGs between EC$^{\Delta Ahr}$ LECs and their EC$^{WT}$ LEC counterparts (15% of which were shared with BEC DEGs), with enrichments in oxidative phosphorylation, reactive oxygen species, MYC targets and mesenchymal transition (Extended Data Fig. 5d–f). Combined, these data indicate that AHR is a key component of endothelial cell quiescence, regulating both angiogenic and inflammatory activation processes in tandem.

To test the proliferative regulation of endothelial cells in EC$^{\Delta Ahr}$ mice in vivo, we first sub-optimally deleted *Ahr* in EC$^{\Delta Ahr}$ mice (with a single tamoxifen dose) and made use of the Cre-induced fluorescent tagging in this model to compare AHR-sufficient (eGFP$^-$) with AHR-deficient (eGFP$^+$) endothelial cells in the same mice. To analyse proliferation, we subjected mice to in vivo 5-ethynyl-2'-deoxyuridine (EdU) labelling over two weeks. We observed a small but significant increase in endothelial cell proliferation in enteric BECs and LECs in the absence of AHR ligand sensing (Fig. 3d and Extended Data Fig. 5g). This higher proliferative capacity of AHR-deficient BECs was maintained even following provision

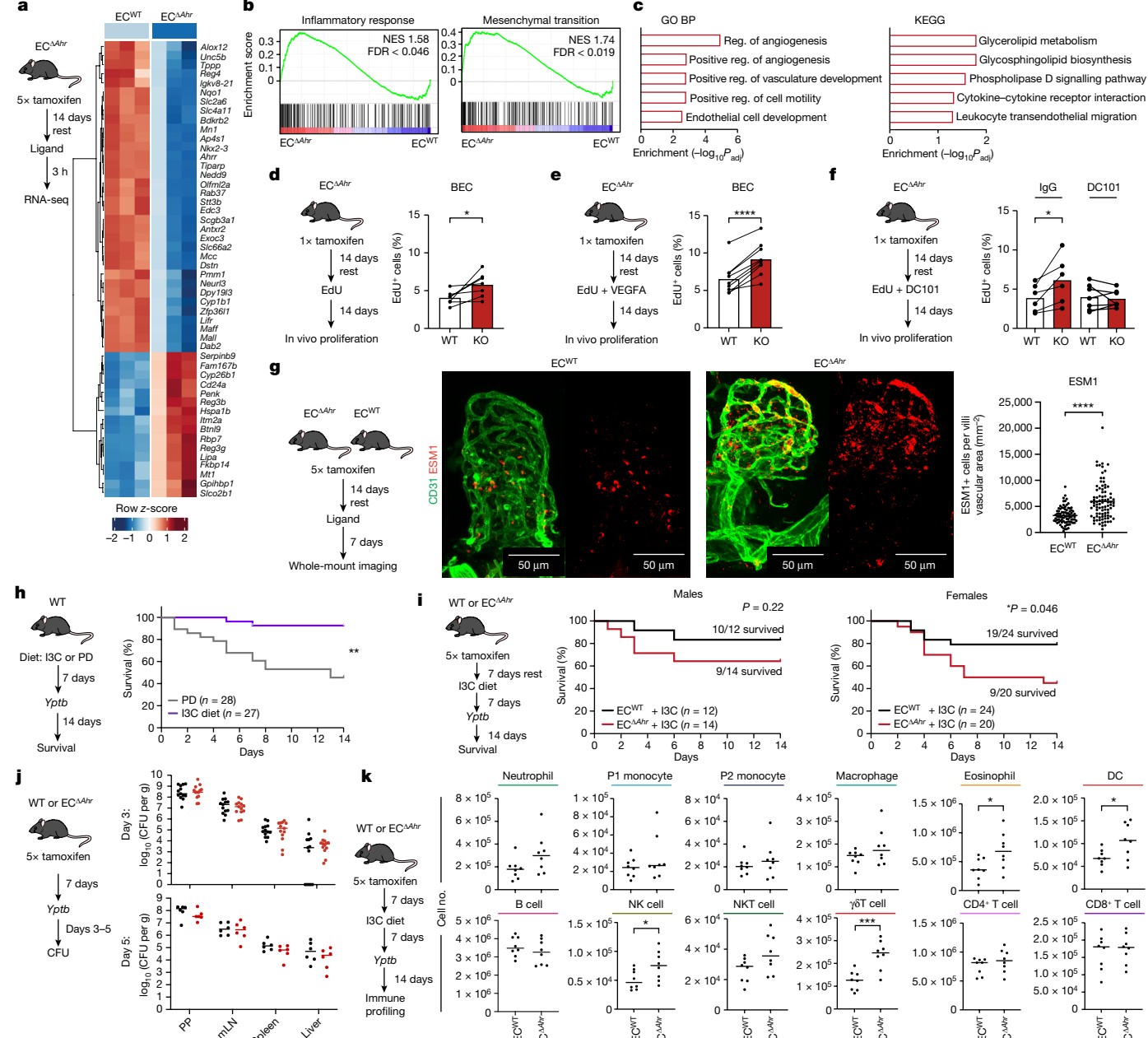

**Fig. 3 | AHR ligands act directly on endothelial cells to promote quiescence and anti-inflammatory programmes. a**, Sorted small intestine BECs from FICZ-treated EC[WT] and EC[ΔAhr] analysed by RNA-seq. Relative expression of top 50 DEGs. **b**, Barcode plots of gene set enrichment analysis (GSEA) on selected Hallmark gene sets. FDR, false discovery rate; NES, normalized enrichment score. **c**, BEC top 5 biological processes (BP) and KEGG gene sets upregulated in EC[ΔAhr] compared with EC[WT]. Reg., regulation. **d**–**f**, Proliferation (as percentage of EdU[+] cells) among wild-type and AHR-deficient (KO) BECs within the same mice following single-dose tamoxifen treatment and 14 days after feeding with EdU: at homeostasis (**d**; $n = 7$ per group), following 2-week VEGFA administration (**e**; $n = 9$ per group) and following 2-week treatment with VEGFR2-blocking antibody (DC101) or IgG control (IgG) (**f**; $n = 6$–7 per group). **g**, ESM1 expression within villi vasculature (CD31[+] cells) in the small intestine of EC[ΔAhr] or EC[WT] mice analysed seven days after treatment with 3-MC. Representative images (left) and quantification (right) of ESM1[+] cells normalized to villi vasculature area between groups (EC[WT] $n = 85$ villi, EC[ΔAhr] $n = 88$ villi). Points represent individual villi combined from 4 mice per group and bar height represents mean. **h**, Survival

curve comparing *Yptb*-infected wild-type mice fed with purified diet (PD; $n = 28$) or purified diet containing I3C (I3C diet; $n = 27$). Data combined from 4 individual experiments ($5 \times 10^7$ colony-forming units (CFU) per mouse). **i**, Survival curves of I3C-fed EC[ΔAhr] and EC[WT] male and female mice after *Yptb* infection ($5 \times 10^7$ CFU/mouse). Data combined from two or three independent cohorts. The proportion of surviving mice is shown. **j**, *Yptb* CFU number in 4 tissues 3 or 5 days after infection of EC[ΔAhr] and EC[WT] mice with $5 \times 10^8$ CFU per mouse. Dots show individual mice, and lines show mean values (day 3, $n = 13$ per group, 2 independent experiments; day 5, $n = 6$ per group, 1 independent experiment). mLN, mesenteric lymph node; PP, Peyer's patches. **k**, Immune cell profiling in small intestine lamina propria 3 days after infection with $5 \times 10^7$ CFU *Yptb* per mouse. Data show total cell numbers of ten immune cell populations. $n = 8$ per group. Dots represent individual mice and lines show means. Population underline colours indicate gating origin (Extended Data Fig. 7a and Extended Data Fig. 8d). DC, dendritic cell; NK, natural killer. *P* values calculated by Fisher's exact tests (**c**), paired *t*-tests (**d**–**f**), unpaired *t*-tests (**g**,**j**–**k**) or Gehan–Breslow–Wilcoxon tests (**h**,**i**).

of recombinant vascular endothelial growth factor A (VEGFA), a potent vascular mitogen (Fig. 3e and Extended Data Fig. 5h), demonstrating the importance of the AHR in promoting enteric endothelial cell proliferative restraint during homeostasis. To test whether this enhanced proliferation phenotype was owing to increased VEGFA sensitization, we used VEGFR2 blockade (with the DC101 antibody) concomitant with EdU feeding (Fig. 3f and Extended Data Fig. 5i). Although the proliferative advantage of AHR-deficient LECs was maintained following DC101 treatment, the difference between AHR-sufficient and AHR-deficient BECs was lost, suggestive of a mechanism whereby AHR restricts VEGFA signalling. These combined data support the view of AHR as a proliferative rheostat in the gut for blood endothelial cell homeostasis.

At homeostasis, endothelial cell AHR deficiency altered neither the gut immune cell composition (Extended Data Fig. 6a,b) nor the inflammatory activation profile of intestinal epithelial cells (assessed by expression of MHC-II, ICAM-1 and CD74 (refs. 31–33)) (Extended Data Fig. 6c). Whole-tissue RNA-seq analysis of small intestine from EC$^{\Delta Ahr}$ and EC$^{WT}$ mice following 3 h of FICZ treatment revealed very few differences between the groups (12 DEGs; Extended Data Fig. 6d,e). Whole-mount imaging of 3-MC-treated EC$^{\Delta Ahr}$ and EC$^{WT}$ mice (after a full five doses of tamoxifen) illustrated that endothelial AHR signalling did not impact villi blood vascular density or branching, villus vascular cage height or lacteal length (Extended Data Fig. 6f). To study intestinal vascular integrity, we again treated mice with 3-MC for 1 week before injecting 100-nm fluorescent microspheres intravenously and analysed their tissue dissemination after 5 min of circulation time. As previously reported, substantial leakage was observed from MADCAM1$^+$ submucosal venules with minimal leakage in villi or crypt vasculture[34], but this did not differ between the two groups within any vessel (Extended Data Fig. 6g). However, in agreement with in vivo proliferation detected by EdU incorporation (Fig. 3d–f), and tissue-wide enrichment of *Esm1* mRNA (Extended Data Fig. 6d,e), the expression of the tip cell marker and VEGFA target ESM1 (ref. 34) was enriched in EC$^{\Delta Ahr}$ mice (Fig. 3g). Together, these data suggest that endothelial AHR activation in the adult gut at homeostasis primarily functions to limit endothelial proliferation and angiogenesis[35].

## Endothelial AHR restrains inflammatory responses

Next, to understand how dietary AHR ligands influence endothelial cell-intrinsic responses to inflammation, EC$^{\Delta Ahr}$ mice were first fed ad libitum with diet containing AHR pro-ligand indole-3-carbinol (I3C), a vegetable-derived phytochemical converted into the high-affinity AHR ligands 3,3-diindolylmethane (DIM) and indolo[3,2b]carbazole (ICZ) by exposure to stomach acid. After one week, we challenged the mice with LPS or vehicle (PBS). Using suboptimal *Ahr* depletion, we observed increased expression of the inflammatory markers ICAM-1, VCAM-1, PD-L1 and BST2, the fatty acid transporter and pro-inflammatory mediator CD36, and the proliferative markers CD105 and CD24 (refs. 36–38) in *Ahr*-deficient BECs from LPS-treated mice (Extended Data Fig. 7a). CD80 expression was low in BECs, but the expression of CD86 was enhanced in AHR-deficient endothelial cells following LPS treatment (Extended Data Fig. 7a). Notably, inflammatory challenge was required to reveal these AHR-linked expression changes, as in vehicle-treated mice BST2, CD86 and CD105 were the only gut endothelial cell activation markers that were increased in AHR-deficient endothelial cells (Extended Data Fig. 7a). Similarly, in LECs, LPS treatment was required to reveal the full extent of these expression changes, with enhanced expression of ICAM-1, VCAM-1, MHC-II and BST2, but repressed inhibitory ligand PD-L1 in AHR-deficient LECs (Extended Data Fig. 7b). Notably, endothelial AHR deficiency promoted large increases in the expression of key endothelial cell inflammatory makers (ICAM-1, VCAM-1 for BECs and MHC-II for LECs) similar to those induced by LPS, demonstrating the anti-inflammatory potency of AHR signalling in endothelial cells.

To determine whether endothelial AHR influences the outcome of enteric infection, we studied responses to the enteric pathogen *Yersinia pseudotuberculosis* (*Yptb*). Our data demonstrate that *Ahr* germline-deficient mice are highly susceptible to *Yptb* infection, with markedly reduced survival and increased bacterial load in peripheral organs compared with wild-type controls (Extended Data Fig. 8a–c). Correspondingly, wild-type mice fed an I3C diet rich in AHR pro-ligands showed increased survival compared to wild-type mice fed a purified diet low in AHR ligands (Fig. 3h).

To understand whether dietary AHR ligands mediate these protective effects through endothelial cells, we infected EC$^{\Delta Ahr}$ and EC$^{WT}$ mice on I3C diet with *Yptb*. Our data suggest that endothelial AHR partially contributes to protection from *Yptb* infection, with statistical significance reached in female mice (Fig. 3i). Unlike in the global *Ahr*$^{-/-}$ mice, there were no clear differences in bacterial dissemination to the spleen or liver, and no difference in the bacterial load in Peyer's patches (Fig. 3j), suggestive of a role for vascular AHR in promoting disease tolerance rather than gut vascular barrier integrity or direct anti-bacterial immunity[39,40]. Accordingly, EC$^{\Delta Ahr}$ displayed increased eosinophil, dendritic cell, natural killer cell and γδ T cell abundance in the gut lamina propria three days after *Yptb* infection, whereas numbers of putatively resident populations (macrophages) and adaptive immune cells remained unchanged. (Fig. 3k and Extended Data Fig. 8d). Concomitant profiling of inflammatory markers in BEC showed the most notable increases in VCAM-1, BST2 and CD86 expression, suggesting that the increased inflammatory endothelial cell phenotype in EC$^{\Delta Ahr}$ mice contributes directly to the altered immune composition (Extended Data Fig. 8e). Together, our data suggest a role for endothelial AHR in promoting disease tolerance to enteric infection through modulating intestinal immune composition to limit inflammation.

## AHR evokes quiescence in human endothelial cells

Finally, to ascertain whether the observed vasculoprotective programmes translate to human endothelial cells, we cultured primary human umbilical vein endothelial cells (HUVECs) with AHR ligand FICZ or vehicle control. Exposure of HUVECs to FICZ led to transient AHR pathway activation (Extended Data Fig. 9a) and RNA-seq revealed the full spectrum of AHR-regulated genes in HUVECs (Fig. 4a and Supplementary Table 11). Exposure of HUVECs to AHR ligand promoted transcriptional signatures associated with endothelial cell quiescence while inhibiting cell proliferation (Fig. 4b). Further assessment of endothelial cell proliferation by flow cytometry revealed an increase in the frequency of cells in G0/G1, and a corresponding decrease of cells in S phase upon acute (6 h) ligand exposure (Fig. 4c). Conversely, *AHR* knockdown in HUVECs had the opposite effect, with fewer cells in G0/G1 and more in S phase (Extended Data Fig. 9b). Quantitative single-cell immunofluorescence revealed that FICZ-stimulated HUVECs contained fewer cells in S-phase (EdU$^+$), decreased E2F protein and phosphorylated retinoblastoma protein (Rb) that promote cell cycle progression, and increased expression of the cell cycle inhibitor p27 (Fig. 4d,e and Extended Data Fig. 9c).

Finally, following stimulation with LPS, we observed that the induced inflammatory response was significantly dampened in the presence of FICZ, demonstrating the potent anti-inflammatory potential of AHR ligands (Fig. 4f). Together, our data in human endothelial cells are consistent with those seen in mice, lending further support to the conserved role of AHR ligands as important environmental cues for the maintenance of endothelial quiescence.

## Discussion

Activation of the blood and lymphatic vasculature that supply and drain the gut must be tightly regulated to preserve tissue homeostasis and prevent aberrant inflammatory responses. In this study we

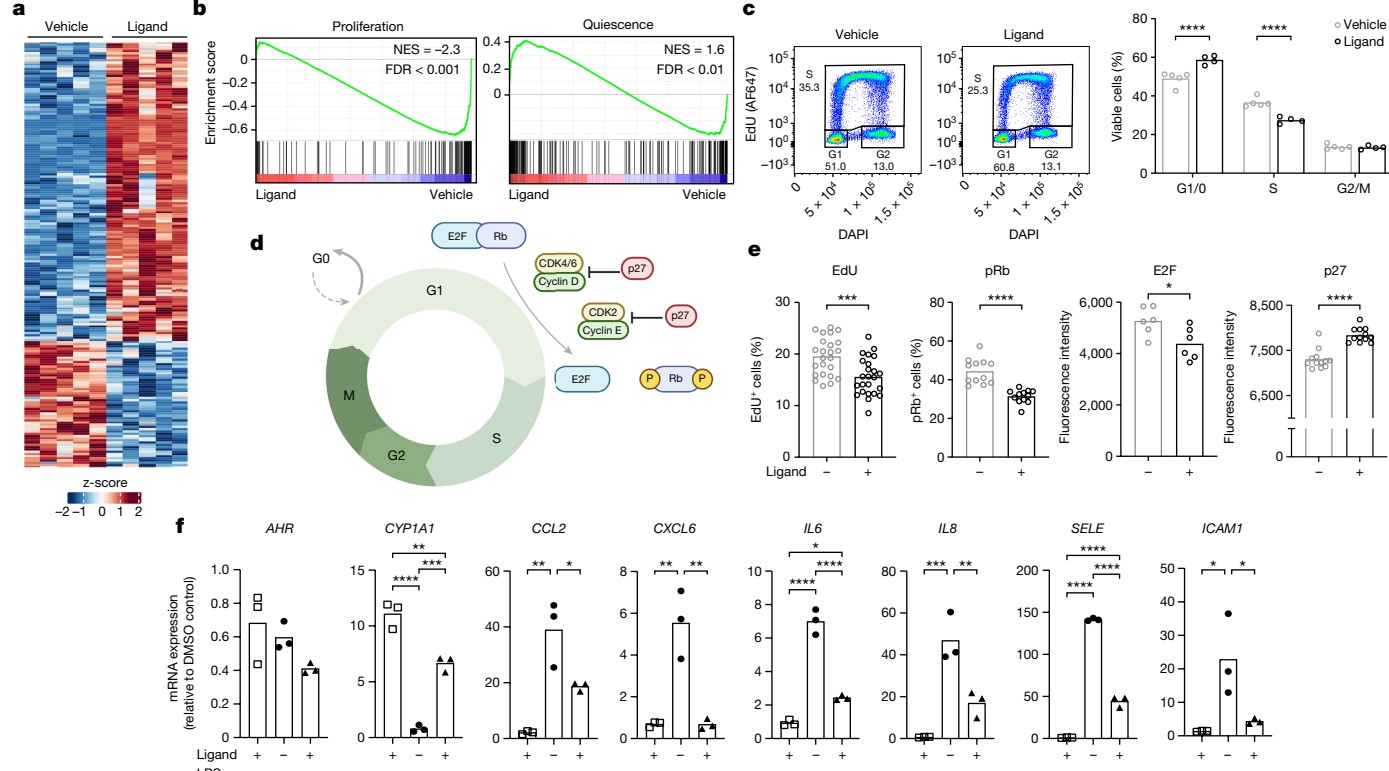

**Fig. 4 | AHR facilitates vasculoprotective pathways in human endothelial cells. a**, Heat map showing unsupervised hierarchical clustering of DEGs following bulk RNA-seq of FICZ-treated HUVECs compared with HUVECs treated with vehicle (*n* = 5 per group). **b**, Enrichment plots showing GSEA using HUVEC-specific proliferation and quiescence gene sets. Tested gene sets were from ref. 50. **c**, Flow cytometric cell cycle analysis of HUVECs treated with 100 nM FICZ (*n* = 4 replicates) compared with DMSO vehicle control (*n* = 5 replicates). **d**, Schematic of cell cycle regulators (created with BioRender.com). **e**, Single-cell imaging of cell cycle regulators in HUVECs treated with 100 nM

FICZ (+) or vehicle (−). Symbols represent mean expression from individual wells (*n* = 6–24). **f**, HUVECs were treated with 100 nM FICZ (for 2 h), followed by LPS stimulation (for 4 h) and profiled for expression of AHR pathway genes and endothelial inflammatory markers by quantitative PCR (qPCR). Data are representative of 3 independent experiments. Bar height shows mean throughout. *P* values calculated by two-way ANOVA with Šidák's multiple comparisons tests (**c**), unpaired *t*-tests (**e**) and one-way ANOVA with Tukey's multiple comparison tests (**f**).

demonstrate that the AHR provides a direct functional link between nutritional signals and the active maintenance of endothelial quiescence in mice and humans.

We uncovered the full cellular complexity of the mouse small intestinal endothelial compartment at single-cell resolution, building on work by Kalucka et al.[5]. We identified features including four distinct LEC subtypes (including a post-capillary vein-like LEC 2b), emergence of a novel angiogenic capillary population, and a shear stress-related artery cluster. Moreover, we identified many endothelial cell subtype-specific transcriptional regulons—programmes that shape cellular identity and will advance understanding of enteric endothelial cell heterogeneity. Among these findings, we emphasize the role of vitamin A metabolites in gut capillary transcriptional regulation and function, through predominant *Rara* and *Rarg* activity in capillary arterial and capillary 2 populations, respectively. Future work will investigate how these different endothelial cell subtypes communicate, cooperate and respond to homeostatic perturbations.

Despite the substantial heterogeneity in small intestine endothelial cell populations, we found that responsiveness to AHR ligand stimulation was a universal feature of endothelial cells, highlighting the role of this environmental sensor as a key facet of gut endothelial cell biology. AHR ligand sensing was crucial in the promotion of several aspects of functional endothelial cell quiescence—an active process that requires coordinated suppression of proliferative, migratory and inflammatory programmes that are necessary for the maintenance of vascular normalcy and organ homeostasis[1].

First, AHR activation suppressed endothelial proliferation. This is highly consistent throughout our transcriptomic (Figs. 2c,f, 3a–c and 4a,b and Extended Data Figs. 4a,f and 5d,e) and functional data (Figs. 3d–f and 4c–e and Extended Data Fig. 9) in mice and human cells and is supported by previous in vitro studies[16,41]. Although the majority of endothelial cells in vivo are maintained in quiescence, some endothelial cells undergo homeostatic proliferation in non-pathogenic angiogenesis[42]. We show here that AHR signalling limits this proliferation by reducing VEGFA sensitivity, an important checkpoint in preventing aberrant angiogenesis. This is particularly notable given the reliance of the intestinal vasculature and other fenestrated beds on continuous low-level VEGFA signalling for endothelial maintenance[1,43]. However, given that we did not observe any differences in intestinal vascular morphology or leakage in EC$^{\Delta Ahr}$ mice, we propose that AHR fine tunes the VEGFA response, acting primarily to restrain excess proliferation, without leading to a loss of enteric endothelial identity.

Second, AHR ligand sensing in endothelial cells acts as a potent anti-inflammatory signal. Exposure to AHR ligands dampened inflammatory activation following LPS challenge in vivo and in vitro in our study, through the downmodulation of adhesion molecules, cytokines and chemokines. AHR is known to negatively regulate type I IFN and NF-κB signalling pathways[44,45], and global *Ahr* deficiency is associated with heightened susceptibility to endotoxaemia[46]. Here, endothelial AHR-mediated dietary ligand sensing was required for optimal responses to enteric infection. There was a significant effect on survival

in female EC$^{\Delta Ahr}$ mice, and the small intestines of EC$^{\Delta Ahr}$ mice contained more eosinophils, dendritic cells, natural killer cells and γδ T cells. Together, these data suggest key perturbations in the composition of the inflammatory milieu, possibly through dysregulated immune recruitment at early stages of infection. Sensing of dietary AHR ligands may therefore provide a way to restrain endothelial activation and avert sustained inflammatory responses to enteric pathogens, analogous to the role of AHR in promoting disease tolerance following lung infection described in the accompanying Article[47]. This role of AHR in gut endothelial cells adds a new dimension to the holistic dependence of the intestine on AHR for optimum enteric immunity, adding an endothelial component to the described roles of epithelial cells and immune populations[10,11,48].

Finally, our data also show that endothelial cell responsiveness to nutritional AHR ligands may extend to other organs, potentially affecting endothelial cell quiescence and function systemically. The intestinal lumen is a rich source of AHR ligands, which not only act locally within the intestinal compartment but can also reach distal organs via the vasculature[49]. Sensing of gut-derived AHR ligands within the intestine can be described as 'outside–in' with epithelial cells as frontline responders followed by immune and structural cells, including endothelial cells. However, in organs other than the intestine, endothelial cells represent the main portal of entry for gut-derived AHR ligands into the tissue parenchyma, representing an 'inside–out' route of ligand exposure. How these two scenarios differ in terms of relative contribution of AHR-responsive cell types to organ homeostasis and integration of gut-derived environmental cues requires further study. The accompanying Article dissects the function of AHR in lung endothelial cells and reveals a role of 'inside–out' ligand exposure along the gut–lung axis for protection from virus-induced lung damage[47].

In summary, our study sheds light on intestinal adaptations to environmental cues, demonstrating that endothelial AHR ligand sensing acts as a crucial node for the maintenance of vascular normalcy across endothelial cell subtypes. With endothelial dysfunction increasingly recognized as a hallmark of chronic inflammatory disease, our data point towards a potential role for AHR activation through dietary ligand supplementation as a therapeutic strategy to facilitate organ homeostasis and disease resilience.

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

## Methods

### Mouse models

$Cdh5(PAC)^{creERT2}Ahr^{fl/f}$, $Cdh5(PAC)^{creERT2}Ahr^{fl/fNuTRAP}$, $Cyp1a1^{Cre}R26^{LSL-eYFP}$, and $Ahr^{-/-}$ mice were bred and maintained at the Imperial College London Central Biological Services facility. Wild-type mice used in scRNA-seq experiments were purchased from Charles River, UK. All mice were on a C57BL/6 background. Mice were bred and maintained in individually ventilated cages under specific-pathogen free conditions according to UK Home Office and local ethics committee (AWERB) approval. Cage and age-matched littermates served as experimental controls. Mice were housed in individually ventilated cages, at ambient temperatures (19–21 °C), and subjected to a standard 12:12 h light:dark cycle. Mice were between 6 weeks and 16 weeks at time of experiments, and male mice used throughout, except infection studies where a mixture of male and female mice were used. scRNA-seq: 3 mice per group; bulk RNA-seq: 2–3 mice–group used to obtain required cell frequencies.

### In vivo interventions

**Tamoxifen.** Mice were dosed with tamoxifen intraperitoneally (100 mg kg$^{-1}$ in corn oil) either once, or every other day for 14 days at 5–8 weeks of age[51].

**AHR ligand administration.** FICZ (Sigma, UK) was prepared to 20 mg ml$^{-1}$ in DMSO and diluted to 2 mg ml$^{-1}$ working stock in corn oil in glass containers. 3-MC (Sigma, UK) was prepared in corn oil to 5.3 mg ml$^{-1}$. FICZ and 3-MC were then injected intraperitoneally at 5 µl g$^{-1}$ to 10 mg kg$^{-1}$ and 26.5 mg kg$^{-1}$, respectively. For dietary interventions, mice were fed either a purified diet, or a purified diet supplemented with 1,000 mg kg$^{-1}$ I3C (Ssniff) ad libitum for 8 days.

**LPS treatment.** LPS (Sigma, UK) was given intraperitoneally at 10 mg kg$^{-1}$ in sterile PBS for 24h.

**EdU feeding.** EdU (ThermoFisher) was first dissolved in DMSO to 50 mg ml$^{-1}$ and then further diluted to working dilution to be given to mice at 30 mg kg$^{-1}$ intraperitoneally (30 mg kg$^{-1}$) on day 0. EdU was added to the drinking water at 0.3 mg ml$^{-1}$ of drinking water as previously described[52]. Edu-water was replaced fresh every 2–3 days.

**VEGFA administration.** Mouse VEGFA$_{165}$ (Peprotech) stock was made in 100 µg ml$^{-1}$ in sterile PBS and administered intraperitoneally at 5 µg per mouse in sterile PBS twice per week over 2-week EdU feeding period (4× doses).

**VEGFR2 blockade.** VEGFR2-blocking antibody (DC101) or control IgG (both BioXCell, USA) were injected bi-weekly intraperitoneally at 40 mg kg$^{-1}$ concomitant with 2-week EdU feeding (4× doses).

**Administration of 100-nm microspheres.** The 100-nm fluorescent microspheres (580/605 FluoSpheres, ThermoFisher) were vortexed thoroughly, diluted 1/5 in sterile PBS and administered intravenously at 100 µl per mouse. Small intestine tissues were taken for whole-mount imaging analysis after 5 min as previously described[34].

**_Y. pseudotuberculosis_ infection.** _Y. pseudotuberculosis_ (32777) was grown (27 °C, 300 rpm) overnight in 2× YT medium supplemented with 2 µg ml$^{-1}$ Irgasan (Sigma). Mice were infected with bacterial solution washed twice and resuspended in PBS (200 µl per mouse; $5 \times 10^7$ or $5 \times 10^8$ CFU as described).

### Tissue digestion

Following collection and fat removal, small intestine and colon were cut open longitudinally, and underwent an IEL wash: incubated with IEL wash buffer (IMDM + 1% FCS, 5 mM EDTA, 10 mM HEPES,

penicillin/streptomycin, and 2 mM DTT) for 20 min at 37 °C with 200 rpm with shaking. Small intestine was washed and vortexed at low speeds in small intestine PBS (PBS + 5 mM EDTA + 10 mM HEPES) 30 s, 3 times or until clear, and then vortexed at low speed in PBS. Colon was vortexed in PBS once. Both gut tissues were then cut into small pieces and incubated in digestion buffer. All other tissues (brown adipose tissue, inguinal white adipose tissue, liver, lung and kidney spleen) were cut into small pieces and incubated directly in digestion buffer.

For endothelial cell-tailored digests, all organs were digested by incubation in collagenase A digestion buffer (4 ml per tissue: HBSS + 20 mM HEPES, 10 mg ml$^{-1}$ Collagenase A (Sigma), 8 U ml$^{-1}$ Dispase II (Sigma), 50 µg ml$^{-1}$ DNase I (Sigma)) for 20 min at 37°C with 200 rpm shaking. For immune cell digests, organs were incubated in Collagenase VIII digestion buffer (5 ml per tissue: IMDM + 1% FBS, 10 nM HEPES, penicillin/streptomycin, 1 mg ml$^{-1}$ Collageanse VIII (Sigma), 50 µg ml$^{-1}$ DNase I). All reactions stopped through addition of 1:1 complete medium (IMDM + 1% FCS, penicillin/streptomycin, 1× Glutamax) and: passing through 100-µm filters (small intestine, colon, kidney, lung and spleen); debris removal through 2× 1-min centrifugation at 60$g$, collecting supernatant and then passing through 100-µm filters (liver); or 250$g$ 10-min centrifugation and careful floating adipocyte fraction removal (brown adipose tissue and inguinal white adipose tissue). After centrifugation (400$g$, 8 min), small intestine and colon were subjected to 40% Percoll (Amersham) density gradient centrifugation (400$g$, 8 min) to remove debris, while all other tissues underwent resuspension in 0.5 ml ACK lysis buffer for 2 min and washing to remove erythrocytes. Finally, cells were filtered (40 µm) and counted before downstream analysis.

### Primary cell culture

Primary HUVECs (Lonza, UK) obtained from pooled donors were seeded at 60–80% confluency and cultured in Endothelial Cell Growth Medium-2 medium (EGM-2 Bulletkit; Lonza) supplemented with penicillin/streptomycin at 5% CO$_2$ and 37 °C. Cells were switched to minimal growth medium (EGM-2 Bullet kit basal medium, Lonza; supplemented with penicillin/streptomycin) for 24 h prior to experimentation. For FICZ stimulation experiments, FICZ (between 0.1 nM and 100 nM), or DMSO control were added to the cells for indicated time points. For LPS/FICZ dual stimulation experiments, cells were treated with (a) DMSO only (1:1,000, Fisher Scientific, UK), (b) LPS only (50 ng ml$^{-1}$ final concentration, InvivoGen), (c) FICZ only (100 nM final concentration, Enzo) or (d) FICZ + LPS for 6 h in total. Fresh DMSO, LPS or FICZ were added to (a), (b) and (c) after 2 h. The FICZ + LPS group was treated with FICZ for 2 h initially followed by stimulation with FICZ and LPS for 6h. HUVECs between passage 1 and 4 were used in experiments.

### Transfection

HUVECs were grown to 60–80% confluency, before addition of 1 µg of ON-TARGETplus Human AHR (196) siRNA–SMART pool (SiScr) (Horizon, L-004990-00-0050) or 1 µg ON-TARGETplus Non-Targeting Pool (SiScr) (Horizon, D-001810-10-50) for 24 h. Medium was changed 1 h prior to transfection.

### Quantitative real-time PCR

RNA was purified from frozen HUVEC using TRIzol Plus RNA Purification Kit (ThermoFisher), or using RNeasy Plus micro kit (Qiagen) for FICZ/LPS-treated HUVEC experiments. cDNA was generated using High Capacity cDNA reverse transcription kit (ThermoFisher). qPCR was performed with Sso advanced universal SYBR Green Supermix (Biorad) run through Biorad CFX Maestro v1.1 using the following primers. *AHR*: forward 5′-GCCCTTCCCGCAAGATGTTAT-3′; reverse 5′-CAAAGCCATTCAGAGCCTGT-3′; *CYP1A1*: forward 5′-CAATGAGTTTGGGGAGGTTACTG-3′; reverse 5′-CAATTCGGATCTGCAGCACG-3′; *CCL2:* forward 5′-AGACTAACCCAGAAACATCC-3′; reverse 5′-ATTGATTGCATCTGGCTG-3′; *CXCL6*: forward 5′-CCTCTCTTGACCACTATGAG-3′; reverse 5′-GTTTGGGGTTTACTCTCAG 3′; *IL6*:

forward 5′-GCAGAAAAAGGCAAAGAAT 3′; reverse 5′-CTACATT TGCCGAAGAGC-3′; *IL8:* forward 5′-GTTTTTGAAGAGGGCTGAG-3′; reverse 5′-TTTGCTTGAAGTTTCACTGG-3′; *SELE*: forward 5′-GA GAATTCACCTACAAGTCC-3′; reverse 5′-AGGCTTGAACATTTTACCAC-3′; *ICAM1*: forward 5′-ACCATCTACAGCTTTCCG-3′; reverse 5′-TCA CACTTCACTGTCACC-3′.

### Flow cytometry and FACS
For mouse experiments, cell suspensions were incubated with anti-mouse CD16/CD32 (TrustainFx, Biolegend), before incubation with LIVE/DEAD Dye (Zombie Near Infra-red, Biolegend), and incubation with surface antibodies (Supplementary Table 12). All antibodies and subsequent washes in FACS buffer (PBS + 2% FCS, 2 mM EDTA). For EdU detection, Click-iT EdU proliferation kit Pacific Blue (ThermoFisher) was used, according to manufacturer's instructions. DAPI nuclear stain added directly less than 5 min prior to analysis and sorting. A combination of compensation beads (OneComp/Ultracomp eBeads, ThermoFisher) for antibody controls, and single-stained cells for dyes and fluorescent proteins were used for controls and for compensation/spectral unmixing.

For HUVEC proliferation experiments, Click-iT EdU proliferation kit AlexaFluor647 was used according to manufacturer's instructions. Briefly, this involved a 1-h EdU pulse to cells prior to washing, fixing, DAPI staining and analysis.

Samples were run on a LSR II flow cytometer running FACSDiva v9 software (BD), a Cytek Aurora spectral cytometer running Spectoflo v3 software (Cytek); or sorted on an Aria Fusion with FACSDiva v9 (BD). Data analysed offline using FlowJo v10.6 (BD). Multi-immune cell gating based on refs. 53,54.

For scRNA-seq experiments, total live endothelial cells were FACS sorted (80-μm nozzle size, <5.0 flow rate) into collection buffer (PBS + 10% FCS), counted manually, and taken forward to single-cell barcoding in PBS + 0.04% BSA. For mouse RNA-seq experiments cells were FACS sorted (settings as above) into RLT buffer (Qiagen) containing 1% β-mercaptoethanol (Sigma) and placed immediately onto dry ice before RNA isolation.

### Single-cell RNA sequencing
Sorted mouse CD31⁺CD45⁻ single-cell suspensions (viability >80%) were obtained from three samples across two conditions (1 vehicle-treated, 2 FICZ-treated) aiming for 15,000 cells per library and a sequencing depth of 50,000 reads per cell. Both FICZ-treated samples were from the same pool of cells, used in the experiment to match cell numbers with the vehicle sample. Cells were partitioned using the 10X chromium controller and the resulting GEMs (gel bead in emulsion) were converted into scRNA-seq libraries using the 10x Single Cell 3′ v3 kit according to manufacturer specifications. Libraries were sequenced on an Illumina 2500 generating Paired-End 100bp reads which were processed using the 10x Genomic Cell Ranger pipeline v4.0.0.

**Data preprocessing and QC.** scRNA-seq from 3 samples (Ficz_15k, Veh_15k, Ficz_60k) was generated by Hiseq2500. Demultiplexing was performed with CellRanger (v.4.0.0) mkfastq using bcl2fastq v2.17.1.14 based on the 10x library indices (allowing 0 mismatches). CellRanger count pipeline was used to perform alignment against mouse genome mm10 (using STAR), filtering, barcode counting and unique molecular identifiers (UMIs) counting. UMI count matrices were then imported to Seurat (v3.2.0)[55] with the following arguments: min.cells = 10 and min.features = 100. We further filtered cells based on the commonly used QC metrics suggested by ref. 56 with the following thresholds: percentage of mitochondrial counts (per.mt) <6%; total number of molecules detected within a cell (nCount_RNA) between 500 and 8,000 for Ficz_60K and nCount_RNA between 500 and 15,000 for Ficz_15k and Veh_15k. After data preprocessing, SCTransform normalization was performed[57]. Linear dimensional reduction was performed using Seurat

RunPCA() with argument npcs = 30, RunUMAP(), FindNeighbors() and FindClusters() with argument resolution = 0.5. Cells subsequently identified as doublets by DoubletFinder[58] with parameters pN = 0.25, pk = 0.16 for Veh_15k, 0.22 for Ficz_15k and Ficz_60k were removed and the remaining cells were processed with Seurat SCTransform again and samples from Ficz and Veh were integrated using PrepSCTIntegration(), FindIntegrationAnchors() and IntegrateData() functions and the linear dimensional reduction with Seurat as described above. Both FICZ samples were integrated to form a single sample before integration with the vehicle sample for condition-wise comparisons. Cells in cluster 8, 12, 13, 14 were identified as contaminant clusters and were removed after visualization with FeaturePlot() and VlnPlot() functions.

**Clustering analysis and conserved markers.** The "clean" dataset was processed with the Seurat pipeline mentioned above and in the FindClusters(), using argument resolution = 0.4. Conserved makers were further defined using the FindConservedMarkers() function from Seurat.

**DEG and pathway analysis.** DEG analysis for the scRNA-seq was performed by using FindMarkers() function from Seurat to define genes that are differentially expressed between stimulated and control clusters. Gene set analysis on these separate lists of up-, and downregulated DEGs was performed using EnrichR[59].

**Sub-clustering and overdispersion analysis with Pagoda2.** Based on the "clean" dataset, 11 clusters were identified by under resolution = 0.4. Cluster 2, cluster 3, cluster 5, cluster 6, cluster 8, cluster 9, and cluster 10 were classified as BECs and cluster 0, cluster 1, cluster 4 and cluster 7 were classified as LECs. Sub-clustering analysis was performed on BECs and LECs respectively using the SCTransform() for data normalization and PrepSCTIntegration(), FindIntegrationAnchors() and IntegrateData() functions for data integration. The linear dimensional reduction with Seurat was performed as described above but with resolution = 0.2 for both BECs and LECs.

Functional analysis for the Vehicle dataset for BECs and LECs was performed by pagoda2 (version 1.0.8[18]), using curated endothelial-related pathways (built on from ref. 5) (see Supplementary Table 3) and gene sets obtained via the msigdbr() function from the R Bioconductor package msigdbr (v. 7.4.1) for MSigDB Collections C2, C5 and C7. Raw counts from BECs and LECs were imported to pagoda2 (ref. 18) using basicP2proc() function for data processing with arguments "n.cores = 4, min.cells.per.gene = 10, n.odgenes = 2e3, get.largevis=FALSE, make. geneknn=FALSE", then followed by makeKnnGraph(), getKnnClusters(), and getEmbedding(), getKnnClusters(), getDifferentialGenes() functions. Pathway overdispersion was estimated via pagoda2 with argument "correlation.distance.threshold = 0.8".

**Transcription factor regulon analysis with SCENIC.** SCENIC version 1.2.4 (ref. 60) for regulatory network analysis was performed for the scRNA-seq vehicle dataset. The workflow started from identifying potential gene regulatory network (GRN) by using runGenie3() and runSCENIC_1_coexNetwork2modules() functions then followed by selecting potential regulons with runSCENIC_2_createRegulons() function after applying mm10_refseq-r80_500bp_up_and_100bp_down_ tss.mc9nr.feather motif dataset retrieved from cisTarget databases (https://resources.aertslab.org/cistarget/). The final step was to score the regulons in the cells using runSCENIC_3_scoreCells() function. Regulon specificity scores for each cell clusters were calculated using calcRSS () function[61]. The final list of 167 regulons excluded non '_extended' duplicates when '_extended' versions were present.

**scRNA-seq data visualization.** Dot plots were created by R package ggplot2 (v.3.3.3). Heatmaps were created by using R Bioconductor package ComplexHeatmap v2.2.0 (ref. 62). Featureplots and violin plots

were created by using function FeaturePlot() and VlnPlot() functions from the R Seurat package.

## Bulk RNA-seq analysis

RNA from FACS-sorted BEC and LEC from EC^WT and EC^ΔAhr mice ($n = 3$ per group) was extracted with the RNeasy plus micro kit (Qiagen). RNA-seq libraries were prepared with the NEBNext Single Cell/Low Input RNA Library Prep Kit for Illumina (NEB) from 1 ng total RNA. Sequencing was then carried out with NextSeq500 using paired-end 40-bp reads. Illumina RTA (version 2.11.3) software and bcl2fastq (2_2.20.0) were used for basecalling and demultiplexing (allowing 0 mismatches). Raw RNA-seq reads were aligned against mm10 and transcript annotations using STAR v.2.2.7a[63]. Data normalization was performed using the DESeq2 Bioconductor package[64] and was rlog transformed to allow for visualization by PCA and heatmaps. Unwanted batch effects were controlled by using R Bioconductor package RUVseq[65] with RUVg function and $k = 2$. A ranked DEG list was generated based on Wald statistics from DESeq2 results, and GSEA was performed using hallmark gene sets[66,67]. EnrichR was applied on separate positive and negatively enriched lists.

For mouse whole-tissue RNA-seq, pieces of small intestine were taken, washed briefly in PBS and snap frozen in liquid nitrogen. Tissue pieces were prepared using the TRIzol plus RNA purification kit (ThermoFisher) according to manufacturer's instructions, and RNA-seq libraries prepared using NextSeq2000 using paired-end 36bp reads to achieve ~120M reads per sample. Reads were aligned against mouse genome GRCm38 using STAR 2.7.7a. Data normalization was performed using the DESeq2 Bioconductor package (v.1.30.1)[64] and was rlog transformed to allow for visualization by PCA and heatmaps. A ranked DEG list was generated based on Wald statistics from DESeq2 results.

For human data, HUVECs were treated with 100 nM FICZ or DMSO control, for 6 h, processed to RNA using the TRIzol Plus RNA Purification Kit as described above, and RNA-seq libraries prepared using NextSeq500 using paired-end 40-bp reads to achieve ~40M reads per sample. Reads were aligned to Ensembl human genome (GRCh37) using tophat2 version 2.0.11 (ref. 68). Mapped reads that fell on genes were counted using featureCounts from Rsubread package[69]. Generated count data were then used to normalize and identify DEGs using DESeq2 and DEGs were defined with Benjamini-Hochberg adjusted $P < 0.05$. Gene Set Enrichment Analysis was performed using GSEA on pre-ranked lists generated by the DESeq2 package.

## Whole-mount gut imaging

Whole-mount gut imaging protocol adapted from Bernier-Latmani et al.[70]. Briefly, small intestines were harvested, the mesenteric fat removed, small intestines divided into duodenum, jejunum and ileum pieces, opened longitudinally and gut contents carefully cleaned in ice-cold PBS. Tissue underwent 20 min intraepithelial lymphocyte wash at 37 °C, 200 rpm as described above. Supernatant was poured through strainer and washed in small intestine PBS thrice (to facilitate epithelial cell removal), then PBS once (as above) before tissue pieces pinned at 0.5-cm intervals to silicone plates containing ice-cold PBS. Tissue pieces gently brushed to further remove villus epithelial cells, PBS replaced with 4% PFA and incubated at 4 °C, gentle rocking for 2 h. After 2×10-min washes with PBS, 3h incubation with 10% sucrose solution, then 16 h with 20% sucrose +10% glycerol solution (all gentle rocking, 4 °C); tissues were rinsed once more with cold PBS and cut into approximately 1cm pieces. Blocking buffer (PBS + 5% donkey serum, 0.5% BSA, 0.3% Triton X, 0.1% NaN₃) was added for 3 h (rocking 4 °C) before primary antibodies added in blocking buffer for 5 days. After 4×1 h washes in wash buffer (PBS + 0.3% Triton X), secondary antibodies added in wash buffer for 12–16 h (see Supplementary Table 12 for primary and secondary antibodies used). Tissues washed in wash buffer (10×30 min, rocking 4 °C) and dissected under dissection microscope into 1–2 villi-wide strips. Strips placed into C3eD clearing solution (described in ref. 71) for 30 min before mounting onto slides fitted with spacers using prolong

diamond mounting medium (ThermoFisher). Images acquired on a Leica SP5 II confocal microscope running LAS-AF v2.7.3.9723 software (Leica), or a Zeiss LSM 780 confocal microscope running ZEN Black v14.0.27.201 (both Zeiss) using 10× and 20× objectives.

Whole-mount image quantification performed with FIJI (v.2.9.0). For ESM1 analysis, ESM1⁺ cells counted in each villi and normalized to CD31⁺ vascular villi area. VEGFR2 density/villi and number of branch-points/villi were calculated using the 'Analyze skeleton' function in FIJI. Relative lacteal length was calculated by lacteal length/villi length (as described in ref. 70). Lacteal filipodia/villi were manually identified[70]. Leakage was quantified by counting areas of bead leakage in proximity to villi vessels, crypt vessels, or veins, and normalizing to number of villi or veins per image respectively.

## Quantitative single-cell imaging

HUVECs were seeded onto 384 well CellCarrier Ultra plates (Perkin Elmer), washed with Hanks Balanced Salt Solution (ThermoFisher) and changed to EBM-2 medium (Lonza) + 2% FBS (Sigma) + penicillin/streptomycin (ThermoFisher) for 24h. Cells were treated with 100 nM FICZ or DMSO control for 6 h, and then fixed in 4% formaldehyde (Sigma) in PBS. Cells were permeabilized in PBS/0.5% Triton X-200 for 20 min, blocked in 1% BSA/PBS for 30 min and incubated with primary antibodies, diluted in blocking buffer, overnight at 4 °C (phospho-Rb clone D20B12, Rb clone 4H1, p27 kip1 clone D37H1 (Cell Signalling Technologies); E2F1 clone EPR3818(3) - Abcam). Cells were washed three times in PBS followed by incubation with Alexa fluorophore labelled secondary antibodies, diluted 1:1,000 in blocking buffer, for 1 h at room temperature in the dark (goat anti-mouse IgG AlexaFluor488, Goat anti-rabbit IgG AlexaFluor568; both Invitrogen). Cells were washed three times in PBS and finally nuclei were labelled with 1 μg ml⁻¹ Hoechst 33258 diluted in PBS for 15 min at room temperature, in the dark before a final 3 washes in PBS and storing the cells in PBS. Cells were imaged with a 10× NA 0.4 objective on an Operetta CLS high-content microscope (PerkinElmer). Quantitative analysis of fixed cells was performed using Harmony v4.9 software (PerkinElmer). Nuclei were segmented based on Hoechst intensity. Nuclei at the edge of the image were excluded. Fluorescence intensity of individual proteins and EdU was calculated within each nucleus. A threshold for EdU- and phospho-Rb-positive cells was calculated as in ref. 72.

## Statistical analysis and reproducibility

Statistical analyses were performed using GraphPad Prism v9 software. Statistical tests were used as follows: unpaired Student's $t$-test and paired $t$-tests to compare two groups; one-way ANOVA with Tukey's multiple comparisons tests, and two-way ANOVA with Šidák's multiple comparisons tests to compare multiple groups; and Gehan–Breslow–Wilcoxon tests to compare survival curves. All statistical comparisons were two-sided. Within scRNA-seq data, conserved marker significance (between clusters), and differential marker significance (between conditions for each cluster) were assessed with Wilcoxon rank sum tests as part of the Seurat v3 package. Within scRNA-seq and RNA-seq data, gene set overrepresentation analyses were assessed Fisher's exact tests as part of the EnrichR tool. We consider a $P$ value of 0.05 significant. Significance levels of *$P < 0.05$, **$P < 0.01$, ***$P < 0.001$, ****$P < 0.0001$ used throughout.

## Reporting summary

Further information on research design is available in the Nature Portfolio Reporting Summary linked to this article.

## Data availability

All sequencing data (mouse scRNA-seq, mouse RNA-seq and human RNA-seq datasets) have been deposited in the NCBI Gene Expression Omnibus as a superseries under accession number GSE201789.

Mouse genome mm10 sequences were retrieved from GENCODE mouse genome (GRCm38), version M23 (Ensembl 98) https://www.gencode-genes.org/mouse/release_M23.html. In SCENIC analysis, RcisTarget was used (https://resources.aertslab.org/cistarget/) and database mm10_refseq-r80_500bp_up_and_100bp_down_tss.mc9nr.feather was downloaded for our analysis. scRNA-seq metrics/metadata, conserved and DEG lists from sequencing experiments, Regulons (generated by SCENIC), gene sets that comprise Pagoda2 outputs (Aspects) and input gene sets for Pagoda2 overdispersion analysis are all provided as supplementary tables. All other flow cytometry, images and qPCR data are presented within the manuscript. All raw data are provided as source data files accompanying the manuscript. Source data are provided with this paper.

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

**Acknowledgements** The authors thank staff at the Imperial College Central Biological Services for help with animal breeding, maintenance and colony management, in particular S. Satchell and A. Miller for their invaluable assistance; J. Elliot, B. Patel and R. Maggio for the support they provided in FACS; L. Game, I. Andrew and J. Haywood for assistance with sequencing; the LMS microscopy team and S. Rothery for their help with imaging and image analysis; I. Brodsky for providing *Yptb*; E. Bieberstedt for helpful discussions and troubleshooting with Pagoda2 analysis; and H. Gleneadie for their help with editing the manuscript. This work was made possible through funding from the Wellcome Trust (C.S. and B.G.W., 211080/Z/18/Z; G.F. and J.S.-G., 224282/Z/21/Z), Fondation Acteria, the Medical Research Council (LMS Core funding), the British Heart Foundation (A.M.R., BHF RG/17/4/32662; G.M.B., PG/20/16/35047) and Cancer Research UK (A.R.B. and B.R.P., C63833/A25729).

**Author contributions** B.G.W. and C.S. conceived the project. B.G.W., C.S., A.B., J.S.-G. and G.F. designed experiments. B.G.W., A.B. and N.G. generated data. B.G.W., Y.-F.W., A.B., N.G., M.D. and C.C. analysed data. Y.-F.W. performed extensive scRNA-seq analysis. M.D. and C.C. performed RNA-seq analysis. B.G.W., C.S., A.B., Y.-F.W., J.M.V.W., B.R.P., A.R.B., G.M.B. and A.M.R. critically evaluated the data. B.G.W. and C.S. wrote the manuscript; B.G.W., C.S., Y.-F.W., A.B., M.D., C.C. and N.G. constructed figures. J.S.-G., G.F. and S.V. provided critical help with mouse experiments. C.S. acquired funding and supervised the project. All authors reviewed the manuscript. Y.-F.W. and A.B. are joint second authors. B.G.W. is first and co-corresponding author.

**Competing interests** The authors declare no competing interests.

**Additional information**
**Correspondence and requests for materials** should be addressed to Benjamin G. Wiggins or Chris Schiering.

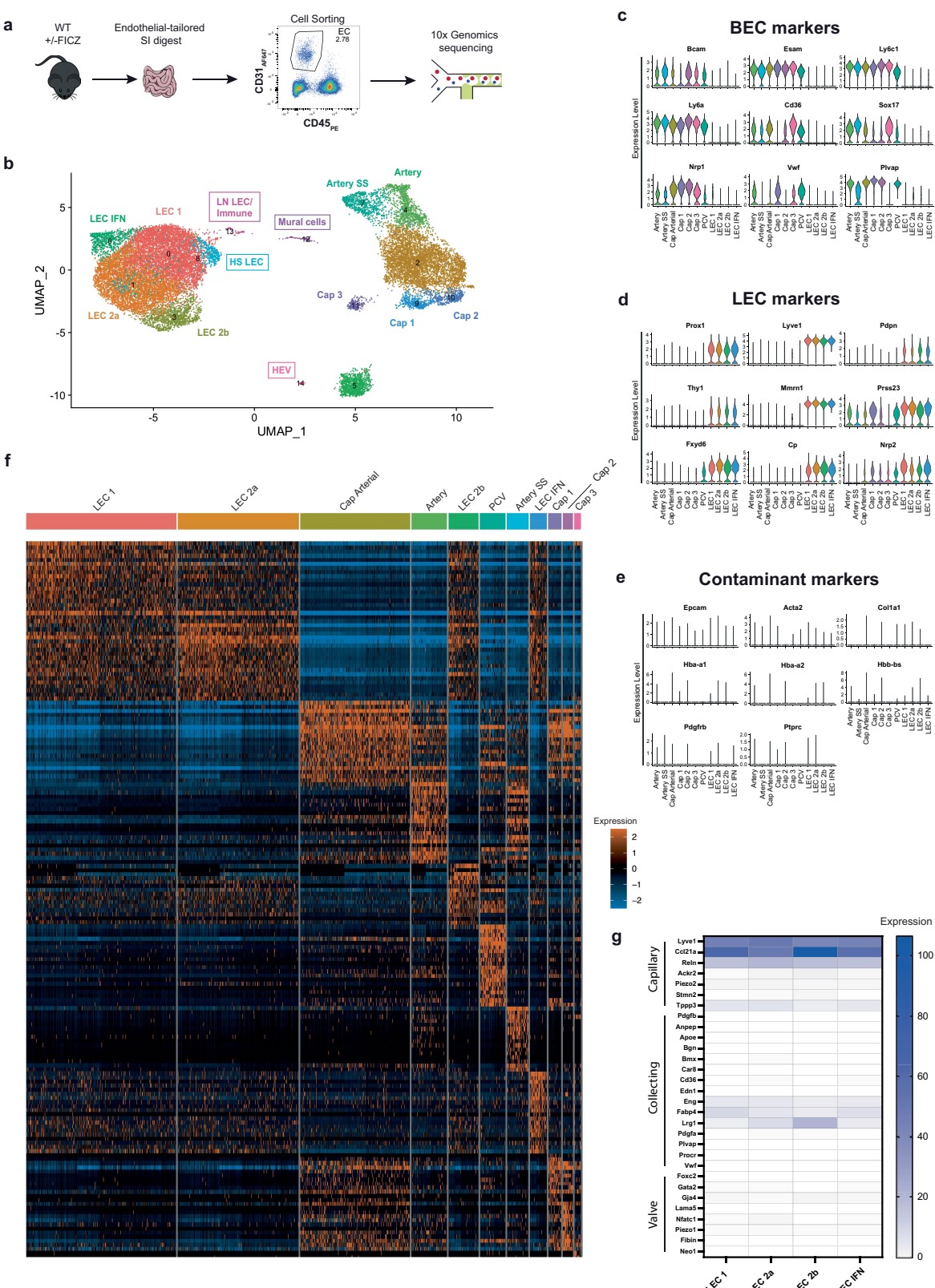

**Extended Data Fig. 1 | Design and cluster identification in the enteric endothelial single cell RNAseq dataset. a**, Experimental design. WT mice (n = 3/group) treated with FICZ or vehicle control for 3h, before small intestine digest, total live CD31+ cell sorting by FACS; single-cell barcoding, library generation and sequencing. **b**, UMAP showing data structure post QC and doublet removal. Boxes indicate the four contaminant populations that were removed based on cluster size (clusters 12–14), or dominant heat-shock signature (cluster 8). **c-e**, Violin plots for verification, showing expression of common BEC markers (**c**), common LEC markers (**d**), and potential contaminant markers (**e**) (epithelial cells, pericytes, mural cells, fibroblasts, erythrocytes, and immune cells) in combined dataset. **f**, Heatmap of average expression of lymphatic capillary, collecting duct and valve genes in vehicle dataset (based on reference in LEC clusters). **g**, Heatmap displaying relative expression of capillary, valve, and collecting duct LEC markers amongst our LEC clusters.

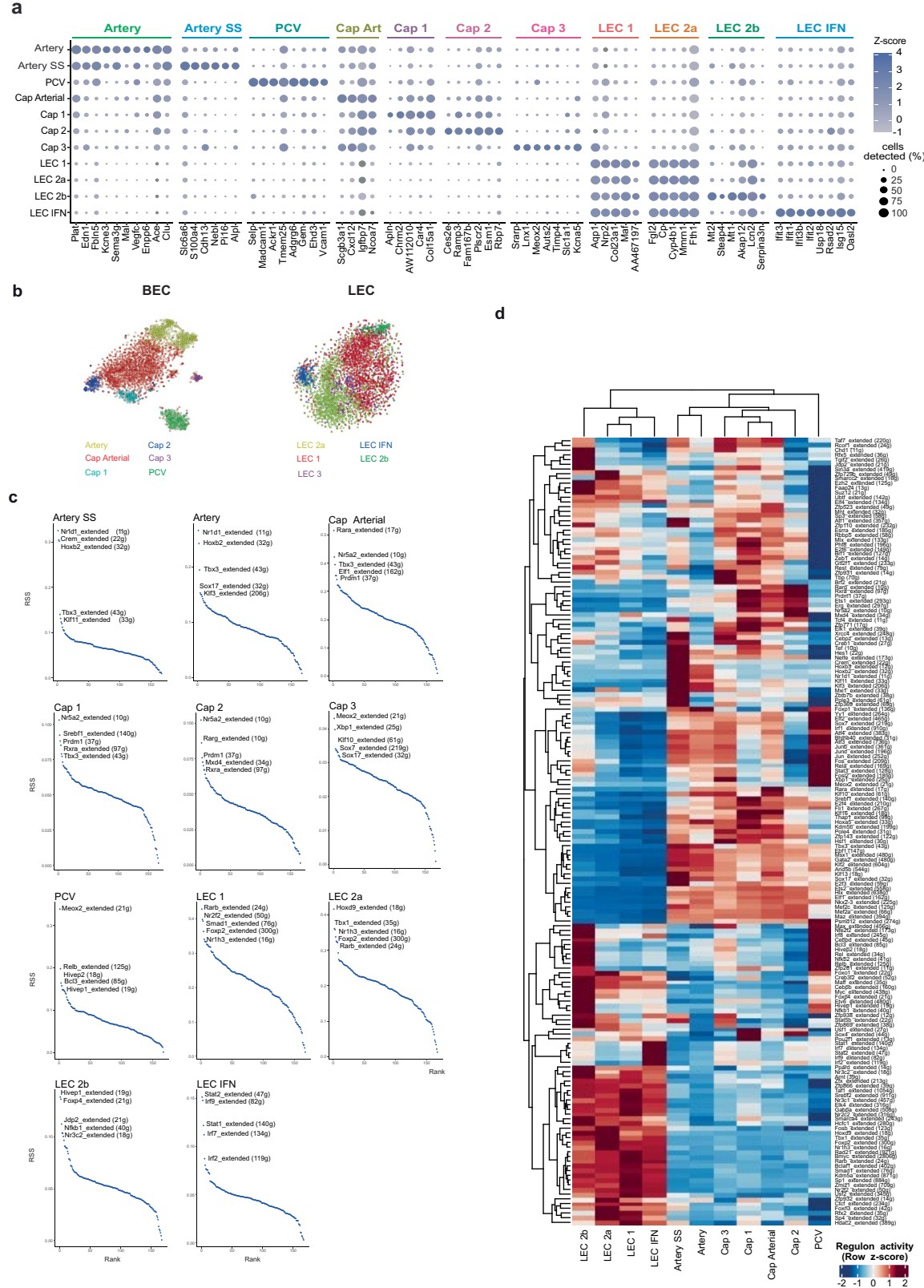

**Extended Data Fig. 2 | Characterising intestinal endothelial heterogeneity with single cell transcriptomics. a**, Dot plot showing selected top marker gene expression for each cluster scaled across all clusters (colour intensity), and percentage expression within that cluster (dot size). **b**, Annotated tSNE projection of BEC, and LEC subclustering used in Pagoda2 analysis pipeline (Fig. 1c). **c**, Regulon specificity score (RSS) for each cluster. Top 5 regulons for each type are annotated on the plots. **d**, Heatmap showing normalised regulon activity (AUC) score in each cluster for all 167 identified regulons.

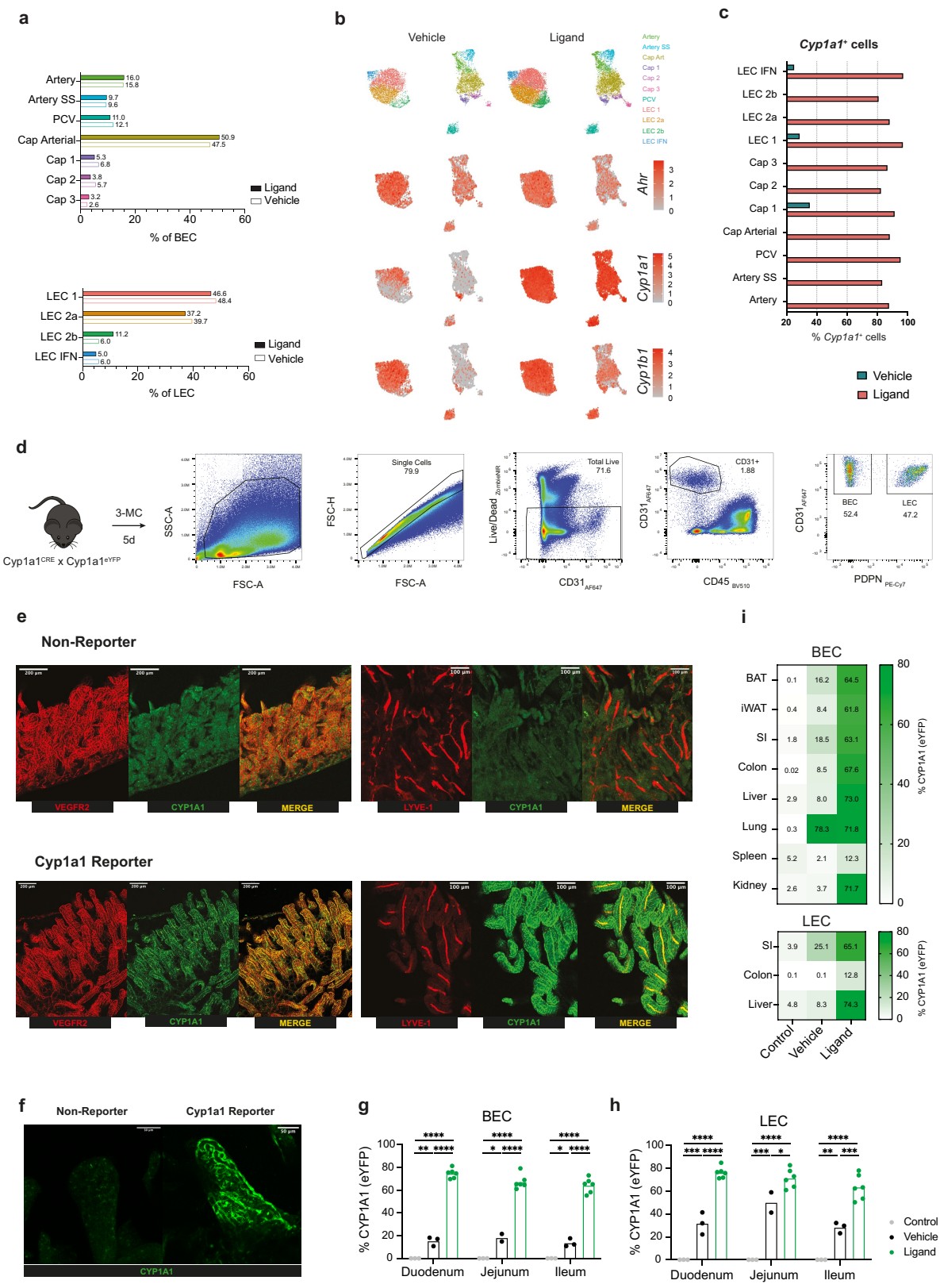

**Extended Data Fig. 3** | See next page for caption.

**Extended Data Fig. 3 | Interrogating EC AHR activation with flow cytometry, fluorescent imaging, and scRNAseq. a**, Percentage composition of each cluster in vehicle (open bar), and ligand-treated conditions (closed bar). **b**, *Ahr*, *Cyp1a1*, and *Cyp1b1* expression shown across the clusters in vehicle and ligand-treated mice. **c**, Percentages of *Cyp1a1*⁺ cells in ligand-treated compared to vehicle-treated mice. Bar lengths represent % of cells (over relative expression threshold of 1) expressing *Cyp1a1* in vehicle-treated (blue) and ligand-treated (red) data. **d**, Experimental design and gating strategy for *Cyp1a1*-reporter mice by flow cytometry (Fig. 2c). *Cyp1a1*-reporter mice treated with AHR ligand 3-MC (5d prior small intestinal tissues digest and analyses by flow cytometry. Gates sequentially demonstrate exclusion of debris, exclusion of multiplets, exclusion of dead cells, and selection of CD31⁺ endothelial cells. **e**, Whole mount gut staining of *Cyp1a1*-reporter mice, and WT non-reporter controls both treated with 3-MC for 5d. Panels show VEGFR2, Cyp1a1-eYFP and merged staining at 10x magnification. Representative images of 3–6 images/condition. **f**, 20x representative images of CYP1A1 staining in the small intestinal villi of WT non-reporter mice (left) compared to Cyp1a1-reporter mice (right). Representative of minimum 3 images. **g**, **h**, Percentage *Cyp1a1*-eYFP expression in BEC (**g**) and LEC (**h**) from different SI segments following treatment with 3-MC (n = 6 mice), vehicle (n = 2-3 mice), or from non-reporter control mice (n = 3 mice). Bar heights represent the mean, symbols show individual mice. **i**, Heatmaps showing % Cyp1a1-eYFP expression in 5d 3-MC/vehicle treated *Cyp1a1*-reporters (n = 4-5 per group except liver n = 6–8, kidney/lung/spleen n = 3, and colon LEC n = 2). SI - small intestine, BAT - brown adipose tissue, iWAT - inguinal white adipose tissue. *$P < 0.05$, **$P < 0.01$, ***$P < 0.001$, ****$P < 0.0001$ as calculated by one-way ANOVA with Tukey's multiple comparisons tests.

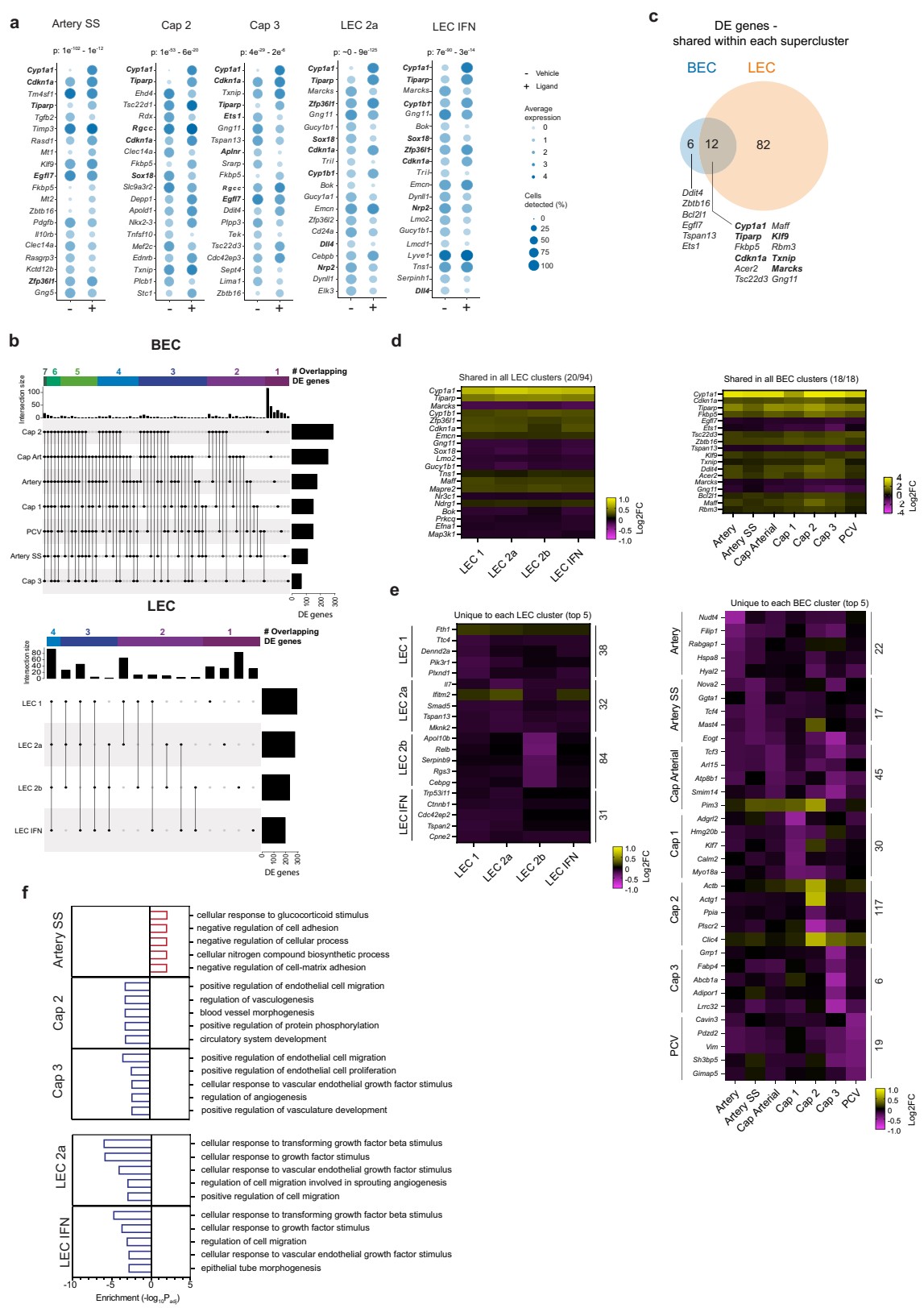

**Extended Data Fig. 4** | See next page for caption.

**Extended Data Fig. 4 | Further differential gene and pathway analysis on each cluster. a**, Dot plots showing top 20 DE genes for remaining 5 clusters (other 6 in Fig. 2e). + and – indicate ligand- and vehicle-treated conditions respectively. Colour intensity represents average expression level, dot size – average % expression of each gene. Genes relating to canonical AHR pathway, or proliferation in bold. **b**, UpSet plots showing DE gene overlaps between each cluster, within BEC (top) or LEC (bottom) supercluster. Black connecting lines in columns represent all possible overlaps, intersection size the number of shared genes in each overlap. Total DE genes for each cluster shown in right bars. Coloured bars at top of plot indicate number of overlaps within range. Only genes expressed in all BEC or LEC clusters were included in this analysis. **c**, Venn diagram showing overlap between DE genes shared in all BEC and DE genes shared across all LEC. **d,e**, Heatmaps to compare DE gene expression across clusters. Heatmaps show DE gene expression (as log2 fold-change compared to vehicle only) in either shared genes in all parent BEC/LEC clusters (**d**), or top 5 unique genes for each cluster (**e**). Numbers to right of unique gene heatmaps indicate total number of unique DE genes within each cluster. **f**, As in Fig. 2e, Top 5 enriched genesets (GO Biological Processes) in 5 remaining clusters (red – upregulated geneset, blue – downregulated geneset). P values calculated by Wilcoxon Rank Sum tests.

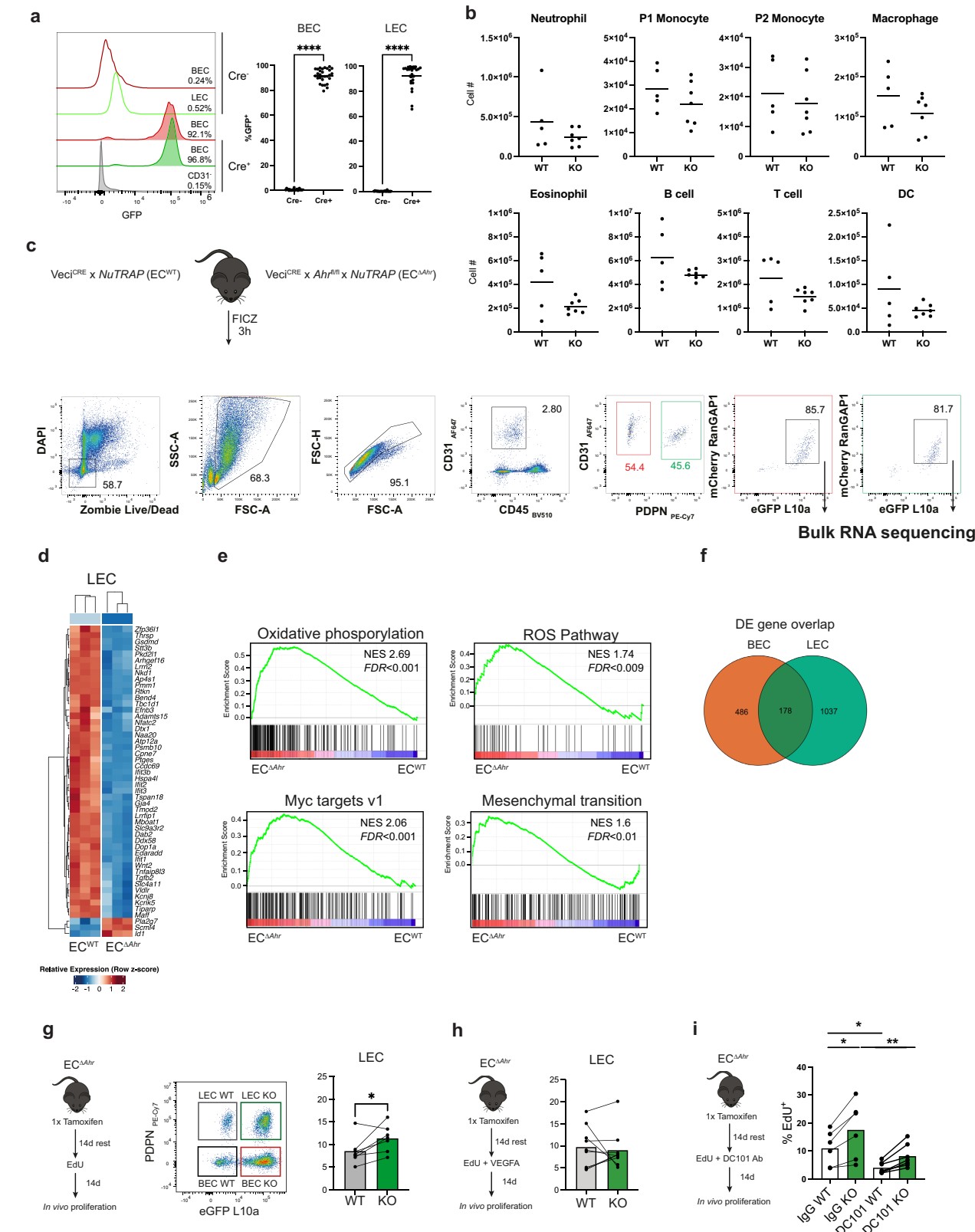

**Extended Data Fig. 5 |** See next page for caption.

**Extended Data Fig. 5 | RNA sequencing of AHR-deficient intestinal LEC.**
**a**, Histogram overlays showing Cre efficiency and specificity in SI BEC
(CD45⁻CD31⁺PDPN⁻; red) and LEC (CD45⁻CD31⁺PDPN⁻; green) from EC$^{\Delta Ahr}$ mice
treated with a full (5x) tamoxifen regimen (shaded histograms) comapred to
tamoxifen-treated littermate EC$^{WT}$ controls. Grey bar shows negligable Cre⁺
cells within CD31⁻ gate. Graphs show summary data (EC$^{WT}$ n = 24; EC$^{\Delta Ahr}$ n = 29).
**b**, Composition of immune cell subsets between EC$^{WT}$ and EC$^{\Delta Ahr}$ mice during
tamoxifen treatment (analysed day 7 of 14 day course, n = 5–7 mice/genotype).
Immune cell subsets defined as in Extended Data Fig. 6a. **c**, Experimental setup
and gating strategy for RNA sequencing experiments. 3h post-FICZ treatment,
NuTRAP-reporter⁺ (mCherry-RanGAP1⁺eGFP-L10a⁺) BEC and LEC from SI and
colon of EC$^{\Delta Ahr}$ and EC$^{WT}$ controls were sorted by FACS and sequenced. **d**, Small
intestinal LEC – expression of top 50 DE genes (based on adjusted p value)

between EC$^{\Delta Ahr}$ (*Ahr*-deficient) and EC$^{WT}$ (*Ahr* WT). **e**, Results of GSEA analysis
of selected Hallmark pathways in EC$^{\Delta Ahr}$ LEC, displayed as a barcode plot
(NES, normalised enrichment score; FDR, false discovery rate). **f**, Venn diagram
showing number of DE genes that overlap in BEC and LEC. **g**, Homeostatic
proliferation: EdU detection following 2-week EdU feeding in WT and KO
LEC from suboptimally-depleted EC$^{\Delta Ahr}$ mice (n = 7 mice). Gating example of WT
and KO BEC and LEC selection (within total CD31⁺ gate) shown. **h**, VEGFA
administration: EdU detection in EC$^{\Delta Ahr}$ mice WT and KO LEC following 2-week
VEGFA$_{165}$ administration alongside EdU feeding (n = 9 mice). **i**, DC101-mediated
blockade: EdU detection in EC$^{\Delta Ahr}$ mice WT and KO LEC following VEGFR2
blockade with DC101 antibody (n = 7 mice), or control IgG alongside EdU
feeding (n = 6 mice). *P < 0.05, **P < 0.01, ****P < 0.0001, as calculated by paired
t-tests (**g**,**i**). or unpaired t-tests (**i** – WT-WT, and KO-KO comparisons only).

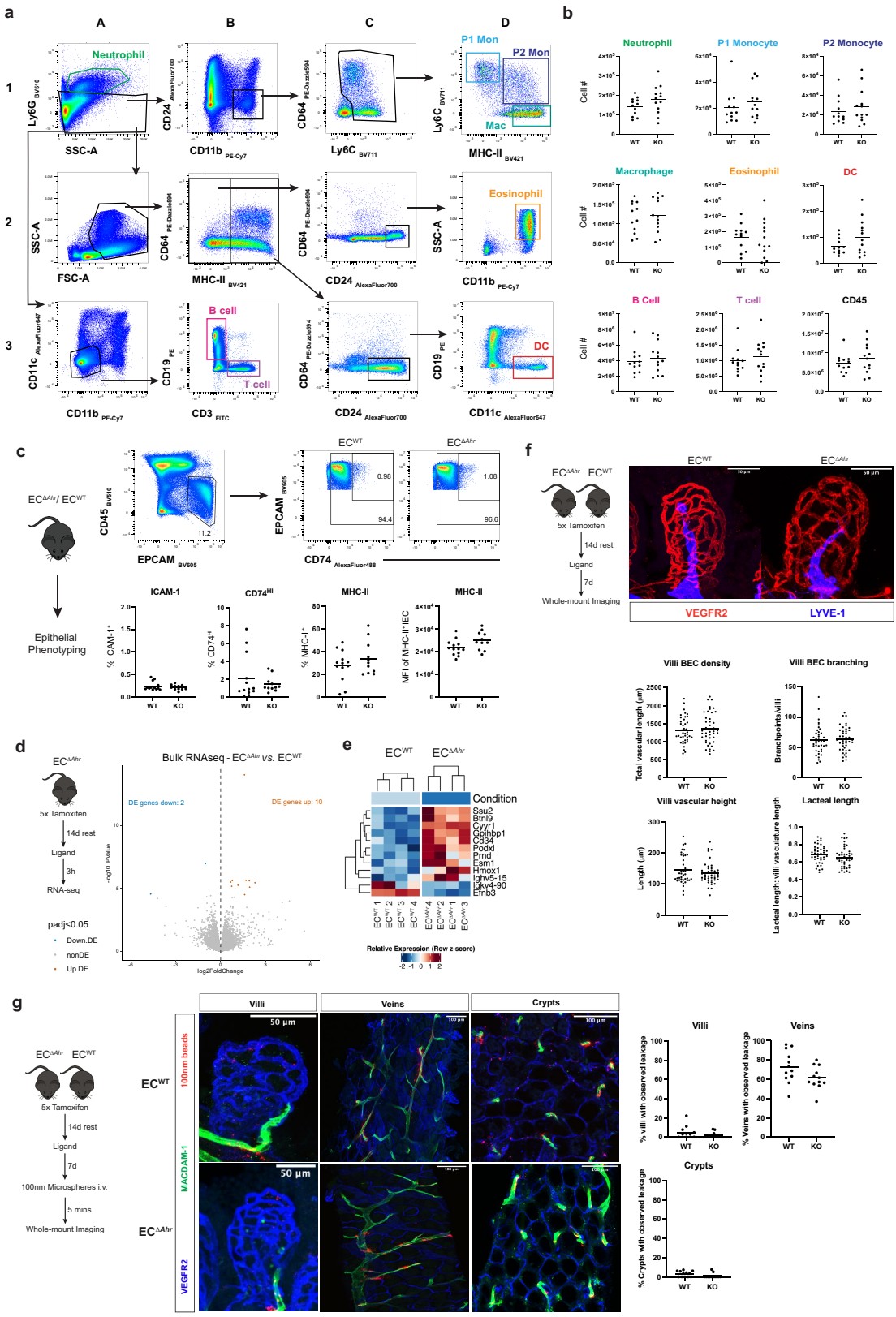

**Extended Data Fig. 6** | See next page for caption.

**Extended Data Fig. 6 | Endothelial AHR-deficiency does not directly impact enteric stromal cell activation. a**, Representative gating strategy to target immune populations in the gut, based off refs. 53,54 Arrows indicate direction of gating. Coloured gates indicate final population designation. **b**, summary data of 8 defined populations between fully tamoxifen treated EC$^{WT}$ (WT, n = 12) and EC$^{\Delta Ahr}$ (KO, n = 13) as total cell numbers/SI lamina propria. Data from two independent experiments shown. **c**, SI epithelial cells (EPCAM$^+$) activation marker profiling between fully-tamoxifen treated EC$^{\Delta Ahr}$ and EC$^{WT}$ mice. Representative gating shows selection of EPCAM$^+$ intestinal epithelial cells (IECs), and gate setting for CD74$^{LO}$ and CD74$^{HI}$ IECs. Summary data shows marker expression within IECs between EC$^{WT}$ (WT, n = 13) and EC$^{\Delta Ahr}$ (KO, n = 11), data from 2 independent experiments. **d**, Bulk RNA sequencing experimental set-up and resulting volcano plot comparing whole tissue sampling from fully-tamoxifen treated EC$^{\Delta Ahr}$ and EC$^{WT}$ mice. Mice (n = 4/group) were treated with FICZ 3 h before tissue collection. Volcano plot DE genes shown in red (significantly increased) and blue (significantly decreased). **e**, Relative expression of all significant genes between EC$^{\Delta Ahr}$ and EC$^{WT}$: individual replicates shown in columns. Genes in rows. **f**, VEGFR2 (red) and LYVE1 (blue) whole-mount immunostaining in EC$^{\Delta Ahr}$ and EC$^{WT}$ mice treated with 3-MC (7 days). Quantification shows blood vascular density per villi, blood vascular branchpoints/villi, blood vascular cage height, and relative lacteal length per villi (lacteal height: blood vascular cage height). EC$^{WT}$ n = 41, EC$^{\Delta Ahr}$ n = 43 villi from 4 mice/group. Points represent individual villi, lines represent means. **g**, EC$^{\Delta Ahr}$ and EC$^{WT}$ mice treated as in **f**, with additional infusion of 100 nm fluorescent microspheres 5 min before tissue collection. Images show representative villi, submucosal veins, and vasculature around villus crypts (VEGFR2 – blue, MADCAM-1 – green, 100 nm beads – red). Quantification shows % of villi, crypt vessels, or veins per image with clear bead distribution outside the vessels. Points represent images taken (n = 12/group from 4 mice/group), lines show means.

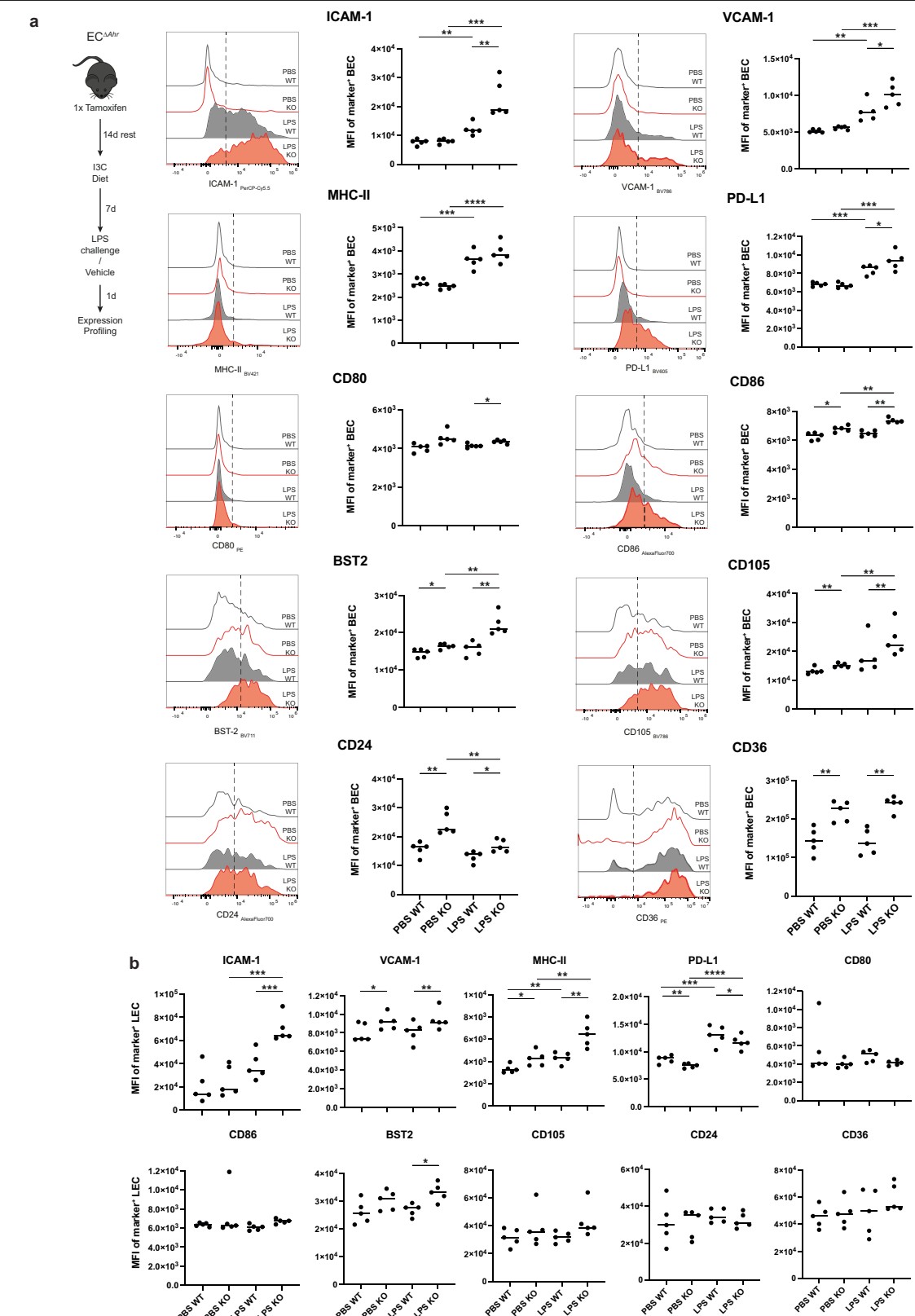

**Extended Data Fig. 7 | AHR-deficiency alters gut EC inflammatory activation profiles. a**, Suboptimally deleted EC$^{ΔAhr}$ mice were fed I3C containing diet for 7 days followed by 24h i.p. LPS challenge (or vehicle control – PBS), before activation and inflammatory surface markers compared by flow cytometry in WT and KO BEC within the same animals within both PBS-, and LPS-treated conditions. Representative histograms show position of positive gates, bar plots show MFI of positive BEC for each marker in the 4 conditions (n = 5/group, lines represent means). **b**, as in part a, but examining WT and KO LEC from suboptimally deleted EC$^{ΔAhr}$ mice following LPS or PBS vehicle treatment ((n = 5/group, lines represent means). *P < 0.05, **P < 0.01, ***P < 0.001, ****P < 0.0001 as calculated by unpaired t-tests (**a-b**, PBS vs. LPS comparisons), and paired t-tests (**a-b**, WT vs. KO within same mouse).

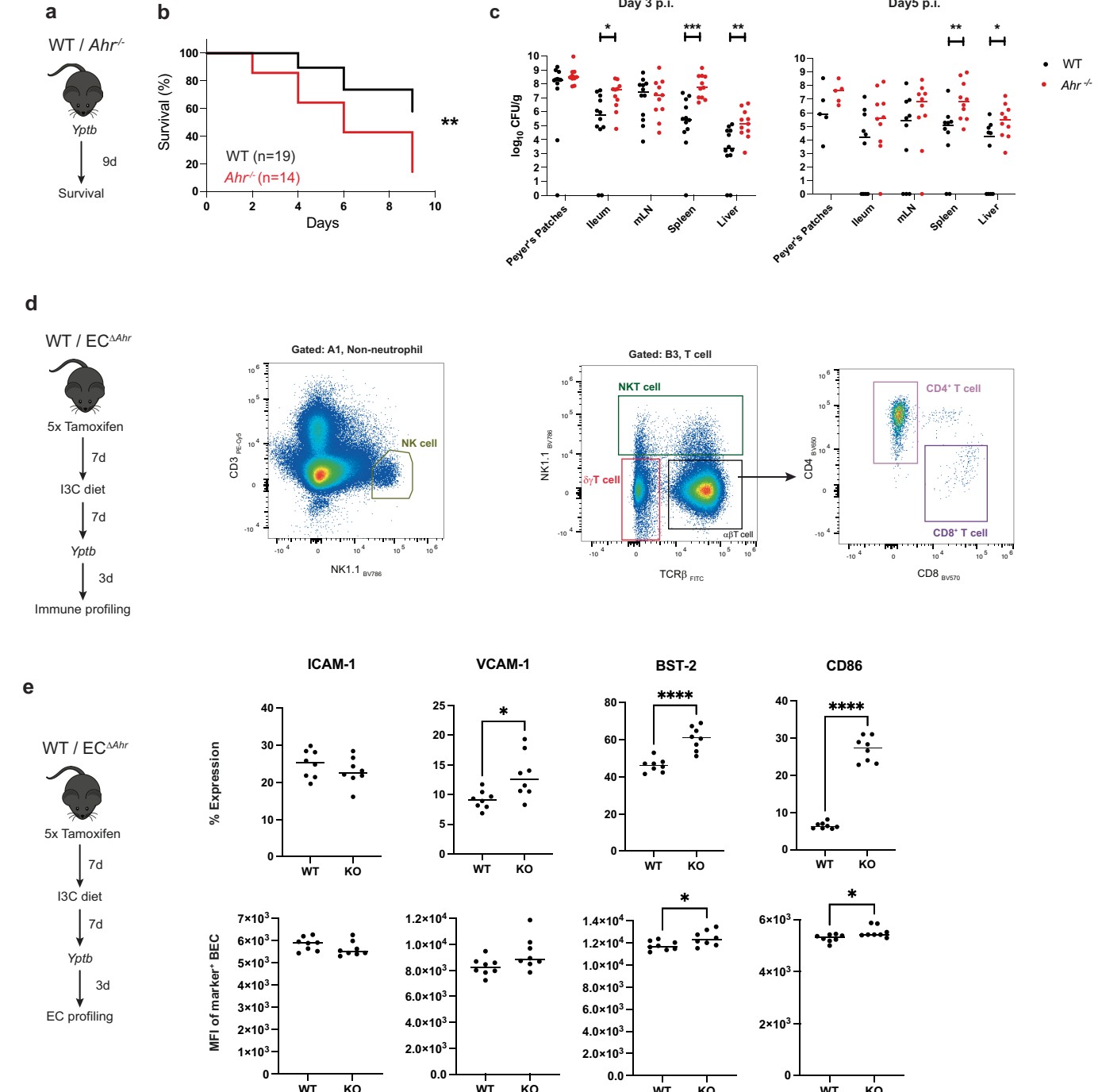

**Extended Data Fig. 8 | Global *Ahr*-deficiency leads to increased susceptibility to enteric infection. a**, Globally *Ahr*-deficient (*Ahr*⁻/⁻) mice or WT controls were infected with *Yptb* (5 x 10⁸ CFU/mouse) and survival recorded. **b**, Survival curve comparing *Ahr*⁻/⁻ (n = 14) and WT (n = 19) mice. Data combined from two experiments. **c**, colony-forming unit (CFU) determination in the 5 tissues shown at day 3 (left, n = 11-12/group) or day 5 (right, n = 5–10/group) following *Yptb* infection (5 x 10⁸ CFU/mouse) in WT and *Ahr*⁻/⁻ mice. Data combined from two experiments. **d**, Gating strategy for additional immune cell populations quantified in Fig. 3k. Gating for NK cells, NKT cells, γδT cells, CD4⁺ αβT cells, and CD8⁺ αβT cells shown. Input gates (as represented in Extended Data Fig. 6a) listed above plots. **e**, Inflammatory activation marker profiling of BEC in WT or EC^ΔAhr mice 3 days after *Yptb* infection (5 x 10⁷ CFU/mouse) by flow cytometry (n = 8 mice/group). *P < 0.05, **P < 0.01 ***P < 0.001, *P < 0.0001 as calculated by Gehan-Breslow-Wilcoxon tests (**b**), by unpaired t-tests(**c**), or by paired t-tests (**e**).

**a**

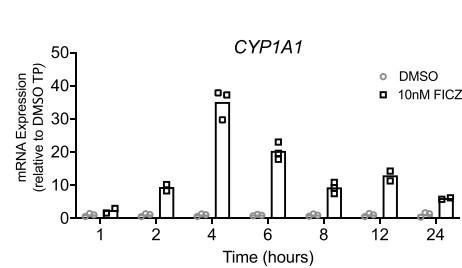

**b**

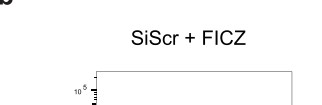

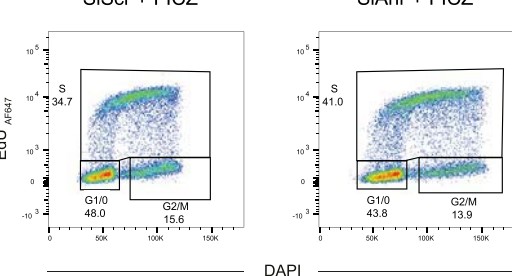

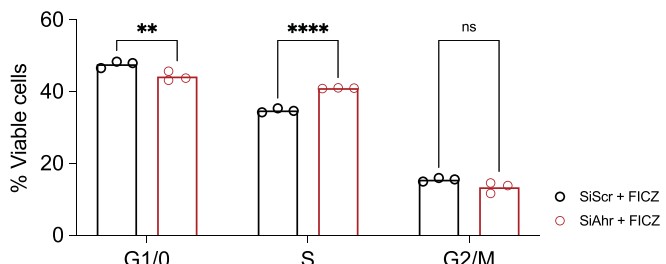

**c**

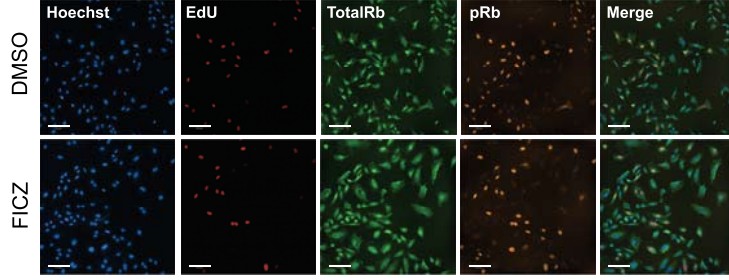

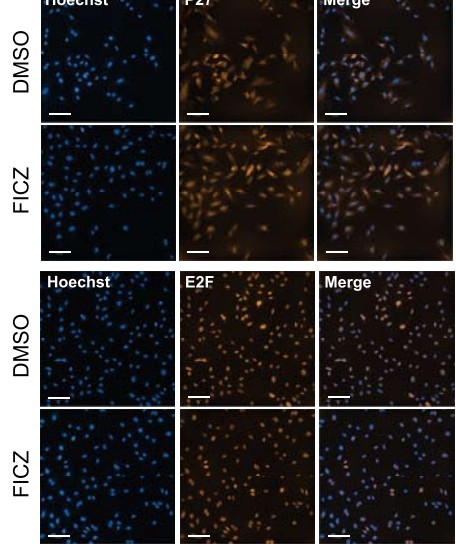

**Extended Data Fig. 9 | Responses of human endothelia to AHR ligand stimulation and AHR deficiency. a**, CYP1A1 expression (qPCR) in HUVECs at different timepoints following either FICZ, or DMSO control treatment (n = 2-3/timepoint). **b**, Representative flow plots and combined data of AHR-knockdown HUVECs (siAHR) or siRNA control (siScr) subjected to flow cytometric cell cycle analysis. Both AHR-knockdown and control HUVEC cultured with 100 nM FICZ following siRNA treatment (n = 3/group). **c**, Representative images from combined single-cell imaging data shown in Fig. 4e. Hoechst – blue, EdU – red, totalRb – green, pRb – orange, p27 – orange. Scale bars represent 100 μm. Data representative of 3 independent experiments. Bar heights represent means throughout. **$P$ < 0.01, ****$P$ < 0.0001 as calculated by two-way ANOVA with Šidák's multiple comparisons tests.

# Reporting Summary

## Statistics

For all statistical analyses, confirm that the following items are present in the figure legend, table legend, main text, or Methods section.

| n/a | Confirmed | |
|---|---|---|
| ☐ | ☒ | The exact sample size (*n*) for each experimental group/condition, given as a discrete number and unit of measurement |
| ☒ | ☐ | A statement on whether measurements were taken from distinct samples or whether the same sample was measured repeatedly |
| ☐ | ☒ | The statistical test(s) used AND whether they are one- or two-sided<br>*Only common tests should be described solely by name; describe more complex techniques in the Methods section.* |
| ☒ | ☐ | A description of all covariates tested |
| ☐ | ☒ | A description of any assumptions or corrections, such as tests of normality and adjustment for multiple comparisons |
| ☐ | ☒ | A full description of the statistical parameters including central tendency (e.g. means) or other basic estimates (e.g. regression coefficient) AND variation (e.g. standard deviation) or associated estimates of uncertainty (e.g. confidence intervals) |
| ☐ | ☒ | For null hypothesis testing, the test statistic (e.g. *F*, *t*, *r*) with confidence intervals, effect sizes, degrees of freedom and *P* value noted<br>*Give P values as exact values whenever suitable.* |
| ☒ | ☐ | For Bayesian analysis, information on the choice of priors and Markov chain Monte Carlo settings |
| ☒ | ☐ | For hierarchical and complex designs, identification of the appropriate level for tests and full reporting of outcomes |
| ☒ | ☐ | Estimates of effect sizes (e.g. Cohen's *d*, Pearson's *r*), indicating how they were calculated |

*Our web collection on statistics for biologists contains articles on many of the points above.*

## Software and code

Policy information about availability of computer code

| | |
|---|---|
| Data collection | The following software was used for data collection: scRNAseq/bulk RNAseq data - Illumina RTA v2.11.3; flow/spectral cytometry - BD FACSDiva v9 and Cytek Spectroflo v3; Confocal Imaging - Leica LAS-AF v2.7.3.9723 and Zeiss ZEN Black v14.0.27.201; qPCR - BioRad CFX Maestro v1.1; Single cell imaging - PerkinElmer Harmony 4.9 software. |
| Data analysis | Code used for data analysis of scRNAseq dataset included CellRanger v.4.0.0 (demultiplexing, basecall conversion, genome alignment, barcode/UMI counting), blc2fastq v.2.17.1.14 (file conversion), Seurat v.3.2.0 (filtering, clustering, conserved and DE gene analysis), EnrichR web app (gene set enrichment), Pagoda2 v.1.0.8 (overdispersion analysis), and SCENIC v.1.2.4 (regulatory network analysis). ggplot2 v.3.3.3 and ComplexHeatmap v2.2.0 were used for scRNAseq data visualisation. Code used for data analysis of RNAseq datasets (3 mouse datasets, 1 human dataset) included Illumina RTA v.2.11.3 (demultiplexing and basecalling), STAR v.2.2.7a and tophat2 v.2.0.11 (genome alignment for mouse and human datasets respectively), DESeq2 v1.30.1 (normalisation and DE analysis). Flow cytometry data was analysed with FlowJo v10.6 (FlowJO LLC), Confocal data with FIJ v.2.9.0 (ImageJ), and single cell imaging with Harmony v4.9 software (PerkinElmer). Data presentation and statistical analysis performed with GraphPad Prism v9 for non-sequencing data. |

For manuscripts utilizing custom algorithms or software that are central to the research but not yet described in published literature, software must be made available to editors and reviewers. We strongly encourage code deposition in a community repository (e.g. GitHub). See the Nature Portfolio guidelines for submitting code & software for further information.

# Data

Policy information about availability of data

All manuscripts must include a data availability statement. This statement should provide the following information, where applicable:
- Accession codes, unique identifiers, or web links for publicly available datasets
- A description of any restrictions on data availability
- For clinical datasets or third party data, please ensure that the statement adheres to our policy

All sequencing data (murine scRNAseq, murine RNAseq, and human RNAseq datasets) have been deposited in the NCBI gene expression omnibus as a superseries – accession number GSE201789. Mouse genome mm10 sequences were retrieved from retrieved from GENCODE mouse genome (GRCm38), version M23 (Ensembl 98) https://www.gencodegenes.org/mouse/release_M23.html. In SCENIC analysis, RcisTarget was used (https://resources.aertslab.org/cistarget/) and database mm10__refseq-r80__500bp_up_and_100bp_down_tss.mc9nr.feather was downloaded for our analysis. scRNAseq metrics/metadata, conserved and DE gene lists from sequencing experiments, Regulons (generated by SCENIC), genesets that comprise Pagoda2 outputs (Aspects) and input genesets for Pagoda2 overdispersion analysis are all provided as supplementary tables. All other flow cytometry, images and qPCR data are presented within the manuscript. All raw data is provided as source data files accompanying the manuscript.

# Human research participants

Policy information about studies involving human research participants and Sex and Gender in Research.

| Reporting on sex and gender | N/A |
| --- | --- |
| Population characteristics | N/A |
| Recruitment | N/A |
| Ethics oversight | N/A |

Note that full information on the approval of the study protocol must also be provided in the manuscript.

# Field-specific reporting

Please select the one below that is the best fit for your research. If you are not sure, read the appropriate sections before making your selection.

☒ Life sciences  ☐ Behavioural & social sciences  ☐ Ecological, evolutionary & environmental sciences

For a reference copy of the document with all sections, see nature.com/documents/nr-reporting-summary-flat.pdf

# Life sciences study design

All studies must disclose on these points even when the disclosure is negative.

| Sample size | No statistical methods were used to estimate sample sizes before the study. We based our sample numbers on a combination of preliminary data from pilot studies, previous experience with in vivo and in vitro systems (Schiering et al. Nature 2017, Metidji et al. Immunity 2018), and standards in the field. |
| --- | --- |
| Data exclusions | scRNAseq thresholding involved excluding cells with abnormally high or low transcripts, and based on high mitochondrial gene content as is standard practice for downstream analysis of this sequencing data. We also excluded doublets, and removed contaminant clusters to improve stringency, as described in the methods. For each BEC and LEC RNAseq, 1 control and 1 experimental repeat were excluded from the original n=4/group based on low Cyp1a1 activation and highly divergent transcriptome (experimental samples), and highly divergent transcriptome only (control samples). |
| Replication | With the exception of scRNAseq due to the nature of the analysis, all experiments were replicated 2-3 times in this study unless otherwise stated. |
| Randomization | Within genotypes, all animals were age and sex matched, and randomized prior to experimentation. When genotype constituted experimental group, mice were age and sex matched as close as possible. Experimental and control cage-matched littermates were used in all mouse experiments with two exceptions: 1) bulkRNAseq experiments where control and experimental mice were both Cre+ from different strains, 2) infection studies where mixing would confound the experiment due to possible enhanced contagiousness of control and experimental groups. For infections of WT mice, mice were randomly assigned to experimental or control groups before the experiments, cage densities matched, and rack positions remained as close as possible. Experimental procedures always involved processing control and experimental samples in a random order, rather than by condition. |
| Blinding | Blinding was not used in this study as the researchers must predetermine genotype before the study in order to obtain equal group sizes, minimize animal usage, achieve appropriate randomization and prevent infection-based artifacts (see above). |

# Reporting for specific materials, systems and methods

We require information from authors about some types of materials, experimental systems and methods used in many studies. Here, indicate whether each material, system or method listed is relevant to your study. If you are not sure if a list item applies to your research, read the appropriate section before selecting a response.

## Materials & experimental systems

| n/a | Involved in the study |
|-----|----------------------|
| ☐ | ☒ Antibodies |
| ☒ | ☐ Eukaryotic cell lines |
| ☒ | ☐ Palaeontology and archaeology |
| ☐ | ☒ Animals and other organisms |
| ☒ | ☐ Clinical data |
| ☒ | ☐ Dual use research of concern |

## Methods

| n/a | Involved in the study |
|-----|----------------------|
| ☒ | ☐ ChIP-seq |
| ☐ | ☒ Flow cytometry |
| ☒ | ☐ MRI-based neuroimaging |

## Antibodies

| Antibodies used | For flow cytometry:<br>anti-mouse CD105 BV786, clone MJ7/18, BD Biosciences cat #564746<br>anti-mouse CD11b PE-Cy7, clone M1/70, BD Biosciences cat #552850<br>anti-mouse CD11c AlexaFluor647, clone N418, Biolegend cat #117312<br>anti-mouse CD19 PE, clone 6D5, cat #115508<br>anti-mouse CD24 AF700, clone M1/69, Biolegend cat #101836<br>anti-mouse CD3 FITC, clone 17A2, Biolegend cat #102405<br>anti-mouse CD3 PE-Cy5, clone 17A2, Biolegend cat #100274<br>anti-mouse CD31 AlexaFluor647, clone MEC13.3, Biolegend cat #102516<br>anti-mouse CD4 BV650, clone RM4-5, Biolegend cat #100545<br>anti-mouse CD45 BV510, clone 30-F11, BD Biosciences cat #563891<br>anti-mouse CD45 PerRCP-Cy5.5, clone 30-F11, Biolegend cat #103132<br>anti-mouse CD64 PE-Dazzle594, clone X54-5/7.1, Biolegend cat #139320<br>anti-mouse CD74 AlexaFluor488, clone In1/CD74, Biolegend cat #151005<br>anti-mouse CD8α BV570, clone 53-6.7, Biolegend cat #100740<br>anti-mouse CD80 PE, clone 16-10A1, Biolegend cat #104708<br>anti-mouse CD86 AlexaFluor700, clone GL-1, Biolegend cat #105024<br>anti-mouse EpCAM BV605, clone G8.8, Biolegend cat #147303<br>anti-mouse ICAM-1 PeRCP Cy5.5, clone YN1/1.7.4, Biolegend cat #116124<br>anti-mouse Ly6C BV711, clone HK1.4, Biolegend cat #128037<br>anti-mouse Ly6G BV510, clone IA8, Biolegend cat #127633<br>anti-mouse I-A/I-E BV421, clone M5/114.152, Biolegend cat #107632<br>anti-mouse NK1.1 BV786, clone PK136, Biolegend cat #108749<br>anti-mouse PD-L1 BV605, clone 10F.9G2, Biolegend cat #124301<br>anti-mouse PDPN PE-Cy7, clone 8.1.1, Biolegend cat #127412<br>anti-mouse TCRβ FITC, clone H57-597, Biolegend cat #109215<br>anti-mouse VCAM-1 BV786, clone 429 (MVCAM.A), BD Biosciences cat #740865<br>anti-mouse TruStain Fx (blocking antibody anti-CD16/32), clone 93, Biolegend cat #101320<br><br>For confocal microscopy:<br><br>Primary antibodies:<br>AlexaFluor488-conjugated rabbit anti-mouse GFP, polyclonal, ThermoFisher cat #A-21311<br>Goat anti-mouse VEGFR2, polyclonal, R&D Systems cat #AF644<br>Rat anti-mouse Lyve-1, clone 223322, R&D systems cat #MAB2125<br>Rabbit anti-mouse Lyve-1, polyclonal, Abcam cat #Ab14917<br>Rat anti-mouse CD31, clone MEC13.3, BD Biosciences cat #553370<br>Goat anti-mouse ESM1, polyclonal, R&D Systems cat #AF1999<br>Rat anti-mouse MADCAM-1, clone MECA-367, Biolegend cat #120702<br><br>Secondary antibodies:<br>AlexaFluor555-conjugated Donkey anti-goat IgG, ThermoFisher cat #A-21432<br>AlexaFluor647-conjugated Donkey anti-goat IgG, ThermoFisher cat #A-21447<br>AlexaFluor647-conjugated Donkey anti-rabbit IgG, ThermoFisher cat #A-31573<br>AlexaFluor488-conjugated Donkey anti-rat IgG, ThermoFisher cat #A-21208<br>AlexaFluor594-conjugated Donkey anti-rat IgG, ThermoFisher cat #A-21209<br>Secondary antibodies:<br>AF555-conjugated donkey anti-goat IgG, ThermoFisher cat #A-21432<br>AF594-conjugated donkey anti-rat IgG, ThermoFisher cat #A-21209<br><br>Single-cell imaging: |

Primary antibodies:
anti-human phospho-Rb, clone D20B12, Cell Signalling Technology cat #8156
anti-human Rb, clone 4H1, Cell Signalling Technology cat #9309
anti-human p27 kip1, clone D37H1, Cell Signalling Technology cat #3688
anti-human E2F1, clone EPR3818(3), Abcam cat #ab179445

Secondary antibodies:
AF488-conjugated goat anti-mouse IgG, ThermoFisher cat #A-11001
AF568-conugated goat anti-rabbit IgG, ThermoFisher cat #A-11004

Validation

All antibodies are commercially available, and were validated for specificity and application by manufacturers listed above (Biolegend, BD Biosciences ThermoFisher, R&D Systems, Cell Signalling Technology, Abcam). Antibodies for flow cytometry were titrated in digestion-matched small intestinal samples before use by assessing expression within CD31+ endothelial cell populations, compared to CD45+ and CD31-CD45- non endothelial controls. Antibodies for imaging were used at concetrations suggested in previous published methodologies (https://www.nature.com/articles/nprot.2016.092), or titrated in-house.

See following links for manufacturer validation for flow cytometry antibodies:
anti-mouse CD105 BV786, clone MJ7/18, BD Biosciences cat #564746 https://www.bdbiosciences.com/en-us/products/reagents/flow-cytometry-reagents/research-reagents/single-color-antibodies-ruo/bv786-rat-anti-mouse-cd105.564746
anti-mouse CD11b PE-Cy7, clone M1/70, BD Biosciences cat #552850 https://www.bdbiosciences.com/en-us/products/reagents/flow-cytometry-reagents/research-reagents/single-color-antibodies-ruo/pe-cy-7-rat-anti-cd11b.552850
anti-mouse CD11c AlexaFluor647, clone N418, Biolegend cat #117312 https://www.biolegend.com/en-us/products/alexa-fluor-647-anti-mouse-cd11c-antibody-2703
anti-mouse CD19 PE, clone 6D5, cat #115508, https://www.biolegend.com/en-us/products/pe-anti-mouse-cd19-antibody-1530
anti-mouse CD24 AF700, clone M1/69, Biolegend cat #101836, https://www.biolegend.com/en-us/products/alexa-fluor-700-anti-mouse-cd24-antibody-12790
anti-mouse CD3 FITC, clone 17A2, Biolegend cat #102405,  https://www.biolegend.com/en-us/products/fitc-anti-mouse-cd31-antibody-120
anti-mouse CD3 PE-Cy5, clone 17A2, Biolegend cat #100274, https://www.biolegend.com/en-us/products/pe-cyanine5-anti-mouse-cd3-antibody-21198
anti-mouse CD31 AlexaFluor647, clone MEC13.3, Biolegend cat #102516, https://www.biolegend.com/en-us/products/alexa-fluor-647-anti-mouse-cd31-antibody-3094
anti-mouse CD4 BV650, clone RM4-5, Biolegend cat #100545, https://www.biolegend.com/en-us/products/brilliant-violet-650-anti-mouse-cd4-antibody-7634
anti-mouse CD45 BV510, clone 30-F11, BD Biosciences cat #563891, https://www.bdbiosciences.com/en-us/products/reagents/flow-cytometry-reagents/research-reagents/single-color-antibodies-ruo/bv510-rat-anti-mouse-cd45.563891
anti-mouse CD45 PerRCP-Cy5.5, clone 30-F11, Biolegend cat #103132, https://www.biolegend.com/en-us/products/percp-cyanine5-5-anti-mouse-cd45-antibody-4264
anti-mouse CD64 PE-Dazzle594, clone X54-5/7.1, Biolegend cat #139320, https://www.biolegend.com/en-us/products/pe-dazzle-594-anti-mouse-cd64-fcgammari-antibody-12424
anti-mouse CD74 AlexaFluor488, clone In1/CD74, Biolegend cat #151005, https://www.biolegend.com/en-us/products/alexa-fluor-488-anti-mouse-cd74-clip-antibody-16473
anti-mouse CD8α BV570, clone 53-6.7, Biolegend cat #100740, https://www.biolegend.com/en-us/products/brilliant-violet-570-anti-mouse-cd8a-antibody-7377
anti-mouse CD80 PE, clone 16-10A1, Biolegend cat #104708, https://www.biolegend.com/en-us/products/pe-anti-mouse-cd80-antibody-43
anti-mouse CD86 AlexaFluor700, clone GL-1, Biolegend cat #105024, https://www.biolegend.com/en-us/products/alexa-fluor-700-anti-mouse-cd86-antibody-3410
anti-mouse EpCAM BV605, clone G8.8, Biolegend cat #147303, https://www.biolegend.com/en-us/products/pe-anti-mouse-human-cd324-e-cadherin-antibody-9276
anti-mouse ICAM-1 PeRCP Cy5.5, clone YN1/1.7.4, Biolegend cat #116124, https://www.biolegend.com/en-us/products/percp-cyanine5-5-anti-mouse-cd54-antibody-14748
anti-mouse Ly6C BV711, clone HK1.4, Biolegend cat #128037, https://www.biolegend.com/en-us/products/brilliant-violet-711-anti-mouse-ly-6c-antibody-8935
anti-mouse Ly6G BV510, clone IA8, Biolegend cat #127633, https://www.biolegend.com/en-us/products/brilliant-violet-510-anti-mouse-ly-6g-antibody-9121
anti-mouse I-A/I-E BV421, clone M5/114.152, Biolegend cat #107632, https://www.biolegend.com/en-us/products/brilliant-violet-421-anti-mouse-i-a-i-e-antibody-7147
anti-mouse NK1.1 BV786, clone PK136, Biolegend cat #108749, https://www.biolegend.com/en-us/products/brilliant-violet-785-anti-mouse-nk-1-1-antibody-10367
anti-mouse PD-L1 BV605, clone 10F.9G2, Biolegend cat #124301, https://www.biolegend.com/en-us/products/purified-anti-mouse-cd274-b7-h1-pd-l1-antibody-4481
anti-mouse PDPN PE-Cy7, clone 8.1.1, Biolegend cat #127412, https://www.biolegend.com/en-us/products/pe-cyanine7-anti-mouse-podoplanin-antibody-6674
anti-mouse TCRβ FITC, clone H57-597, Biolegend cat #109215, https://www.biolegend.com/en-us/products/alexa-fluor-488-anti-mouse-tcr-beta-chain-antibody-2713
anti-mouse VCAM-1 BV786, clone 429 (MVCAM.A), BD Biosciences cat #740865, https://www.bdbiosciences.com/en-us/products/reagents/flow-cytometry-reagents/research-reagents/single-color-antibodies-ruo/bv786-rat-anti-mouse-cd106.740865
anti-mouse TruStain Fx (blocking antibody anti-CD16/32), clone 93, Biolegend cat #101320, https://www.biolegend.com/en-us/products/trustain-fcx-anti-mouse-cd16-32-antibody-5683

## Animals and other research organisms

Policy information about studies involving animals; ARRIVE guidelines recommended for reporting animal research, and Sex and Gender in Research

| | |
|---|---|
| Laboratory animals | The study used the following mouse strains: Cdh5(PAC)Cre/ERT2Ahrfl/f, Cdh5(PAC)Cre/ERT2Ahrfl/f NuTRAP, Cyp1a1CreR26LSL-eYFP, and Ahr-/-. All mice were bred onto a C57/B6 background. Mice were housed in individually ventilated cages, at ambient temperatures (19-21C), and subjected to a standard 12:12 hour light:dark cycle. Mice were between 6-16 weeks at time of experiments, and subjected to tamoxifen-mediated Cre depletion between 5 and 8 weeks of age. |
| Wild animals | This study did not involve wild animals. |
| Reporting on sex | Male animals were used throughout, except in yersinia pseudotuberculosis infection experiments when a combination of male (Extended Data Fig. 8c, Fig. 3h) and female (Extended Data Fig. 7b, Fig 3g) mice were used. |
| Field-collected samples | This study did not involve samples collected from the field. |
| Ethics oversight | Mouse studies were approved by and in compliance with local AWERB ethics committee as well as UK home office regulations. |

Note that full information on the approval of the study protocol must also be provided in the manuscript.

## Flow Cytometry

### Plots

Confirm that:

☒ The axis labels state the marker and fluorochrome used (e.g. CD4-FITC).

☒ The axis scales are clearly visible. Include numbers along axes only for bottom left plot of group (a 'group' is an analysis of identical markers).

☒ All plots are contour plots with outliers or pseudocolor plots.

☒ A numerical value for number of cells or percentage (with statistics) is provided.

### Methodology

| | |
|---|---|
| Sample preparation | Mouse tissue preparation and staining:<br>Following harvesting and fat removal, small intestine (SI) and colon were cut open longitudinally, and underwent an IEL wash: incubated with IEL wash buffer (IMDM +1%FCS, 5 mM EDTA, 10 mM HEPES, penicillin/streptomycin, and 2 mM DTT) for 20 min at 37°C with 200rpm. shaking. Small intestine was washed and vortexed at low speeds in SI PBS (PBS +5mM EDTA+10mM HEPES) 30seconds x 3 times or until clear, and then vortexed at low speed in PBS. Colon was vortexed in PBS once. Both gut tissues were then cut into small pieces and incubated in digestion buffer. All other tissues (BAT, iWAT, liver, lung, kidney spleen) were cut into small pieces and incubated directly in digestion buffer.<br><br>All organs were digested by incubation in Collagenase A digestion buffer (4mls/tissue: HBSS + 20mM HEPES, 10mg/ml Collagenase A (Sigma), 8U/ml Dispase II (Sigma), 50 g/ml DNase I (Sigma) for 20 minutes at 37 C with 200rpm shaking. Reactions stopped through addition of 1:1 complete media (IMDM +1%FCS, penicillin/streptomycin, 1x glutamax) and: passing through 100 m filters (SI, colon, kidney, lung, spleen); debris removal through 2x 1minute 60g centrifugation, harvesting supernatant and then passing through 100 m filters (liver); or 250g 10-minute centrifugation and careful floating adipocyte fraction removal (BAT, iWAT). After centrifugation (400g, 8 minutes), SI and colon were subjected to 40% Percoll (Amersham) density gradient centrifugation (400g, 8 minutes) to remove debris, while all other tissues underwent resuspension in 0.5ml ACK lysis buffer for 2 minutes and washing to remove erythrocytes. Finally, cells were filtered (40 m) and counted before downstream analysis.<br><br>Cell suspensions were incubated with anti-mouse CD16/CD32 (TrustainFx, Biolegend), before incubation with live/dead dye (zombie near infa-red, Biolegend), and incubation with surface antibodies. All antibodies and subsequent washes in FACS buffer (PBS + 2% FCS, 2mM EDTA). For EdU detection, Click-iT EdU proliferation kit – pacific blue (ThermoFisher) was used, according to manufacturer's instructions. DAPI nuclear stain added directly <5minutes prior to analysis/sorting. A combination of compensation beads (OneComp/Ultracomp eBeads, ThermoFisher) for antibody controls, and single-stained cells for dyes and fluorescent proteins were used for controls and for compensation/spectral unmixing.<br><br>Human cell preparation and staining:<br>Primary human umbilical vein endothelial cells (HUVEC; Lonza, UK) obtained from pooled donors were seeded at 60-80% confluency and cultured in Endothelial Cell Growth Medium-2 media (EGM-2 bulletkit; Lonza) supplemented with penicillin/streptomycin(P/S) at 5% CO2 and 37oC. Cells were switched to minimal growth media (EGM-2 Bullet kit basal medium, Lonza; supplemented with P/S) for 24h prior to experimentation.<br><br>Click-iT EdU proliferation kit AlexaFluor647 (ThermoFisher, C10340) was used according to manufacturer's instructions. Briefly, this involved a 1h EdU pulse to cells prior to washing, fixing, DAPI staining, and analysis. |
| Instrument | BD LSRII flow cytometer; Cytek Aurora spectral cytometer; BD Aria Fusion cell sorter. |

| Software | Data was collected with BD FACSDiva v9 software (BD instruments), or SpectroFlo v3 (Cytek Aurora). All analysis was carried out with FlowJo v10.6 (FlowJo LLC). |
|---|---|
| Cell population abundance | Purity of sorted cells was routinely assessed post-sort, reaching purity of above 95%. |
| Gating strategy | Full gating strategies are shown in Extended Data Fig. 3d, and Extended Data Fig. 5a for flow cytometry and FACS experiments respectively. Gating was set based on following density distributions, verified by comparing known/expected expression levels in BEC and LEC, and compared to non-endothelial populations. |

☒ Tick this box to confirm that a figure exemplifying the gating strategy is provided in the Supplementary Information.

