## [Peer Review File · Nature]

Manuscript Title: Endothelial sensing of AHR ligands regulates intestinal homeostasis

Reviewer Comments & Author Rebuttals

Reviewer Reports on the Initial Version:

Referees' comments:

Referee #1 (Remarks to the Author):

2022-03-03578A

In this work from Wiggins et al., the authors analyze the small intestinal vasculature and endothelial cell responses to AHR ligands. They first performed scRNAseq on small intestinal blood and lymphatic endothelial cells after administration of an AHR ligand and performed in-depth bioinformatic analysis on control ECs. They found a widespread cellular response to the AHR ligand in multiple endothelial cell subsets in the gut and other organs. They then analyzed endothelial gene expression, proliferation and inflammation markers in AHR EC KO mice following injection of an AHR ligand and found increased proliferation and inflammation in AHR KO ECs. They also analyzed AHR KO and AHR BEC KO mice after infection with the pathogen Yptb and found reduced mouse survival. Finally, in vitro, they found AHR ligands restrict proliferation and inflammatory responses.

This manuscript starts with very interesting, high quality scRNAseq data of intestinal BECs and LECs from control mice combined with thorough bioinformatic analysis. There are distinctive, potentially gut-specific, BEC and LEC subsets with interesting properties that could reveal novel endothelial-mediated gut functions. I found the subset-specific transcription factor expression especially fascinating. Unfortunately, however, most of the rest of the manuscript focuses on endothelial cells responses to AHR ligands when AHR expression is not endothelial cell-specific, gut-specific nor seemingly specific to any of the clusters identified by single cell analysis. Endothelial cells present a homogenized response to AHR ligand treatment and conclusions drawn about direct endothelial cell-intrinsic ligand responses could be confounded by secondary effects mediated by AHR ligands on gut immune and epithelial cells. Moreover, the endothelial cell responses to AHR signaling identified by the authors in the gut (reduced proliferation and inflammation) are not novel and have been reported in other contexts. The authors also fail to incorporate their results into emerging themes of intestinal-specific endothelial cell responses to inflammation (recent papers on TAK1 and Casp8 EC KOs) or Vegfa signaling. Lastly, conclusions are often reached without appropriate controls and/or verification of bioinformatic data through FACS or imaging data. With the above concerns I believe this manuscript is better suited to a more specialized journal.

Major comments

1. Single cell data of steady state ECs are of excellent quality and show distinct clusters. Unfortunately, there is no verification of cluster marker genes through other means like FACS or imaging.
2. The authors choose to analyze gut ECs after AHR ligand treatment and observe a mostly homogeneous endothelial response. This is a shame because their data contain interesting observations that could lead to discoveries about gut- or organ-specific endothelial Ahr signaling. Eg, Ahr is expressed in a pan-endothelial fashion in the SI ECs, however, the AHR signaling response gene Cyp1a1 is expressed in only one LEC cluster and Cap 1 and PCV BECs. Identification of the location of these cells by imaging with the Cyp1a1 reporter mouse at steady state could

identify the cellular route whereby AHR ligands enter the bloodstream. Similarly, Ahr expression is higher in LECs than BECs on average (ED Fig 3b), however the steady state expression of Cyp1a1 is low, there is a LEC-specific mechanism to suppress cellular responses to dietary AHR ligands? Along the same lines, ED Fig 3h shows that after AHR ligand treatment BECs and LECs express Cyp1a1 in all organs except the spleen and colon, respectively. Are there organ-specific endothelial mechanisms restricting AHR signaling?

3. Since various immune cells and intestinal epithelial cells express AHR (PMID: 33742166, Shah 2022) and show responses to AHR ligands, with the AHR ligand injection strategy used by the authors it is difficult to assess how much of EC gene expression changes are caused by direct EC AHR signaling or secondary to signaling from other intestinal cell AHR signaling.

4. There are no data showing the deletion efficiency of Ahr in the Ahr fl/fl, Cdh5 CreERT2 mice. The Cre deleter strain used by the authors displays organ-specific deletion efficiency in adult mice.

5. In relation to point 2 and 4, if deletion efficiency is good in these mice, why not perform experiments at steady state to assess the effect of EC Ahr deletion on gut parameters such as inflammation, endothelial proliferation, vessel density, immune cell infiltration, epithelial cells, etc? Ahr germline KO mice display organ-specific decreases in vascular density, this is also happening in the gut vasculature upon AHR BEC KO?

6. The authors state that AHR signaling ensures intestinal ECs quiescence, however work from various labs has shown that intestinal vessels are VEGF-dependent and proliferate in adult animals (Kamba 2006, Latmani 2015, Hong 2020 etc) . Moreover, the EdU incorporation assay (Fig 3d) shows WT animals display EC EdU incorporation. Therefore, the claim that intestinal ECs are quiescent is not supported. Are AHR KO ECs more apt to proliferation upon VEGF signaling? Do they still proliferate with VEGF depletion/blockade?

7. The authors report gene expression changes in AHR KO ECs related to leukocyte recruitment. Over the 14 days of deletion is there a change in the immune cells recruited to the intestine and does this result in altered immune to endothelial cell signaling? Similarly, does AHR BEC KO alter the epithelial cells?

8. ED Fig 6. Labeling makes it unclear if BEC or LECs are analyzed. But beyond that, there are no controls without LPS to assess the expression of inflammation markers. It could be the significant differences observed between WT and AHR BEC KO mice are relatively small compared to changes in these markers' expression between untreated and LPS-treated mice, questioning the biological relevance of these differences.

9. Given the lack of data showing deletion efficiency of endothelial Ahr, potential confounding factors mentioned above and a large difference in survival rates between infected germline and AHR BEC KO mice, the conclusion that endothelial AHR plays a role in host resistance to enteric infections requires more evidence.

10. In the discussion the authors cite several papers reporting AHR ligands repress endothelial cell proliferation and negatively regulates type 1 IFN and NFkB signaling in other contexts. The authors do show that AHR represses EC responses to LPS, but in general I find this mechanism lacks novelty.

Minor comments

Fig 1c- It is unclear to what the authors are referring when they mention that the clustering matched well to inferred biology. What is the inferred biology? Further explanation would help.

ED Fig 3e- Cyp1a1 antibody staining does not look specific.

The statement in the introduction saying the scRNAseq data are "... revealing unparalleled cellular heterogeneity" should be rephrased.

The discussion item about AHR preventing endoMT is quite premature in my opinion. The evidence for this is based solely on bioinformatic data without follow up experiments.

Referee #2 (Remarks to the Author):

Here, Wiggins et al present a characterization of the role of the aryl hydrocarbon receptor (AHR) pathway in the intestine and its role in immune responses. In this study, the authors investigate the role of endothelial cells in response to infection. As a result, they identify the AHR pathway in the endothelial cells as a critical mediator of the inflammatory response. The study highlights a critical role of the AHR pathway in the defense against pathogens and the maintenance of homeostasis.

Overall, the manuscript is well written and the logic is easy to follow, even for a computational biologist such as this reviewer. Judging the computational analyses, everything appears to have been carried out following state of the art procedures and I see no reason to doubt any of the findings based on how the data was processed.

I have the following major concerns

It is very hard to judge the differences in Ahr and Cyp expression between different clusters just by squinting at a UMAP in Fig 2a. The authors should summarize the results more quantitatively and also provide a suitable statistical test to demonstrate which cell types have significantly different expression levels.

I also suggest writing the significant adjusted p value in figure 2a, as the manuscript reports a "striking" difference, and it is unclear how this relates to a significant difference. Also, Figure 2d could be improved with additional information of p values, at least providing the highest (the 20th) adjusted p value of the 20 top DEGs sorted by pvalue (if that is how the genes were sorted?).

Extended Figure 3a suggests that there are more or less equal number of cells for condition in each clusters. The authors should calculate whether there are bias in abundance of condition (vehicle vs ligand) that summarizes these results. Is there abundance bias for any identified clusters, more than the expected ratio from the 2 sample vs 1 sample comparison?

I am not a big fan about claims regarding "borderline significant" results (fig 3g). The authors should either tone down/remove this claim or carry out another round of experiments in the hope that a higher n will ensure significance.

I also have the following minor concerns:

I find fig 1b a bit strange - why use the % as x-value and then also write out the number next to each bar. It seems like redundant information to me, so why not take out the bars and present it as a table?

The extended Figure 1a that describes the design the of the sequencing data does not make it clear that the demultiplexed data contains 2 mice with AHR ligand exposed cells vs 1 mouse with control (vehicle). Authors should reflect this data design clearly (as described in the methods).

I am assuming that there is something wrong in how the p-values are reported in Fig 2e, otherwise there are some absurdly high values (around 10^3 by my judgement). I assume that the authors have taken the absolute value of the log p-value and then given it a sign depending on the direction of the regulation?

Overall, the resolution of the figures is poor and with the small fonts it is often hard to read the text. The authors should improve all of the figures.

Referee #3 (Remarks to the Author):

In this manuscript, entitled "Endothelial sensing of AHR ligands regulates intestinal homeostasis", by Benjamin G. Wiggins et al, the authors undertake comprehensive and very extensive profiling of functionally specialized endothelial cells (ECs). The authors report that the aryl hydrocarbon receptor (AHR) acts as a critical node for EC-sensing of dietary metabolites. The authors first establish a comprehensive single-cell endothelial atlas of the mouse small intestine, uncovering the cellular complexity and functional heterogeneity of blood and lymphatic ECs. Analyses of AHR-mediated responses revealed tissue-protective transcriptional signatures and regulatory networks promoting cellular quiescence and vascular normalcy at a steady state. In contrast, endothelial AHR-deficient mice result in dysregulated inflammatory responses (following stimulation) as well as initiation of proliferative and angiogenic pathways, disrupting organ homeostasis.

They further go on to show that endothelial sensing of dietary AHR ligands is required for optimal protection against enteric infection. The authors close the report by undertaking an analysis of human ECs and report that AHR signaling promotes quiescence and restrained activation by inflammatory mediators.

General comments.

1. It is an elegant and very detailed report thus establishing that the AhR signaling pathway in EC does play a central role in intestinal homeostasis following acute ablation using tamoxifen-induced depletion of AhR. While it has been suggested to be a key regulator in EC, this report conclusively confirms that AhR plays a role in EC-linked homeostasis, repair, and remodeling.
2. The figures are well-presented and support the result section.
3. AhR in intestinal homeostasis has been studied extensively. The novelty in this paper lies in the EC profiling which has not been reported before.
4. What I find surprising in this generous result portfolio, is the validation experiment. Throughout all the experiments, the authors have been using male mice. Then, all of sudden, the authors use a mix of male and female mice. Why introduce the females and confuse the study? Especially since it's known that female estrogen production influence mucus viscosity and that mucin level and thickness change following the menstruation cycle which in mice is about 4 days. Please include a complete set of male mice experiments $n = 8-10$ mice per experiment and at least two independent experiments. The authors may consider omitting the female data since the other figures don't carry data from female mice.
5. Suggestion I: The result section may be improved by providing some more pictures of immunohistochemistry and staining for angiogenesis and lymphatic vessel changes following the deletion of AhR by tamoxifen and by ligand stimulation in mutant and WT mice (of course longer ligand exposing time). The gene profiling is very convincing but also displaying the architecture of the intestinal landscape in the different conditions would strengthen the manuscript. Such staining would help better to understand the infection experiment and the observed organ disruption. Are there changes in the overall barrier integrity and villus structure that pre-set the increased sensitivity to Yersinia infection in the AhR EC mutant mice?
6. Suggestion II: the manuscript could also be improved by providing an additional functional physiological experiment for example using a labeled lipid tracer or a vascular tracer experiment. As such, the possible differences in functional activity between WT and EC AhR mutant mice may be better displayed.

Author Rebuttals to Initial Comments:

Endothelial sensing of AHR ligands regulates intestinal homeostasis (2022-03-03578A): Reviewer comments point-by-point replies

We are greatly appreciative of the reviewers' comments and feedback on our work and have now addressed these in detail. Below we have included point by point replies to reviewer comments, detailing the revisions we have made to our manuscript in blue.

Within the manuscript, we have used the following to show changes from the original:

- Blue underlined text for direct changes as a result of the reviewer comments
- Black underlined text for indirect and minor changes (figure number changes, additional methods, minor improvements to the prose etc.)

Referee #1 (Remarks to the Author):

2022-03-03578A

In this work from Wiggins et al., the authors analyze the small intestinal vasculature and endothelial cell responses to AHR ligands. They first performed scRNAseq on small intestinal blood and lymphatic endothelial cells after administration of an AHR ligand and performed in-depth bioinformatic analysis on control ECs. They found a widespread cellular response to the AHR ligand in multiple endothelial cell subsets in the gut and other organs. They then analyzed endothelial gene expression, proliferation and inflammation markers in AHR EC KO mice following injection of an AHR ligand and found increased proliferation and inflammation in AHR KO ECs. They also analyzed AHR KO and AHR BEC KO mice after infection with the pathogen Yptb and found reduced mouse survival. Finally, *in vitro*, they found AHR ligands restrict proliferation and inflammatory responses.

This manuscript starts with very interesting, high quality scRNAseq data of intestinal BECs and LECs from control mice combined with thorough bioinformatic analysis. There are distinctive, potentially gut-specific, BEC and LEC subsets with interesting properties that could reveal novel endothelial-mediated gut functions. I found the subset-specific transcription factor expression especially fascinating. Unfortunately, however, most of the rest of the manuscript focuses on endothelial cell responses to AHR ligands when AHR expression is not endothelial cell-specific, gut-specific nor seemingly specific to any of the clusters identified by single cell analysis. Endothelial cells present a homogenized response to AHR ligand treatment and conclusions drawn about direct endothelial cell-intrinsic ligand responses could be confounded by secondary effects mediated by AHR ligands on gut immune and epithelial cells. Moreover, the endothelial cell responses to AHR signaling identified by the authors in the gut (reduced proliferation and inflammation) are not novel and have been reported in other contexts. The authors also fail to incorporate their results into emerging themes of intestinal-specific endothelial cell responses to inflammation (recent papers on TAK1 and Casp8 EC KOs) or Vegfa signaling. Lastly, conclusions are often reached without appropriate controls and/or verification of bioinformatic data through FACS or imaging data. With the above concerns I believe this manuscript is better suited to a more specialized journal

We thank the reviewer for highlighting the novelty and high-quality of our scRNAseq dataset and associated analysis pipeline. Indeed, our manuscript reveals novel SI EC populations, functional heterogeneity and SI EC transcriptional networks highly relevant to the field. The reviewer correctly points out that AHR-responsiveness is persevered across EC subtypes in the gut and other organs distal to the gut (as is also verified by the accompanying manuscript by Major et al). To our knowledge this is the first demonstration of AHR responsiveness across endothelia and vascular beds of various organs and establishes AHR as a key environmental sensor in ECs. As expected, our detailed analysis of SI ECs raises several interesting questions about EC heterogeneity, and we hope to delve deeper into this novel area of research in separate studies. The reviewer correctly points out that anti-proliferative and anti-inflammatory effects of AHR ligand sensing have been observed in other cell types, but we are not aware of any other publications demonstrating this in SI ECs. We thank the reviewer for pointing out the relevant literature, which we have incorporated in our discussion. Lastly, we thank the reviewer for acknowledging that our manuscript is suitable for publication and hope that with the additional *in vivo* verification by imaging and flow cytometry; comprehensive profiling of gut endothelial, immune and epithelial cells at steady state; additional evidence for a role of AHR in Yersinia infection and other controls requested by the reviewer detailed below; will merit publication in Nature.

Major comments

1. Single cell data of steady state ECs are of excellent quality and show distinct clusters. Unfortunately, there is no verification of cluster marker genes through other means like FACS or imaging.

First, we would like to thank the reviewer for the feedback and helpful comments. We agree in principle with the reviewer that further validation of our newly identified clusters by flow cytometry and imaging approaches would be very useful for the field to distinguish, study and locate these different populations within the gut. Unfortunately, such detailed validation is currently limited by availability of appropriate reagents (antibodies) to cluster specific markers. We have undertaken extensive flow cytometry surface marker profiling of gut ECs with a view to building a panel that could be used as a starting point for further detailed experiments (e.g. CITE-seq) to characterise gut ECs further. As set out in detail in reviewer Figure 1, while certain populations (PCV, LEC IFN) mapped well to scRNAseq on the basis of surface marker expression profiles, there was only partial evidence for gene-protein overlap with others. We hypothesise this was due in part to the limited scope of antibodies available compared to marker genes identified, and the known imperfect direct correlation between mRNA and protein expression within a cell¹. Additionally, the requirement for cluster-specific markers to be exclusively expressed by a single cluster restricts the potentially useful gene markers further as most markers are commonly expressed at various levels of across related populations. As such, development of suitable antibody panels to distinguish EC clusters identified in our study would require highly customised, high-dimensional cytometry approaches. This said, the work we performed here could help others study enteric LEC IFN through our flow cytometric characterisation and validation and may set the stage for further in-depth protein level characterisation of other subpopulations.

As the reviewer suggested we also performed immunofluorescent staining, examining key transcription factor determinants highlighted by our scRNAseq data. Here we did not observe any specific spatial distribution or subpopulation of gut ECs expressing 3 key TFs. Together these outcomes demonstrate that a) complete validation of EC clusters to the level of single cell resolution likely requires the generation of new monoclonal antibodies, and b) may require approaches such as CITE-seq to further characterise these data, both experimental undertakings we believe are beyond the scope of this manuscript. As noted by the reviewer, our characterisation of SI EC clusters goes well beyond cluster markers and is supported by in depth analysis of transcriptional and functional heterogeneity displayed by each of the clusters.

2. The authors choose to analyze gut ECs after AHR ligand treatment and observe a mostly homogeneous endothelial response. This is a shame because their data contain interesting observations that could lead to discoveries about gut- or organ-specific endothelial Ahr signaling. Eg, Ahr is expressed in a pan-endothelial fashion in the SI ECs, however, the AHR signaling response gene Cyp1a1 is expressed in only one LEC cluster and Cap 1 and PCV BECs. Identification of the location of these cells by imaging with the Cyp1a1 reporter mouse at steady state could identify the cellular route whereby AHR ligands enter the bloodstream. Similarly, Ahr expression is higher in LECs than BECs on average (ED Fig 3b), however the steady state expression of Cyp1a1 is low, there is a LEC-specific mechanism to suppress cellular responses to dietary AHR ligands? Along the same lines, ED Fig 3h shows that after AHR ligand treatment BECs and LECs express Cyp1a1 in all organs except the spleen and colon, respectively. Are there organ-specific endothelial mechanisms restricting AHR signaling?

The reviewer correctly points out that AHR responsiveness is a shared feature across all SI EC subtypes. Importantly there is large heterogeneity in the response to AHR activation across the clusters (Fig.2e-f, Extended Data Fig.4a-f) indicative of distinct functional EC subtypes being able to sense AHR ligands and integrate such signals in a cell-type/cell-state specific manner. We agree with the reviewer that identifying the location of gut EC populations with constitutive AHR activation may be beneficial to help explain the different sensitivities to AHR ligands we found between our clusters. As such, we set out to identify Cyp1a1-expressing cells through imaging approaches as the reviewer suggests (Reviewer Figure 2). While clearly less Cyp1a1-expression is seen in vehicle-treated mice compared to those treated with AHR ligand 3-MC, Cyp1a1+ cells can be seen scattered throughout the villi and submucosal vasculature, and throughout lacteals (Reviewer Figure 2, compared to Figure 2d). This

result suggests that at least some enteric EC phenotypes (cap 1, LEC 1, LEC IFN) identified by scRNAseq may not be positionally distinct, and instead scattered within the gut vascular bed. However, we can only be tentative with these conclusions based on this data as a number of caveats are associated with this approach. First, such data will only provide a snapshot of AHR ligand sensing at the particular time point analysed and will be strongly influenced by fasting/feeding cycles, metabolic activity of the host and intestinal peristalsis, and CYP1A1 enzymatic activity – to list only a few factors contributing to dietary AHR ligand availability in the absence of AHR pathway synchronisation through exogenous ligand exposure (injection of AHR ligand or AHR ligand-rich diet). Furthermore, the Cyp1a1-reporter model tracks lifelong history of AHR ligand exposure rather than current ongoing signalling as revealed by the scRNAseq, suggesting caution when directly comparing the two approaches. Finally, the Cyp1a1-reporter model is less sensitive to AHR ligand stimulation when compared to scRNA sequencing, meaning we could miss out on gut ECs that express Cyp1a1 to a lower level. This is why throughout the manuscript we have primarily used the Cyp1a1-reporter to assess responses to AHR ligands compared to vehicle controls to reveal global response potential to AHR ligands across endothelial subtypes.

Finally, the gene and pathway analysis provided in the paper shows LEC have different transcriptional responses than BEC following AHR ligand stimulation, and interestingly, on cell-type specific suppression of AHR responses, it seems AHR negative feedback gene *Tiparp* is expressed higher in LEC (and PCV) at steady state than BEC (Reviewer Figure 3). However, while we agree that studying organ-specific endothelial mechanisms restricting AHR signalling is important, we feel this is beyond the scope of this study that is focussed on the small intestinal vasculature at the single-cell level. This said, the accompanying manuscript by Major et al. investigates AHR sensing in lung endothelial cells, providing a point of comparison to our work.

3. Since various immune cells and intestinal epithelial cells express AHR (PMID: 33742166, Shah 2022) and show responses to AHR ligands, with the AHR ligand injection strategy used by the authors it is difficult to assess how much of EC gene expression changes are caused by direct EC AHR signaling or secondary to signaling from other intestinal cell AHR signaling.

We used an acute systemic (3h) ligand stimulus to focus predominantly on direct transcriptional changes. We cannot formally exclude indirect contributions through AHR activation in other cell types. However, several observations support the notion that observed transcriptional responses are EC intrinsic. First, the AHR response gene Cyp1a1 is the most highly induced gene in all EC subtypes (except for LEC2b where it is 3rd most highly induced; Fig.2e, Extended Data Fig.4a) demonstrating that all ECs directly sense and respond to AHR signals. Second, we verify our findings by using endothelial specific AHR knockout mice (Figure 3, Extended Data Figures 5 - 8) that match the mechanisms (proliferation, angiogenesis, and inflammation pathways) indicated by the scRNAseq data. Third, we demonstrate direct effects of AHR ligands in our in vitro experiments on HUVECs, showing similar anti-proliferative and anti-inflammatory effects (Fig.4; Extended Data Figure 9).

4. There are no data showing the deletion efficiency of Ahr in the Ahr fl/fl, Cdh5 CreERT2 mice. The Cre deleter strain used by the authors displays organ-specific deletion efficiency in adult mice.

We apologise for this oversight in missing this data out in initial submission. We have since included this, showing high deletion efficiency of 92% on average for BEC and LEC (Extended Data Figure 5b)

5. In relation to point 2 and 4, if deletion efficiency is good in these mice, why not perform experiments at steady state to assess the effect of EC Ahr deletion on gut parameters such as inflammation, endothelial proliferation, vessel density, immune cell infiltration, epithelial cells, etc? Ahr germline KO mice display organ-specific decreases in vascular density, this is also happening in the gut vasculature upon AHR BEC KO?

We thank the reviewer for these suggestions and have substantially expanded our analysis to additional cell types within the tissue in the context of endothelial AHR-deficiency at steady state. In addition to the steady-state endothelial proliferation data we provide in the manuscript (Fig.3d-f), we provide data on immune cell composition (Extended Data Fig.6a-b), epithelial activation (Extended Data Fig.6c), and vascular density (Extended Data Fig.6f) as the reviewer suggests. Additionally, we investigated whole tissue changes through bulk RNA sequencing of RNA isolated from whole tissue to gain a more holistic view of how endothelial AHR deficiency impacts on the tissue at steady state, finding minimal

transcriptomic differences at the whole tissue level. (Extended Data Figure 6d-e). Finally, we would like to thank the reviewer for highlighting the AHR-linked developmental vascular defects in we mentioned in our introduction, as it prompted us to clarify we are dissecting this response in adult vasculature throughout the manuscript.

6. The authors state that AHR signaling ensures intestinal ECs quiescence, however work from various labs has shown that intestinal vessels are VEGF-dependent and proliferate in adult animals (Kamba 2006, Latmani 2015, Hong 2020 etc) . Moreover, the EdU incorporation assay (Fig 3d) shows WT animals display EC EdU incorporation. Therefore, the claim that intestinal ECs are quiescent is not supported. Are AHR KO ECs more apt to proliferation upon VEGF signaling? Do they still proliferate with VEGF depletion/blockade?

We are very grateful for this interesting suggestion, which has directly led to a major improvement in our manuscript. First, we take the reviewer's point while most endothelial cells are quiescent *in vivo*, there are notable exceptions², and so we have modified the text in question to reflect this, making the point that gut EC proliferation is at a low level rather than absent. Correspondingly, preliminary work assessing proliferation *in vivo* by both EdU incorporation and Ki-67 staining demonstrated that assays shorter than the 2-week EdU administration protocol used in the manuscript fail to capture meaningful levels of proliferation (Reviewer Figure 4). Therefore, we reasoned that additional VEGF depletion/blockade will run the risk of decreasing the already low level of proliferation beyond the limit of detection. On the other hand, when administering exogenous VEGF, the enhanced EC proliferation in EC^{ΔAhr} mice was maintained (Fig.3e), providing further evidence of AHR as an important proliferative rheostat even in the presence of elevated angiogenic signals from VEGFA. Finally, we also ensured we discussed VEGF dependence in the small intestine in the text.

7. The authors report gene expression changes in AHR KO ECs related to leukocyte recruitment. Over the 14 days of deletion is there a change in the immune cells recruited to the intestine and does this result in altered immune to endothelial cell signaling? Similarly, does AHE BEC KO alter the epithelial cells?

This is an interesting point. We performed this experiment as the reviewer suggested, showing no significant differences in immune recruitment at the mid-point of the normal (5x) tamoxifen dosing regimen (Extended Data Figure 5b), nor following the full tamoxifen course (Extended Data Figure 6a-b). However, these differences do become apparent in the context of enteric infection, with increased eosinophil, dendritic cells, NK cells, and $\gamma\delta$ T cells in the SI lamina propria of EC^{ΔAhr} mice (Fig.3j; Extended Data Figure 8d). In addition, we checked epithelial cell activation (Extended Data Figure 6c) as well as whole-tissue transcriptomic changes in EC^{ΔAhr} mice (Extended Data Fig.6d-e), showing no clear differences to control mice.

8. ED Fig 6. Labeling makes it unclear if BEC or LECs are analyzed. But beyond that, there are no controls without LPS to assess the expression of inflammation markers. It could be the significant differences observed between WT and AHR BEC KO mice are relatively small compared to changes in these markers' expression between untreated and LPS-treated mice, questioning the biological relevance of these differences.

We initially failed to consider inflammatory marker changes at steady state as we reasoned these would not be impacted enough to observe a difference. However, we overlooked the important biological distinction the reviewer makes between AHR-linked expression changes, and the expression changes caused by the inflammatory agent (LPS) itself, so we are grateful to the reviewer for suggesting this. We therefore since expanded our endothelial cell panels, and conducted an experiment assessing marker changes within EC^{ΔAhr} mice at both steady state, and following LPS treatment (Extended Data Figure 7). We were able to make distinctions between markers of inflammatory activation that also hold true in enteric ECs (ICAM-1, VCAM-1, MHC-II, PD-L1 for BEC; ICAM-1, MHC-II and PD-L1 for LEC); and additional markers like CD24 and CD36 that are more dependent on AHR than on the inflammatory milieu. Intriguingly, as we point out in the results text, for some inflammatory markers (VCAM-1 & ICAM-1 for BEC; ICAM-1 and MHC-II for LEC), the level of upregulation is similar upon AHR deletion as with

LPS induction. This highlights the potency of AHR signalling in dampening inflammation in gut ECs. Another interesting finding that came out of the reviewer's suggestion to include steady state profiling in the manuscript, is that LPS treatment was required to reveal the full potential of differences between WT and KO endothelia (8/20 changes at steady state vs. 14/20 differences with LPS).

9. Given the lack of data showing deletion efficiency of endothelial Ahr, potential confounding factors mentioned above and a large difference in survival rates between infected germline and AHR BEC KO mice, the conclusion that endothelial AHR plays a role in host resistance to enteric infections requires more evidence.

As well as the data on *Ahr* deletion efficiency mentioned above (point 4; Extended Data Figure 5a), we have supplemented our data on the outcome of infection in endothelial AHR-deficient mice: showing statistically survival following infection (Fig.3h), bacterial burden (Fig.3i), and immune composition (Fig.3j) to support our conclusions. With these improvements and increased cohort size we now demonstrate a significant decrease in survival in female $EC^{\Delta Ahr}$ mice, while the differences in male mice did not reach significance. We believe it highly likely that different AHR-responsive cell types in the gut contribute to defence against enteric infection. This is well established for *Citrobacter rodentium* infection where AHR sensing in both enteric immune cells and intestinal epithelial cells are important for optimum clearance³⁻⁵. Moreover, in *Yersinia enterocolitica* infection, the presence of AHR-dependent ILC3s are vital in host defence, reducing bacterial burden, and facilitating survival⁶. Thus, it is highly likely that ILC3s in particular are important in the clearance of *Yptb* also, explaining the more severe phenotype in germline AHR-deficient mice (Extended Data Fig 8a-c). We are proposing here that the EC contribution to this protection lies more in a disease tolerance process, as $EC^{\Delta Ahr}$ mice show a dysregulated immune cell compartment without altered bacterial burden in response to *Yptb* infection (Fig.3i-j). This is in keeping with our scRNAseq/RNAseq transcriptomic data that highlights immune recruitment pathways; and also fits with similar conclusions drawn by the accompanying manuscript by Major et al. who show AHR signalling in lung ECs is important for disease tolerance following influenza infection.

10. In the discussion the authors cite several papers reporting AHR ligands repress endothelial cell proliferation and negatively regulates type 1 IFN and NFkB signaling in other contexts. The authors do show that AHR represses EC responses to LPS, but in general I find this mechanism lacks novelty.

We thank the reviewer for voicing this concern. Alongside identifying novel endothelial populations in the gut and uncovering the transcriptional heterogeneity of these EC populations; our study overlays responses to AHR ligands upon this complexity, demonstrating roles for AHR in endothelial quiescence, inflammatory restraint, and vascular normalcy for the first time. We show this through both profiling responses to AHR ligands in single-cell sequencing data, and by examining endothelial AHR-deficient mice in homeostasis and in response to separate inflammatory and infectious challenges. Although the AHR response to dietary and microbial microenvironmental factors is known to promote anti-inflammatory and tissue-protective in other cell types, to our knowledge our study provides the first *in vivo* description of the transcriptional programs induced by AHR stimulation in gut blood and lymphatic endothelial cells. In our opinion, this helps place endothelial AHR signalling within the far better understood roles of AHR in gut epithelial and immune cells in order to understand the impact of AHR on the gut holistically. Furthermore, given the links between vascular dysfunction and tissue pathology, we believe our findings on endothelial quiescence have important implications in EC biology, organ homeostasis and establish endothelial AHR signalling as a potential therapeutic target.

Minor comments

Fig 1c- It is unclear to what the authors are referring when they mention that the clustering matched well to inferred biology. What is the inferred biology? Further explanation would help.

We were discussing the package Pagoda2 aspects and pathways that infer the biological role of each cluster by this comment but agree this should be clearer in the text. We have now simplified this in the text, instead saying Pagoda2 "allowed us to test how well different endothelial functions mapped onto our clustering" in the first results section.

ED Fig 3e- Cyp1a1 antibody staining does not look specific.

We have now highlighted that although there is some background autofluorescence seen with this channel/antibody, the vascular staining is still extremely clear compared to background. To illustrate this further we have added representative staining examples at 20x magnification (Extended Data Figure 3f) that is easier to distinguish than at the 10x examples previously given.

The statement in the introduction saying the scRNAseq data are "... revealing unparalleled cellular heterogeneity" should be rephrased.

We thank the reviewer for this comment and have rephrased this.

The discussion item about AHR preventing endoMT is quite premature in my opinion. The evidence for this is based solely on bioinformatic data without follow up experiments.

We take the reviewer's point and have now removed this section from the discussion.

Referee #2 (Remarks to the Author):

Here, Wiggins et al present a characterization of the role of the aryl hydrocarbon receptor (AHR) pathway in the intestine and its role in immune responses. In this study, the authors investigate the role of endothelial cells in response to infection. As a result, they identify the AHR pathway in the endothelial cells as a critical mediator of the inflammatory response. The study highlights a critical role of the AHR pathway in the defense against pathogens and the maintenance of homeostasis.

Overall, the manuscript is well written and the logic is easy to follow, even for a computational biologist such as this reviewer. Judging the computational analyses, everything appears to have been carried out following state of the art procedures and I see no reason to doubt any of the findings based on how the data was processed.

We are greatly appreciative to the reviewer for their overwhelmingly positive feedback on our study. We agree with the reviewer that our work identifies a critical role for the endothelial cell AHR pathway in the balance between homeostasis and inflammation, and responses to infection in the small intestine. We would like to thank the reviewer again for praising the manuscript's writing style, logic and clarity and are appreciative that the reviewer highlights in particular the state-of-the-art quality of our computational analyses, an area we dedicated significant time and resources to in order to reach robust and meaningful conclusions.

I have the following major concerns

1) It is very hard to judge the differences in Ahr and Cyp expression between different clusters just by squinting at a UMAP in Fig 2a. The authors should summarize the results more quantitatively and also provide a suitable statistical test to demonstrate which cell types have significantly different expression levels.

We would like to thank the reviewer for this feedback, and agree the presentation style and quantification should be improved for this figure panel. We have done exactly as the reviewer suggested, moving the more easily visualised split violin plots into the main figure (previously Extended Data Figure 3b, now Figure 2b) showing *Ahr*, *Cyp1a1*, and *Cyp1b1* expression. We have also highlighted the statistical differences a) between conditions (vehicle vs FICZ) for each cluster, and b) between clusters within Figure 3b, extracted from gene lists generated from the `FindConservedMarkers()` and `FindMarkers()` functions within the Seurat package. Conserved marker, and differentially expressed gene lists can be found in supplementary tables 2, and 6 respectively.

2) I also suggest writing the significant adjusted p value in figure 2a, as the manuscript reports a "striking" difference, and it is unclear how this relates to a significant difference. Also, Figure 2d could be improved with additional information of p values, at least providing the highest (the 20th) adjusted

p value of the 20 top DEGs sorted by pvalue (if that iss how the genes were sorted?).

We are grateful for these suggestions. In addition to reporting on statistical significance in Fig 2b (previously Figure 2a), we have provided adjusted p value ranges for all the DE gene comparisons shown in Figures 2e and Extended Data Figure 4a. The reviewer is correct in their assumption that these were top 20 DEGs sorted by adjusted p value.

3) Extended Figure 3a suggests that there are more or less equal number of cells for condition in each clusters. The authors should calculate whether there are bias in abundance of condition (vehicle vs ligand) that summarizes these results. Is there abundance bias for any identified clusters, more than the expected ratio from the 2 sample vs 1 sample comparison?

We apologise for the confusion and thank the reviewer for raising this point so we can address this. Although 3 samples were used here (2 FICZ treated, 1 vehicle treated), the 2 FICZ treated samples were from the exact same pool of cells, and importantly were combined initially into one sample that was then compared with vehicle in Extended Data Fig 3a in order to obtain similar numbers of cells per condition. This does not reflect different numbers of mice used or different representation of the data. We have clarified this point in the methodology text also ('*scRNAseq*' and '*Data preprocessing and QC*' subsections of 'single-cell RNA sequencing' section). We have performed extensive comparisons of 2-sample (1 vehicle sample vs 1 FICZ sample) compared to 3-sample (1 vehicle vs 2 FICZ) data, finding these two data sets are highly concordant, and have provided a summary of this work in Reviewer Figure 5.

I am not a big fan about claims regarding "borderline significant" results (fig 3g). The authors should either tone down/remove this claim or carry out another round of experiments in the hope that a higher n will ensure significance.

We thank the reviewer for this comment and have focussed a lot of attention on addressing these concerns with the yersinia infection model. First we have performed further rounds of experiments investigating at the survival of $EC^{\Delta Ahr}$ mice following *Yptb* infection (Fig.3h – 2 cohorts for male, and 2 additional cohort for female mice), analysing bacterial dissemination to different organs (Fig.3i), and assessing immune recruitment during infection (Fig.3j). We demonstrate statistical significance with female $EC^{\Delta Ahr}$ mice and have removed all claims of borderline significance with male mice, tempering our results to describe a partial effect on survival in $EC^{\Delta Ahr}$ mice, but a definite effect on the immune infiltrate, suggestive of an endothelial-specific disease tolerance mechanism, in line with the mechanism proposed in the accompanying manuscript (Major et al.).

I also have the following minor concerns:

I find fig 1b a bit strange - why use the % as x-value and then also write out the number next to each bar. It seems like redundant information to me, so why not take out the bars and present it as a table?

We thank the reviewer for pointing this out and agree having the % written out and as bar width is redundant. We have now removed the written percentages and maintained the cell numbers next to each bar in Fig.1b to make it more intuitive and simpler to read.

The extended Figure 1a that describes the design the of the sequencing data does not make it clear that the demultiplexed data contains 2 mice with AHR ligand exposed cells vs 1 mouse with control (vehicle). Authors should reflect this data design clearly (as described in the methods).

We have clarified our experimental design in the methods text as we were not clear enough here – the data contains the same number of mice (3 mice ligand treated vs 3 mice vehicle), the extra sample comes from another lane that was sequenced from the same sample in the case of ligand-treated condition, in order to obtain similar numbers of cells in each condition.

I am assuming that there is something wrong in how the p-values are reported in Fig 2e, otherwise there are some absurdly high values (around 10^3 by my judgement). I assume that the authors have taken the absolute value of the log p-value and then given it a sign depending on the direction of the regulation?

A very good point, and the reviewer's assumption is correct. The negative enrichment values displayed do not make sense – we will have removed the negative signs for these.

Overall, the resolution of the figures is poor and with the small fonts it is often hard to read the text. The authors should improve all of the figures.

Thank you to the reviewer for highlighting this. We have adjusted the export settings and checked all re-exported images carefully for better resolution.

Referee #3 (Remarks to the Author):

In this manuscript, entitled “Endothelial sensing of AHR ligands regulates intestinal homeostasis”, by Benjamin G. Wiggins et al, the authors undertake comprehensive and very extensive profiling of functionally specialized endothelial cells (ECs). The authors report that the aryl hydrocarbon receptor (AHR) acts as a critical node for EC-sensing of dietary metabolites.

The authors first establish a comprehensive single-cell endothelial atlas of the mouse small intestine, uncovering the cellular complexity and functional heterogeneity of blood and lymphatic ECs. Analyses of AHR-mediated responses revealed tissue-protective transcriptional signatures and regulatory networks promoting cellular quiescence and vascular normalcy at a steady state.

In contrast, endothelial AHR-deficient mice result in dysregulated inflammatory responses (following stimulation) as well as initiation of proliferative and angiogenic pathways, disrupting organ homeostasis.

They further go on to show that endothelial sensing of dietary AHR ligands is required for optimal protection against enteric infection. The authors close the report by undertaking an analysis of human ECs and report that AHR signaling promotes quiescence and restrained activation by inflammatory mediators.

We are again extremely thankful to the reviewer for their positive feedback on our study, highlighting our comprehensive and very extensive profiling of specialised gut ECs using single cell RNA sequencing where we uncover substantial cellular complexity and heterogeneity of blood and lymphatic ECs using three complimentary computational approaches (cluster marker gene enrichment, geneset overdispersion analysis, and transcription factor regulon analysis – Extended Data Figure 2a, Figure 1c, and Figure 1d-e respectively). Our data pinpoint the AHR as a key node in endothelial cells - controlling endothelial cell quiescence and promoting tissue homeostasis. As the reviewer wrote, our data highlights the AHR promotes tissue-protective transcriptional signatures in endothelial cells – promoting vascular normalcy at steady state; while our analyses of endothelial cell AHR deficiency demonstrated dysregulated inflammatory responses and initiation of angiogenic pathways that disrupt organ homeostasis. We thank the reviewer for also highlighting our findings that endothelial AHR-mediated sensing of dietary ligands is required for optimal protection against enteric infection, and that AHR signalling promotes quiescence and restrained responses to inflammation in human ECs as well.

General comments.

1. It is an elegant and very detailed report thus establishing that the AhR signaling pathway in EC does play a central role in intestinal homeostasis following acute ablation using tamoxifen-induced depletion of AhR. While it has been suggested to be a key regulator in EC, this report conclusively confirms that AhR plays a role in EC-linked homeostasis, repair, and remodeling.

We are extremely grateful that the reviewer deems our study elegant and detailed, and for highlighting the importance of our study in the wider field in establishing the central role for AHR in endothelial cells in intestinal homeostasis, complimenting previous work on gut epithelial and immune populations. We thank the reviewers for their observation that our manuscript conclusively confirms that AHR plays a role in EC linked homeostasis, repair, and remodelling.

2. The figures are well-presented and support the result section.

We thank the reviewer for their positive feedback on our figure presentation and interpretation, this is much appreciated.

3. AhR in intestinal homeostasis has been studied extensively. The novelty in this paper lies in the EC profiling which has not been reported before.

We thank the reviewer for their appreciation of the novelty of our work on extensive profiling of gut blood and lymphatic EC complexity by single cell RNA sequencing.

4. What I find surprising in this generous result portfolio, is the validation experiment. Throughout all the experiments, the authors have been using male mice. Then, all of sudden, the authors use a mix of male and female mice. Why introduce the females and confuse the study? Especially since it's known that female estrogen production influence mucus viscosity and that mucin level and thickness change following the menstruation cycle which in mice is about 4 days. Please include a complete set of male mice experiments $n = 8-10$ mice per experiment and at least two independent experiments. The authors may consider omitting the female data since the other figures don't carry data from female mice.

Firstly, thank you again to the reviewer for highlighting the extent of work that went into this study. We have repeated the infections on male mice as the reviewer suggested. We also elected to repeat the experiment 2 further times in female mice to total 3 independent cohorts for this group as well, demonstrating a significant decrease in survival in $EC^{\Delta Ahr}$ mice. We show these results alongside the male mice in Fig.3h.

5. Suggestion I: The result section may be improved by providing some more pictures of immunohistochemistry and staining for angiogenesis and lymphatic vessel changes following the deletion of AhR by tamoxifen and by ligand stimulation in mutant and WT mice (of course longer ligand exposing time). The gene profiling is very convincing but also displaying the architecture of the intestinal landscape in the different conditions would strengthen the manuscript. Such staining would help better to understand the infection experiment and the observed organ disruption. Are there changes in the overall barrier integrity and villus structure that pre-set the increased sensitivity to *Yersinia* infection in the AhR EC mutant mice?

We thank the reviewer for their excellent suggestion. We have improved our results section by performing whole-mount gut imaging on gut sections from our EC AHR mutant and WT mice 1 week after AHR ligand (3-MC) administration. We subsequently examined changes to villi vasculature density and branching (Extended Data Figure 6f), lacteal length, finding no difference in these parameters (Extended Data Figure 6f); and ESM1⁺ tip cell staining, finding an enrichment (Fig.3f) to answer these questions. We also assessed whole-tissue transcriptomic changes with RNAseq, finding very few gene changes (12 genes changed total) that likely would have revealed a barrier disruption phenotype at steady state if this existed (Extended Data Figure 6d). We also showed no changes in immune infiltration at steady state (Extended Data Figure 6b), nor alterations in epithelial cell activation (Extended Data Figure 6c). Importantly, having done more work on the consequences of *Yptb* infection in $EC^{\Delta Ahr}$ mice in the review process, we now propose endothelial AHR is principally acting to limit cellular inflammation in infection (Figure 3j), rather than because of changes in barrier integrity. Supporting this, we found no changes in bacterial dissemination in $EC^{\Delta Ahr}$ mice (Fig.3i), or in gut vascular leakage (Extended Data Figure 6g, discussed fully in reviewer point 6 below).

Overall, we are hypothesising here that endothelial AHR fits into AHR-mediated protection against enteric infections by appropriating immune recruitment early in infection and protecting against disease tolerance, while barrier disruption and direct anti-bacterial immunity are more likely roles of the intestinal epithelial and immune compartments.

6. Suggestion II: the manuscript could also be improved by providing an additional functional physiological experiment for example using a labeled lipid tracer or a vascular tracer experiment. As such, the possible differences in functional activity between WT and EC AhR mutant mice may be better displayed

This is an interesting suggestion that we were very interested to address. We took the approach recommended, and previously published by the Petrova lab (ref 7) of injecting 100nm fluorescent microspheres intravenously and assessing leakage into the gut tissue after 5 minutes by whole-mount

imaging. While no differences in barrier integrity between EC^{ΔAhr} and EC^{WT} mice were observed here (Extended Data Figure 6g), we consider this an important *in vivo* functional experiment in the repertoire of this study, allowing a fuller understanding of the precise facets of the role of the AHR in enteric endothelial populations.

References

- 1 Vogel, C. & Marcotte, E. M. Insights into the regulation of protein abundance from proteomic and transcriptomic analyses. *Nat Rev Genet* **13**, 227-232, doi:10.1038/nrg3185 (2012).
- 2 Ricard, N., Bailly, S., Guignabert, C. & Simons, M. The quiescent endothelium: signalling pathways regulating organ-specific endothelial normalcy. *Nat Rev Cardiol* **18**, 565-580, doi:10.1038/s41569-021-00517-4 (2021).
- 3 Metidji, A. *et al.* The Environmental Sensor AHR Protects from Inflammatory Damage by Maintaining Intestinal Stem Cell Homeostasis and Barrier Integrity. *Immunity* **49**, 353-362 e355, doi:10.1016/j.immuni.2018.07.010 (2018).
- 4 Schiering, C. *et al.* Feedback control of AHR signalling regulates intestinal immunity. *Nature* **542**, 242-245, doi:10.1038/nature21080 (2017).
- 5 Shah, K. *et al.* Cell-intrinsic Aryl Hydrocarbon Receptor signalling is required for the resolution of injury-induced colonic stem cells. *Nat Commun* **13**, 1827, doi:10.1038/s41467-022-29098-7 (2022).
- 6 Seo, G. Y. *et al.* LIGHT-HVEM Signaling in Innate Lymphoid Cell Subsets Protects Against Enteric Bacterial Infection. *Cell Host Microbe* **24**, 249-260 e244, doi:10.1016/j.chom.2018.07.008 (2018).
- 7 Bernier-Latmani, J. *et al.* ADAMTS18(+) villus tip telocytes maintain a polarized VEGFA signaling domain and fenestrations in nutrient-absorbing intestinal blood vessels. *Nat Commun* **13**, 3983, doi:10.1038/s41467-022-31571-2 (2022).

Author Rebuttals to First Revision:

Referees' comments:

Referee #1 (Remarks to the Author):

The authors have made efforts to address my previous comments but still have not addressed major concerns I had about the depth of analysis and lack of controls (especially in the new experiments). This manuscript feels immature where in many places there is only a superficial exploration of the data. There is still little confirmation of specificity of the EC clusters or their location identified by the scRNAseq data. Also there is lack of experimental data showing hints of mechanistic roles for intestinal endothelial Ahr signaling, especially regarding the infection models.

1

The authors state there are no reagents to confirm the distinct identity of the the clusters in their EC scRNAseq data, but even a attempt with RNAscope probes on tissue sections would be better than the evidence presented now. For example, the authors mention an artery shear stress cluster only previously identified in the brain (lines 112-13) identified by specific marker genes, which is very exciting. It would be very interesting to know where these endothelial cells are found in the intestinal vessels. The authors should provide evidence of the distinctiveness of these clusters in both identity and physical space. Multicolor RNAscope (perhaps combined with vessel immunostaining?) would be well-suited to provide this evidence.

2

Along the same lines, the authors now show imaging data with the Cyp1a1 reporter mice of steady state Ahr signaling. The authors' single cell data showed that Cyp1a1 is expressed in one LEC cluster and Cap 1 and PCV BECs. Does the GFP reporter expression overlap with markers of the clusters expressing Ahr? This is likely readily feasible by combining GFP staining with RNAscope probes (if no antibodies are available). This would allow further confirmation of scRNAseq data and perhaps allow the authors to probe how the physical location of Ahr signaling relates to the specific roles shown later in the manuscript such as endothelial cell proliferation or inflammation.

I was confused by the authors' explanation of why the the Cyp1a1 reporter data comes with caveats (response to comment 2). They state the reporter only represents a snapshot of Cyp1a1 activity, while in the next sentence state it tracks the lifelong history of AHR ligand exposure. The influence of feeding etc can be controlled through standardization of experimental timing and these reporter data at steady state at least provide some evidence for the physical location of endothelial Ahr signaling and is better than only single cell data.

3

The authors provide new data that in the EC-specific deletion model that tamoxifen efficiently induces GFP expression in BECs and LECs. However, recombination efficiency of floxed alleles is not homogeneous (doi.org/10.1007/s11248-019-00177-8) and endothelial GFP expression after tamoxifen injection is not proof that Ahr is efficiently deleted in endothelial cells in their EC KO mice. The standard level of evidence is to show that that Ahr is lost at the mRNA or protein level in the EC

Ahr KO intestinal ECs, especially after the 14d rest period.

4

The authors provide new clear, exciting data that Esm1 is highly induced in intestinal vessels of EC Ahr KO mice after Ahr ligand treatment (Fig 3f+Fig ED Fig 6e). Endothelial ESM1 is directly induced by VEGFA and TNFa (PMID 25057127+doi.org/10.1074/jbc.271.34.20458). Does loss of Ahr sensitize intestinal endothelial cells to VEGF or TNFa? This could be directly tested by VEGFA signaling blockade (eg DC101) or anti-TNFa antibodies and would provide more mechanistic detail about the role of intestinal Ahr endothelial cell signaling.

5

The authors provide new data suggesting that lack of EC Ahr sensitizes intestinal ECs to VEGFA signaling by injecting VEGFA protein into mice and showing increased BEC proliferation. Injecting VEGF proteins may not be the best way to show VEGF dependence. Eg, there is controversy about experiments where injected VEGFC protein was reported to induce cardiac lymphangiogenesis but this result could not be reproduced with adenoviruses or genetic models (PMC8516448+ PMC7310303+ 10.1038/nature14483). If this approach is to be used, appropriate controls include measuring intestinal BEC proliferation and ESM1 expression between WT control (irrelevant protein) and VEGFA-injected mice and ELISA to detect increased circulating VEGFA levels.

A more robust approach is to analyze ESM1 expression and EC proliferation in control and EC Ahr KO mice after 2w VEGF blockade (with either B20 or DC101 blocking antibodies).

6

The authors now show that female EC Ahr KO mice fare worse than controls in response to Yptb infection and propose that "...our data suggest a role for endothelial AHR in promoting disease tolerance to enteric infection through modulating intestinal immune composition to limit inflammation". Although this result is exciting, how would endothelial Ahr signaling affect the immune composition? Through angiocrine signaling? By altering specific immune cell extravasation or lymphatic transport? Any hint of mechanistic details are missing and this represents a major weakness as there is little context to place endothelial Ahr signaling in the context of known functions of intestinal vessels and limits the appeal of this manuscript to a wide audience.

Referee #2 (Remarks to the Author):

The authors have addressed all of my comments and I would like to congratulate them on a very fine manuscript.

Referee #3 (Remarks to the Author):

I am satisfied with the revised manuscript and rebuttal letter. I have no further comments

Author Rebuttals to Second Revision:

EC AHR paper Review 2 comments

Referee #1:

The authors have made efforts to address my previous comments but still have not addressed major concerns I had about the depth of analysis and lack of controls (especially in the new experiments). This manuscript feels immature where in many places there is only a superficial exploration of the data. There is still little confirmation of specificity of the EC clusters or their location identified by the scRNAseq data. Also there is lack of experimental data showing hints of mechanistic roles for intestinal endothelial Ahr signaling, especially regarding the infection models.

We would like to begin by thanking the reviewer for taking their time to critically assess our manuscript for a second time, and have carefully taken their comments into consideration. Their comments have once again helped to improve our manuscript into a body of work we are very proud of, and we are extremely grateful for this. As specified below in the point-by-point reply, at the request of the reviewer, **we provide new data on *in vivo* VEGFR2 blockade, show further confirmations of EC^{ΔAhr} mice knockout efficiency, provide additional controls in VEGF-A stimulation experiments, show further colocalization evidence in Cyp1a1-reporter mice, and provide new data in infection that links EC phenotype to immune infiltrate (Fig.3f; Extended Data Fig.5g-i; Extended Data Fig.8e; Reviewer Figures 6-8).** As previously, we have formatted changes to the manuscript in blue text.

However, it is important to point out that we previously addressed every comment given by this reviewer as well as reviewers 2 and 3. The reviewer is now suggesting additional experiments that were not requested in the first round of review, as we discuss in the point-by-point reply below.

Our manuscript dissects the role of AHR in enteric endothelial cells for the first time. We begin the manuscript by precisely characterising enteric endothelial populations by scRNAseq. We perform experiments in both human and mice, using transcriptomics and *in vivo* approaches, studying blood and lymphatic endothelial cells, investigating both AHR pathway stimulation and AHR deficiency a) at steady state, b) using an endotoxemia model, and c) an enteric infection model. **Given this comprehensive body of work, the steady state data provided upon the reviewer's original request, and the mechanistic insights provided here, we believe our manuscript meets the necessary criteria for publication in Nature.**

1

The authors state there are no reagents to confirm the distinct identity of the the clusters in their EC scRNAseq data, but even a attempt with RNAscope probes on tissue sections would be better than the evidence presented now. For example, the authors mention an artery shear stress cluster only previously identified in the brain (lines 112-13) identified by specific marker genes, which is very exciting. It would be very interesting to know where these endothelial cells are found in the intestinal vessels. The authors should provide evidence of the distinctiveness of these clusters in both identity and physical space. Multicolor RNAscope (perhaps combined with vessel immunostaining?) would be well-suited to provide this evidence.

While we share the reviewer's excitement for the identification of the new clusters we uncovered with scRNAseq, and agree that identifying the location of these would provide additional insights to the field, **the request to perform RNAscope was not raised in the original review of this manuscript.** The reviewer previously wrote "*there is no verification of cluster marker genes through other means like FACS or imaging.*" To this end, we performed both FACS and immunostaining (Reviewer Figure 1). Setting up multicolour RNAscope would take at least 6 months with substantial optimisation required. The specific request for inclusion of RNAscope data was not made in the original review and the word imaging, as originally suggested by the reviewer, is certainly not synonymous with imaging using RNA probes i.e. RNAscope.

Furthermore, we are not aware of any publication using a combination of immunostaining and multicolour RNAscope on gut tissue. In order to visualise the gut villi vasculature in detail, whole-mount imaging is required¹. RNAscope on whole-mount gut tissue is a protocol that has not been established in the field, and as such could be a protocol paper in itself. To the best of our knowledge, leading gut imaging experts such as Tania Petrova have not established this (Jeremiah Bernier-Latmani – personal communication), while Taija Mäkinen’s group have only established whole mount RNAscope in the mesenteric adipose, not the gut, and not with combined immunofluorescence imaging. **Therefore, we believe this additional request is currently beyond the capabilities of expert groups in gut endothelial biology and consequently beyond the scope of this study.**

2

Along the same lines, the authors now show imaging data with the Cyp1a1 reporter mice of steady state Ahr signaling. The authors’ single cell data showed that Cyp1a1 is expressed in one LEC cluster and Cap 1 and PCV BECs. Does the GFP reporter expression overlap with markers of the clusters expressing Ahr? This is likely readily feasible by combining GFP staining with RNAscope probes (if no antibodies are available). This would allow further confirmation of scRNAseq data and perhaps allow the authors to probe how the physical location of Ahr signaling relates to the specific roles shown later in the manuscript such as endothelial cell proliferation or inflammation.

I was confused by the authors’ explanation of why the the Cyp1a1 reporter data comes with caveats (response to comment 2). They state the reporter only represents a snapshot of Cyp1a1 activity, while in the next sentence state it tracks the lifelong history of AHR ligand exposure. The influence of feeding etc can be controlled through standardization of experimental timing and these reporter data at steady state at least provide some evidence for the physical location of endothelial Ahr signaling and is better than only single cell data.

To clarify our comments further, the Cyp1a1-reporter model was used in this study to analyse AHR responses to stimulation. While this is a very useful model for studying responses to AHR ligands and capacity for ligand stimulation when comparing to vehicle-treated controls, this model is not an ideal tool to study current ongoing AHR activity due to the nature of the fate reporter construct that is activated in any cell that has ever been exposed to AHR ligands over the animal’s history. We appreciate how our previous statement on the ‘snapshot of exposure’ could have been made clearer. With this we meant that only through synchronising the AHR pathway through exogenous ligand administration can we be sure of cells capable of responding, as background/constitutive responses are influenced by a number of factors (e.g. equal ligand availability to each BEC type). We argue that together these caveats limit the ability to answer such questions.

As the reviewer points out, we have already performed experiments investigating the location of Cyp1a1(eYFP)-expressing ECs within the gut vasculature at the original request of the reviewer (Reviewer Figure 2a). We carried out whole-mount imaging on the intestines of vehicle-treated reporter mice showing Cyp1a1 signal was scattered throughout the villi and submucosal vasculature. The possible interpretations of this result are a) constitutively AHR-activated populations capillary 1 are not positionally distinct, or b) previous developmental/ environmental signals make up an unknown proportion of the Cyp1a1(eYFP)+ cells. We are unable to disentangle these two possibilities, so this was included as a reviewer figure instead of within the main manuscript. **We now also provide new data investigating the steady-state Cyp1a1(eYFP) signal in PCVs suggesting an enrichment of AHR activation in PCVs (Reviewer Figure 6).** This is in keeping with a modest enrichment of Cyp1a1(eYFP)⁺ cells within the PCV population identified by flow cytometry (Reviewer Figure 1), and with the scRNAseq data (Fig.2b), but again whether the signal seen here is current AHR activation or historic, cannot be determined. Furthermore, from the imaging, it appears that the arterial capillary feeding the villi are also strongly Cyp1a1(eYFP)⁺, in opposition to their expected very low/absent AHR signalling expected from the scRNAseq, arguing against the utility of this approach for investigating current AHR activity in line with the scRNAseq. As already explained above, RNAscope on whole-mount gut samples is not an established technology and therefore beyond the scope of this study.

3

The authors provide new data that in the EC-specific deletion model that tamoxifen efficiently induces GFP expression in BECs and LECs. However, recombination efficiency of floxed alleles is not homogeneous (doi.org/10.1007/s11248-019-00177-8) and endothelial GFP expression after tamoxifen injection is not proof that Ahr is efficiently deleted in endothelial cells in their EC KO mice. The standard level of evidence is to show that that Ahr is lost at the mRNA or protein level in the EC Ahr KO intestinal ECs, especially after the 14d rest period.

We agree that indeed, in some cases, recombination efficiency at the reporter locus does not directly correlate with gene deletion deficiency. **We now provide data from our RNAseq experiment demonstrating clear lack of *Cyp1a1* in our fully-tamoxifen treated endothelial AHR-deficient mouse model** (Reviewer Figure 7). This is direct functional proof of the efficiency of our knockout. Notably, *Tiparp* and *Ahrr*, both AHR-dependent genes were also highly decreased in EC^{ΔAhr} mice (Fig.3a), while *Ahr* mRNA were not significantly reduced, which is expected given that to the genetic model is based on a deletion of only exon 2 of the *Ahr* gene. Deletion efficiency cannot be shown at protein level due to lack of a reliable antibody for detecting AHR in mice (see papers leading labs in AHR research - Gitta Stockinger, Marc Veldhoen, Paige Lawrence and others). **Lack of functional responsiveness downstream of AHR, as evidenced by lack of *Cyp1a1* expression, is in our hands the strongest possible evidence supporting the validity of our experimental approach.**

4

The authors provide new clear, exciting data that Esm1 is highly induced in intestinal vessels of EC Ahr KO mice after Ahr ligand treatment (Fig 3f+Fig ED Fig 6e). Endothelial ESM1 is directly induced by VEGFA and TNFa (PMID 25057127+doi.org/10.1074/jbc.271.34.20458). Does loss of Ahr sensitize intestinal endothelial cells to VEGF or TNFa? This could be directly tested by VEGFA signaling blockade (eg DC101) or anti-TNFA antibodies and would provide more mechanistic detail about the role of intestinal Ahr endothelial cell signaling.

We thank the reviewer for this interesting question. While we were initially sceptical about incorporating VEGFR2 blockade into our EdU feeding protocol (we reasoned that given the low level of homeostatic proliferation observed in intestinal ECs, reducing this further would be below the resolution of difference), we can report here that the reviewer's suggestion has directly led to a very interesting mechanistic insight that elevates this section of the manuscript. We carried out the experiment as the reviewer requested – administering DC101 or control IgG antibodies concomitant with EdU feeding in a similar strategy employed as by the Petrova lab^{2,3}. The overall level of proliferation was not decreased over two weeks in DC101-treated compared to control IgG-treated mice, but interestingly **where the proliferative advantage of KO BEC was maintained in IgG control, DC101 abrogated this difference, suggesting - as the reviewer hypothesised - that loss of AHR sensitised intestinal endothelial cells to VEGF-A stimulation (Fig.3f)**. This new insight has been included in the main figure and results sections, as well as mentioned in the discussion and introduction.

5

The authors provide new data suggesting that lack of EC Ahr sensitizes intestinal ECs to VEGFA signaling by injecting VEGFA protein into mice and showing increased BEC proliferation. Injecting VEGF proteins may not be the best way to show VEGF dependence. Eg, there is controversy about experiments where injected VEGFC protein was reported to induce cardiac lymphangiogenesis but this result could not be reproduced with adenoviruses or genetic models (PMC8516448+ PMC7310303+ 10.1038/nature14483). If this approach is to be used, appropriate controls include measuring intestinal BEC proliferation and ESM1 expression between WT control (irrelevant protein) and VEGFA-injected mice and ELISA to detect increased circulating VEGFA levels.

A more robust approach is to analyze ESM1 expression and EC proliferation in control and EC Ahr

KO mice after 2w VEGF blockade (with either B20 or DC101 blocking antibodies).

As discussed above (point 4), we have now carried out DC101-mediated VEGFR2 blockade, showing that *Ahr*-deficient BEC are more responsive to VEGF-A signals than WT controls. Further, **we provide additional data in relation to our previous VEGF-A stimulation experiment (Fig.3e) that shows a clear, significant increase in proliferation (EdU⁺ detection) in VEGF-A-treated compared to vehicle-treated mice (Reviewer Figure 8)**. All EdU only control mice were treated with PBS vehicle controls (Reviewer Figure 8). Therefore, we have also validated that *in vivo* VEGF-A administration works to increase proliferation in our study.

We performed ESM1 staining in answer to a question from a different reviewer, reviewer 3, as part of our assessment of angiogenic changes. Reviewer 3 accepted our data in answer to their question.

6

The authors now show that female EC *Ahr* KO mice fare worse than controls in response to *Yptb* infection and propose that "...our data suggest a role for endothelial AHR in promoting disease tolerance to enteric infection through modulating intestinal immune composition to limit inflammation". Although this result is exciting, how would endothelial *Ahr* signaling affect the immune composition? Through angiocrine signaling? By altering specific immune cell extravasation or lymphatic transport? Any hint of mechanistic details are missing and this represents a major weakness as there is little context to place endothelial *Ahr* signaling in the context of known functions of intestinal vessels and limits the appeal of this manuscript to a wide audience.

At the first round of review of this manuscript, this reviewer stated that "*the conclusion that endothelial AHR plays a role in host resistance to enteric infections requires more evidence.*" **We have since provided such evidence**, showing statistically significant decreases in survival in infected female $EC^{\Delta Ahr}$ mice, and altered immune infiltration. Furthermore, as the reviewer mentions above, new data generated during the review process are exciting additions: **we show for the first time that endothelial-specific AHR deficiency leads to a survival defect in enteric infection**. This is despite intact AHR signalling remaining in the gut immune, and epithelial cell compartments that are canonically the major players in barrier immunity. **We believe this a major breakthrough and would be of broad interest to the readership of nature.**

Nonetheless, at the new request of the reviewer, and to link responses to enteric infection to EC phenotype we carried out another *Yptb* infection on $EC^{\Delta Ahr}$ mice or EC^{WT} controls, and examined the EC activation/inflammatory profile at day 3 p.i. Interestingly, we observed increased expression of inflammatory markers in the $EC^{\Delta Ahr}$ mice, including VCAM-1, BST-2 and CD86 which **links EC phenotype to immune infiltrate and is suggestive of a mechanism whereby the activation state of the mutant EC is likely causative (Extended Data Fig.8e)**. We further speculate in the discussion that this may occur through an immune recruitment mechanism.

In response to the reviewer's previous requests we already provided substantial new supporting data on *Yptb* infection: demonstrating a defect in survival in female $EC^{\Delta Ahr}$ mice, without differences in bacterial burden in the gut or dissemination to peripheral organs, but with key changes to immune cell composition in the intestine of these animals. This is in keeping with the disease tolerance-promoting role of endothelial AHR which is demonstrated in the supporting back-to-back manuscript (Major et al.) using the same $EC^{\Delta Ahr}$ mouse model in lung inflammation. Furthermore, *Yptb* infection is just one challenge model we employed alongside LPS-mediated endotoxaemia. On top of this, we compiled a body of work at steady state at the previous request of this reviewer.

Lastly, as we observed multiple immune cell population changes in $EC^{\Delta Ahr}$ mice after *Yptb* infection, likely each with distinct contributions to the protective mechanism, it is unclear what the reviewer expects us to do experimentally. Would the "mechanistic details" requested require the creation of further genetic crosses with various deletions of different immune populations or endothelial-expressed

genes? Or additional deletion or reconstitution of certain genes in combination with the EC^{ΔAhr} mice we already created? If so, **this kind of experimental work would take somewhere in the region of 1.5-2 years and would be firmly beyond the scope of this manuscript.**

Referee #2:

The authors have addressed all of my comments and I would like to congratulate them on a very fine manuscript.

We are extremely appreciative of the reviewer's comments. Their acknowledgement of our hard work is very much appreciated. Their excellent suggestions have improved our manuscript and we are very grateful to them for this.

Referee #3:

I am satisfied with the revised manuscript and rebuttal letter. I have no further comments.

We are extremely appreciative of the reviewer's comments. Their acknowledgement of our hard work is very much appreciated. Their excellent suggestions have improved our manuscript and we are very grateful to them for this.

References

- 1 Bernier-Latmani, J. & Petrova, T. V. High-resolution 3D analysis of mouse small-intestinal stroma. *Nat Protoc* **11**, 1617-1629, doi:10.1038/nprot.2016.092 (2016).
- 2 Bernier-Latmani, J. *et al.* DLL4 promotes continuous adult intestinal lacteal regeneration and dietary fat transport. *J Clin Invest* **125**, 4572-4586, doi:10.1172/JCI82045 (2015).
- 3 Bernier-Latmani, J. *et al.* ADAMTS18(+) villus tip telocytes maintain a polarized VEGFA signaling domain and fenestrations in nutrient-absorbing intestinal blood vessels. *Nat Commun* **13**, 3983, doi:10.1038/s41467-022-31571-2 (2022).

Reviewer Reports on the Second Revision:

Referees' comments:

Referee #4 (Remarks to the Author):

In the study under review (2022-03-03578), Schiering and co-workers investigate the role of AHR ligands in modulating the endothelial phenotype in the small intestine. Starting point is the scRNAseq-based analysis of the transcriptomic response to AHR ligands of small intestinal (si) blood (BEC) and lymphatic (LEC) endothelial cells. The EC response was pronounced and a bioinformatic analysis identified 7 BEC and 4 LEC response clusters. Endothelial AHR activation suppressed proliferation by desensitizing ECs to VEGF-A, acted anti-inflammatory by dampening the response to inflammatory stimuli; like LPS; and provided limited protection to an enteric infection (Yptb). The consequences of AHR stimulation on endothelial proliferative and pro-inflammatory phenotype and the intestinal inflammatory response have been analysed by the authors in superb detail and with exquisite detail. A minor piece of information that one may find lacking would be if and to what extent altered EC proliferation, either in EC Δ AHR mice or under prolonged dietary stimulation, would over time change the structure of the vessel bed e.g. of the villi in the si (parameters to analyse: branching, diameter, straightness, overall villus length)? Certainly, the underlying EC profiling is equally extensive and comprehensive. What the manuscript cannot provide by the extensive and elaborate analysis is a mechanistic connection / explanation how the distinct EC transcriptional phenotypes described translate into the reported biological outcomes. A seemingly straightforward approach towards this goal would be to image the distinct endothelial populations within the tissue or to verify them independently by cytometry. Knowledge on their location and behaviour during the investigated challenges e.g. enteric infection would provide important information. The authors explored immunostaining based approaches in conjunction with wholemount imaging and cytometry, however, due to a lack of suitable markers they were unable to identify the putative populations.

Point to consider there: It appears that the majority of the imaging analysis was focussed on wholemount stained intestinal villi, in particular, the lymphatic vessel beds of the submucosa, Peyer's patches and serosa are not or not prominently documented in the manuscript. As all of these tissue areas contain lymph capillaries, their analysis might have been informative.

A suggestion raised during the review process was to circumvent the lack of suitable antibodies through a combination of RNA-FISH and immunostaining e.g. multicolour RNA-scope in conjunction with vascular markers. This suggested technology is technically demanding. It must be carefully optimized for different RNA-probes and success highly depends on the transcript level. If the tissue distribution of the cells belonging to a particular cluster was dispersed their identification will require wholemount imaging or at least analysis of thick sections (150-300 μ m). Presently there are no published data on the application of the RNA-scope technology to this kind of sample in adult mouse tissue. Analysis of thin sections (also suggested) is technically more feasible, but would only be constructive if the ECs of a particular transcriptional group clustered spatially, in part or entirely, in a localized vessel area. Generally, establishment of a multi-color RNAscope / immunostaining-protocol is a significant task that may take considerable development time and its feasibility within the restraints of the advanced review process of this manuscript appears questionable.

In summary, the manuscript describes, based on an excellent bioinformatics analysis, the

transcriptional response of distinct siECs populations to AHR stimulation. The study goes on to document in detail consequences of the ensuing endothelial behaviour, which translate into an altered inflammatory response and promotion of infection tolerance. However, the involvement of AHR receptor signalling in the regulation of inflammation and proliferation during development has been previously reported. In this study, attempts remained unsuccessful to map back the identified EC transcriptional populations in the tissue. Hence, the study does not accomplish to demonstrate selectivity in the response of ECs within the blood and lymphatic vessel bed to AHR stimulation. Therefore, the mechanisms how the change in inflammatory response and the altered susceptibility to infection comes about as a consequence AHR ligand exposure remains, despite a large amount of experimental data provided, vague.

Minor point: Typo in the headline to extended data Fig. 5 Intestinal LEC.